

# An improved and extended parameterization of the $CO_2$ $15\,\mu\mathrm{m}$ cooling in the middle/upper atmosphere

Manuel López-Puertas[1], Federico Fabiano[2], Victor Fomichev[3,†], Bernd Funke[1], and Daniel R. Marsh[4]

[1]Instituto de Astrofísica de Andalucía, CSIC, Granada, Spain.
[2]Institute of Atmospheric Sciences and Climate, CNR-ISAC, Bologna, Italy
[3]York University, Centre for Research in Earth and Space Science, Toronto, Canada
[4]School of Physics & Astronomy, University of Leeds, UK
[†]Retired since May 2016.

**Correspondance:** M. López-Puertas (puertas@iaa.es)

**Abstract.** The radiative infrared cooling of $CO_2$ in the middle atmosphere, where it emits under non-Local Thermodynamic Equilibrium (non-LTE) conditions, is a crucial contribution to the energy balance of this region and hence to establishing its thermal structure. The non-LTE computation is too CPU time-consuming to be fully incorporated in climate models and hence it is parameterized. The most used parameterization of the $CO_2$ $15\,\mu\mathrm{m}$ cooling for the Earth's middle and upper atmosphere was developed by Fomichev et al. (1998). The valid range of this parameterization with respect to $CO_2$ volume mixing ratios (VMR) is, however, exceeded by the $CO_2$ of several scenarios considered in the Coupled Climate Model Intercomparison Projects; in particular, the abrupt-4×$CO_2$ experiment. Therefore, an extension, as well as an update, of that parameterization is both needed and timely. In this work, we present an update of the parameterization developed by Fomichev et al. (1998), which now covers $CO_2$ volume mixing ratios in the lower atmosphere from ∼0.5 to over 10 times the $CO_2$ pre-industrial value of 284 ppmv (i.e., 150 ppmv to 3000 ppmv). Furthermore, it is improved by using a more contemporary $CO_2$ line list and collisional rates that affect the $CO_2$ cooling rates. Overall, accuracy is improved when tested against reference line-by-line calculations and by using measured global temperature profiles of the middle atmosphere. On average the errors are below $0.5\,\mathrm{K\,day^{-1}}$ for the present-day and lower $CO_2$ VMRs. The errors increase to ∼1–2 $\mathrm{K\,day^{-1}}$ at altitudes between 100–120 km for $CO_2$ concentrations of two to three times the preindustrial values. For very high $CO_2$ concentrations (four to ten times the pre-industrial abundances) the errors are below ∼1 $\mathrm{K\,day^{-1}}$ for most regions and conditions, except at 110–120 km where the parameterization overestimates them by ∼1.5%. When applied to a large dataset of global (pole-to-pole and four seasons) measured temperature profiles, the errors of the parameterization are generally below $0.5\,\mathrm{K\,day^{-1}}$, except between $5\cdot10^{-3}$ hPa and $3\cdot10^{-4}$ hPa (∼85–95 km), where they can reach biases of 1–2 $\mathrm{K\,day^{-1}}$. However, for elevated stratopause events, it underestimates the cooling rates by 3–7 $\mathrm{K\,day^{-1}}$ (∼10%) at altitudes of 80–100 km and the parameterized cooling rates show a large spread when compared to reference calculations.

## 1 Introduction.

Carbon dioxide is the major infrared cooler of the atmosphere from the lower stratosphere up to the lower thermosphere, at which height emission by nitric oxide becomes important (López-Puertas and Taylor, 2001). However, the $CO_2$ infrared



emissions in the $\nu_2$ bands near 15 $\mu$m that are responsible for the cooling, are in non-local thermodynamic equilibrium (non-LTE) above around 70 km (López-Puertas and Taylor, 2001). The computation of the cooling under those non-LTE conditions requires the solution of the radiative transfer equation (RTE) which is a non-local problem and requires a large amount of CPU time. Therefore, the solving the RTE in general circulation models (GCMs) or climate models that extend in height above the stratopause is impractical and efficient parameterizations of the $CO_2$ infrared cooling have been developed and implemented
in such models.

The most used parameterizations of the $CO_2$ 15 $\mu$m cooling for the Earth's middle/upper atmosphere was developed by Fomichev et al. (1998). That parameterization is applicable for a limited range of $CO_2$ abundances, up to double the pre-industrial $CO_2$ concentration. Nowadays, however, with the rapid increase of the $CO_2$ concentration in the atmosphere, and its expected increase in the coming decades, climate model projections are being carried out in much higher $CO_2$ scenarios
(van Vuuren et al., 2011; O'Neill et al., 2016), that is, even quadrupling the pre-industrial $CO_2$ abundance. For example, such scenarios are considered in the Coupled Climate Model Intercomparison Projects. Therefore, parameterizations coping with such large $CO_2$ concentrations are highly demanded. That is the aim of this work. Several parameterizations of the $CO_2$ 15 $\mu$m cooling rates have been developed in the past. In the case of the Earth's atmosphere, it worth to mention the comprehensive review of the early works reported by Fomichev et al. (1998), the summary presented in Sec. 5.8 of López-Puertas and Taylor
(2001), and the more recent work of Feofilov and Kutepov (2012). For Mars and Venus atmospheres, where $CO_2$ is the most abundant species, the problem has been tackled in several studies, e.g. López-Valverde et al. (1998, 2008) and Gilli et al. (2017, 2021). In our case we have the option of developing a completely new parameterization, to adapt other $CO_2$ parameterizations (as those cited above), or to extent and improve the parameterization of Fomichev et al. (1998). Attending mainly to practical reasons of promptness, we opted for the later.

The paper is structured as follows. A very basic description of the parameterization is presented in Sec. 2. Sec. 3 describes the input atmospheric parameters used in the parameterization and required for calculating the reference cooling rates. In Sec. 4 we describe the calculations of the reference LTE and non-LTE cooling rates. The detailed description of the parameterization is presented in Sec. 5. The testing and accuracy of the parameterization against the reference cooling rates and for currently measured temperature fields of the middle atmosphere are discussed in Secs. 6 and 7, respectively. The previous cooling rate
parameterization was used together with a parameterization of the $CO_2$ near-IR heating rates (Ogibalov and Fomichev, 2003). As we have extended the former to higher $CO_2$ volume mixing ratios (vmrs), and we do not plan to extend the latter to higher $CO_2$ vmrs in the near future, we assess in Sec. 8 that the current near-IR heating parameterization can still be safely used with $CO_2$ vmrs up to at least five times the pre-industrial values. In Sec. 9, we summarize the main conclusions of the study.

## 2   Framework of the parameterization

As discussed above, this parameterization is essentially based on that of Fomichev et al. (1998). For computing the $CO_2$ cooling rate, the atmosphere is divided into five regions (see Fig. 11): the LTE and four different non-LTE regions. The method and approximations for computing the cooling rates in those regions are described in detail in Sec. 5. The new parameterization has



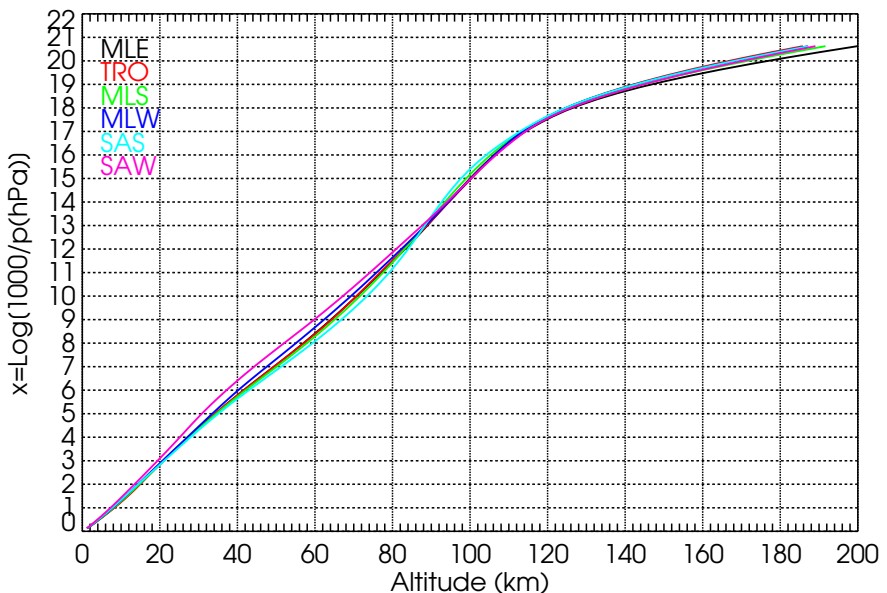

**Figure 1.** The relationship between pressure and geometrical altitude for the reference temperature profiles.

a finer grid and, because it has been developed to cover a larger range of $CO_2$ vmrs, the different non-LTE regions were revised and usually, their upper boundaries were extended to upper altitudes. The scheme consists of 83 levels in $x = \log(1000/p(hPa))$,

covering from $x = 0.125$ to $x = 20.625$ spaced by 0.125. The relationship between pressure and the geometrical altitude for the reference temperature profiles is shown in Fig. 1. To a first approximation, the geometric altitude $z$ below $\sim 120$ km is related to $x$ by $z(km) \approx 7x$.

The parameterization computes cooling rates for given inputs of temperature and concentrations of $CO_2$, $O(^3P)$, $O_2$ and $N_2$ as a function of pressure. Further, the collisional deactivation of $CO_2(v_2)$ levels by the main atmospheric molecules ($N_2$ and

$O_2$) and by $O(^3P)$, can also be prescribed. To compute the different coefficients employed by the parameterization (see Sec. 5), reference LTE and non-LTE cooling rates are required (see Secs. 4.1 and 4.2). These are calculated for selected reference atmospheres, which are described in the next section.

## 3    The reference atmospheres

### 3.1    Temperature

We used the same six pressure-temperature reference atmospheres as in Fomichev et al. (1998) for altitudes below $\sim 120$ km. Above this altitude, they were extended up to $\sim 200$ km with the empirical US Naval Research Laboratory Mass Spectrometer Incoherent Scatter Radar version 2.0 (MSIS2) model (Emmert et al., 2021) for medium conditions of solar activity, $F_{10.7} = 103$ sfu (June 2011) for all atmospheres except for MLE which was $F_{10.7} = 142$ sfu (September 2011). These six $p$-$T$ profiles cover very well the envelope of the climatological zonal mean temperatures of the current middle atmosphere, e.g. as





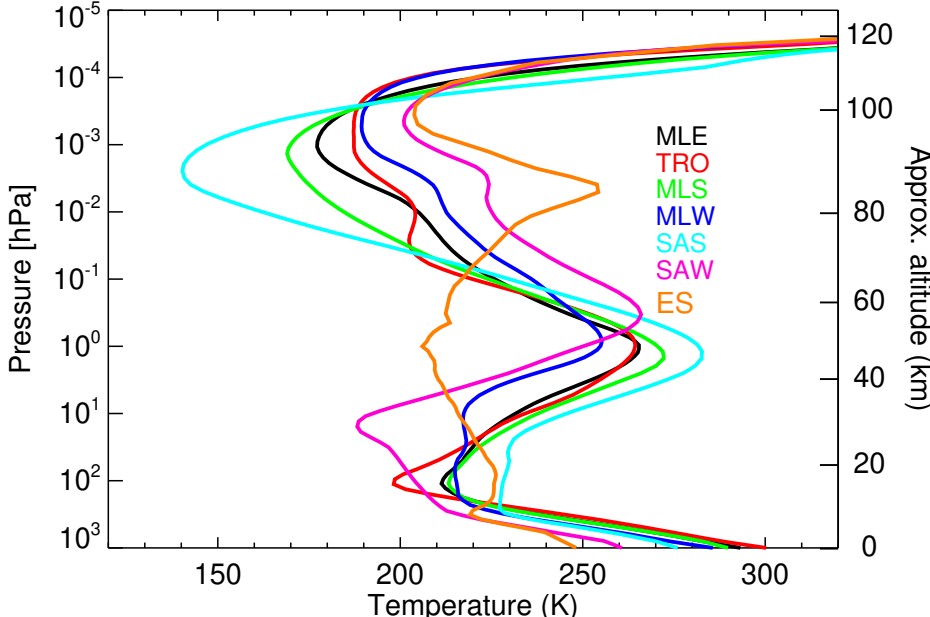

**Figure 2.** The six temperature profiles used in the reference calculations up to the lower thermosphere. MLE: midlatitude equinox (September, 40°N); TRO: tropics (June, equator); MLS: midlatitude summer (June, 40°N); MLW: midlatitude winter (June, 40°S); SAS: subarctic summer (June, 70°N); and SAW: subarctic winter (June, 70°S). For the entire altitude range see Fig. 11. A profile typical of an elevated stratopause event (ES, mean of MIPAS temperature measured for latitudes of 70°N–90°N for 15 February 2009) is also shown for comparison and is discussed in Sec. 7.2. Here and in the following figures, the geometric altitude is approximate, corresponding to the pressure/altitude relationship of the MLE reference atmosphere.

measured by MIPAS from 2007 to 2012 (see Sec. 7). Of course, they do not cover the short scale temporal and spatial temperature variability (see, Fig. 20). The performance of the parametarization for such variability is addressed in Sec. 7. Further, the range of the six temperature profiles do not cover well the episodes of stratospheric warming with elevated stratopause. During these events, the altitude region of the typical stratopause, at about 50 km, is much colder, being even 50 K colder than during normal conditions, and the altitude of the typical mesopause, near 85–90 km, is warmer by a similar amount (see Fig. 2). For

these conditions the temperature profile is nearly isothermal from the tropopause up to about 0.1 hPa and exhibits an inversion above, with a peak near the mesopause. We anticipate that for these conditions the error incurred by this parameterization can be significant (see Sec. 7.2).

We should also mention that the envelope of these reference atmospheres does not fully cover the predicted temperatures for the end of this century for projections with high $CO_2$ emissions. In particular, Whole Atmosphere Community Climate Model

(WACCM) simulations for this century under the RCP6.0 scenario (Marsh, 2011; Marsh et al., 2013; Garcia et al., 2017) yields zonal mean temperatures which are colder in the middle atmosphere. In order to cover such predictions, the envelope of the six $p$-$T$ profiles assumed here would have to be widened by about $-30$ K in the upper stratosphere and by about $-20$ K in the mesosphere. The parameterization accuracy for such predicted temperatures has not been fully assessed in this work as we



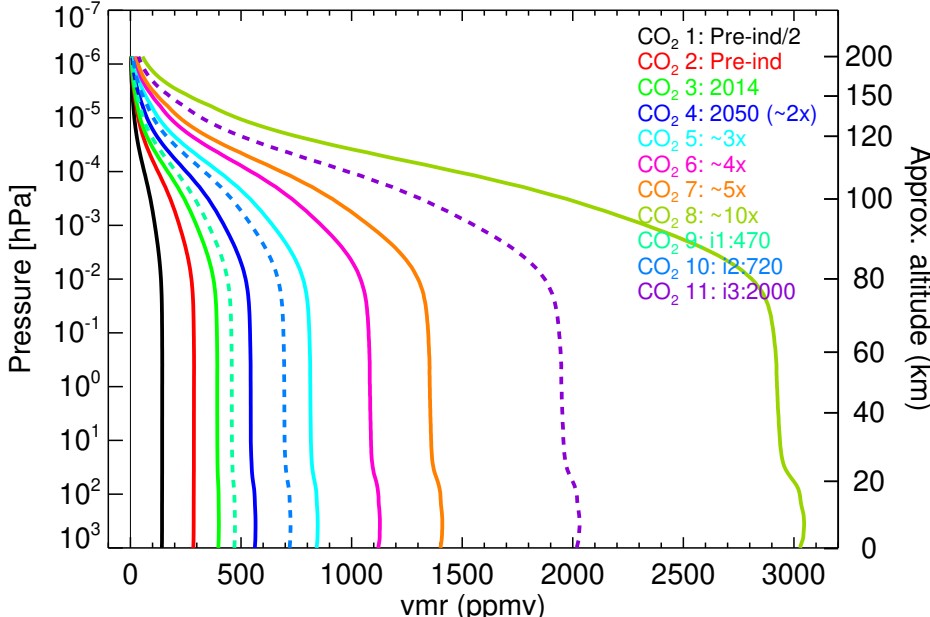

**Figure 3.** $CO_2$ volume mixing ratio profiles used in this work. Solid lines: the profiles used in the reference calculations; dashed lines: those used to test the parameterization.

have considered only the projection of high $CO_2$ vmr profiles but not the corresponding predicted temperature fields. This will

be the subject of future work.

### 3.2   $CO_2$, $O(^3P)$, $O_2$ and $N_2$ abundances

The valid range of the parameterization of Fomichev et al. (1998) with respect to $CO_2$ volume mixing ratios is exceeded by the $CO_2$ concentration of several scenarios considered in the 6th Coupled Climate Model Intercomparison Project (CMIP6); in particular for the $4\times CO_2$ experiment. Several $CO_2$ scenarios have been proposed for the future. That is, van Vuuren et al.

(2011) proposed the scenario RCP2.6, which reaches tropospheric $CO_2$ values near 1000 ppmv by the end of the century. Likewise, Meinshausen et al. (2011) suggested the high $CO_2$ scenario of RCP8.5 (CMIP5) which has $CO_2$ concentrations of 2000 ppmv in the second half of the 23rd century; or even higher than 2000 ppmv (e.g., SSP5-8.5 in CMIP6, see O'Neill et al., 2016). Here we used a wide range of tropospheric $CO_2$ values ranging from about half of the pre-industrial (1851) value of 285 ppmv, to about ten times this value (see Fig. 3). The specific profiles were built from a WACCM run under the CMIP6

SSP5-8.5 scenario (Marsh, 2011; Marsh et al., 2013; Garcia et al., 2017). Global annual mean profiles of $CO_2$ were taken from WACCM simulations for years: 1851 (pre-industrial), $CO_2$ profile #2; 2014, $CO_2$ profile #3; 2050, $CO_2$ profile #4 ($\sim 2\times$pre-industrial); and 2099, $CO_2$ profile #6 ($\sim 4\times$pre-industrial). In addition, we set up, the low $CO_2$ profile (#1) by halving the pre-industrial profile #2, the intermediate $CO_2$ profile #5 ($\sim 3\times$pre-industrial profile) from the mean of WACCM outputs for 2050 and 2099, the high $CO_2$ profile #7 ($\sim 5\times$pre-industrial profile) by multiplying WACCM output for 2099 by a factor of

1.25, and the highest $CO_2$ profile #8 ($\sim 10\times$pre-industrial profile) by multiplying WACCM output for 2099 by a factor of 2.7.





In addition to those $CO_2$ vmr profiles we also composed intermediate profiles #9, #10 and #11, for testing the parameterization (see Sec. 6.2), shown in Fig. 3 with dashed lines. Profiles #9 and #11 were obtained by multiplying WACCM outputs for the years of 2050 and 2099 by factors of 0.979 and 1.8, respectively. Profile #10 was calculated by weighting the WACCM annual mean for years 2050 and 2099 by 0.76875 and 0.256250, respectively. WACCM provides the $CO_2$ vmr profiles up to about 110    130 km. Above that altitude, they were calculated by using a WACCM-X run for 2008, which provides $CO_2$ vmr up to near 500 km, and scaling them, in pressure levels, by the $CO_2$ value of the corresponding $CO_2$ profile at a pressure of $5 \cdot 10^{-6}$ hPa.

As discussed above, the parameterization requires the $N_2$, $O_2$ and $O(^3P)$ volume mixing ratio profiles for the six $p$-$T$ reference atmospheres. They were taken from the MSIS2 model (Emmert et al., 2021) and are shown in Fig. A1.

## 4   Cooling rates for the reference atmospheres

We describe in this section the non-LTE cooling rates used as a reference. To compute the coefficients of the parameterization and the boundaries of the different layers they also require the calculations of the cooling rates in LTE, which are also described in this section. Further, we have assessed the accuracy of the LTE cooling rates by comparing them with those calculated by an independent code, the Reference Forward Model (RFM, Dudhia, 2017).

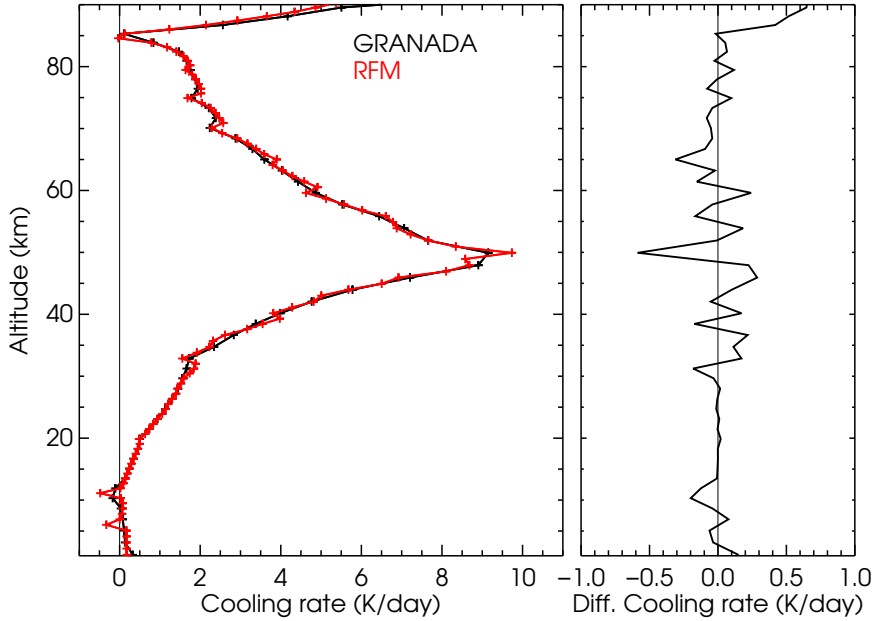

**Figure 4.** LTE cooling rates for the US standard temperature profile and the $CO_2$ vmr of Fomichev et al. (1998) computed by the GRANADA algorithm and the RFM code.





## 4.1 Reference LTE cooling rates

The LTE cooling rates have been computed using a modified Curtis matrix formulation (Funke et al., 2012). That formulation uses as the basis for the radiative transfer calculations (e.g., the optical depths and transmittances) the Karlsruhe Optimised and Precise Radiative Transfer Algorithm (KOPRA, Stiller et al., 2002) code. KOPRA is a well-tested general-purpose line-by-line radiative transfer model that includes all the known relevant processes for performing accurate radiative transfer calculations in planetary atmospheres. We used the $CO_2$ line list of HITRAN 2016 (Gordon et al., 2017) and the line shapes were modelled

with a Voigt profile including the pressure and temperature dependencies of the Doppler and Lorentz halfwidths. The flux transmittances were computed using a 10-point Gaussian quadrature. The wavenumber grid was $0.0005\,cm^{-1}$. The LTE cooling rates have been computed for the $CO_2$ bands associated with the vibrational states of the $\nu_1\nu_2$ mode manifold covering the spectral range from $540\,cm^{-1}$ to $800\,cm^{-1}$ in intervals of $10\,cm^{-1}$. All bands listed in HITRAN 2016 for the six most abundant isotopes in those spectral regions were included in the calculation. For reference, the accurate cooling rates computed assuming

LTE conditions for the six $p$-$T$ profiles and the reference eight $CO_2$ vmrs are shown in Figs. B1 and B2.

In order to assure the accuracy of these LTE cooling rates we have compared them with those obtained with another very well tested and widely used radiative transfer code, RFM (Dudhia, 2017). This code has been used in many studies relevant to the MIPAS instrument (Fischer et al., 2008) and for the retrieval of MIPAS level 2 data obtained by the University of Oxford. It worth mentioning that RFM uses a classical Curtis matrix method (double flux transmittance differences) while we use the

modified Curtis matrix method. Fig. 4 shows the results of the comparison for the US standard temperature profile and the $CO_2$ vmr of Fomichev et al. (1998). Here we used a common fine altitude grid of $0.5\,km$. We see that the agreement between both codes is very good, with differences at most of the altitudes smaller than $0.1$–$0.2\,K\,day^{-1}$. Note that some of the major differences appear to be associated with small oscillations in the RFM results.

The same formulation has been used to calculate the Curtis matrices of all the $CO_2$ $\nu_2$ bands which are required to compute

the coefficients of the parameterization in the LTE region (see Sec. 5).

## 4.2 Reference non-LTE cooling rates

The reference line-by-line non-LTE cooling rates have been computed by using the GRANADA non-LTE code. The details of the method for solving the system of equations for $CO_2$ are given in Funke et al. (2012). In addition to the solution of the statistical and radiative transfer equations described in that work for the calculation of the non-LTE populations of the $CO_2$

levels, here, in order to compute accurate non-LTE cooling rates and to account for the overlapping between the different $CO_2$ $\nu_2$ bands, we included an additional final iteration computing the radiation fields in all the bands by using the Lambda iteration method. This algorithm shares the radiative transfer algorithm with KOPRA (Stiller et al., 2002). Thus, the details about the radiative transfer calculation related to KOPRA, e.g. line shape, spectroscopic data, wavenumber grid, etc., given in the LTE Sec. 4.1 above, applies also to the non-LTE calculations described here.

For this case of non-LTE cooling rates, each ro-vibrational band contributes according to the non-LTE populations of their upper and lower levels. The non-LTE cooling rates calculated here comprise 16 $\nu_2$ vibrational bands emitting/absorbing in





the $15\,\mu$m region. That is, the fundamental $\nu_2$ band, three first hot $\nu_2$ bands and seven $\nu_2$ second hot bands of the major $CO_2$ isotopologue, and the $\nu_2$ fundamental bands of isotopologues $^{16}O^{13}C^{16}O$, $^{16}O^{12}C^{18}O$, $^{16}O^{12}C^{17}O$, $^{16}O^{13}C^{18}O$ and $^{16}O^{13}C^{17}O$. The contribution of other weaker $\nu_2$ bands arising from higher $v_2$ levels, e.g. $v_2$=4, 5 or 6, are included in the

calculation but have negligible contributions for the conditions of the Earth's atmosphere.

**Table 1.** Main collisional processes affecting the $CO_2$ vibrational levels included in the calculations of the non-LTE cooling rates (extracted from Table 5 of Funke et al., 2012).

| No. | Process | Rate coefficient[†] |
|---|---|---|
| 1a | $CO_2{}^i(v_2) + N_2 \rightleftharpoons CO_2{}^i(v_2-1) + N_2$ | $k\times7\times10^{-17}\sqrt{T}+6.7\times10^{-10}\exp(-83.8/T^{1/3})$[‡] |
| 1b | $CO_2{}^i(v_2) + O_2 \rightleftharpoons CO_2{}^i(v_2-1) + O_2$ | $k\times7\times10^{-17}\sqrt{T}+1.0\times10^{-9}\exp(-83.8/T^{1/3})$[‡] |
| | | ($k=1, 3, 4.5, 6.75, 9, 12$ for $v_2$=1, ..., 6) |
| 1c | $CO_2{}^i(v_2) + O \rightleftharpoons CO_2{}^i(v_2-1) + O$ | $3.5\times10^{-13}\sqrt{T}+2.32\times10^{-9}\exp(-76.75/T^{1/3})$ |
| 2a | $CO_2{}^i(v_2,v_3)+N_2 \rightleftharpoons CO_2{}^i(v_2+2, 3\text{ or }4,v_3-1)+N_2$ | $1.1\times10^{-15}+1.14\times10^{-10}\exp(-72.3/T^{1/3})+$ $2.3\times10^{-40}T^9$ |
| 2b | $CO_2{}^i(v_2,v_3)+O_2 \rightleftharpoons CO_2{}^i(v_2+2, 3\text{ or }4,v_3-1)+O_2$ | $1.82\times10^{-15}+3.1\times10^{-11}\exp(-63.3/T^{1/3})+$ $2.0\times10^{-31}T^6$ |
| 2c | $CO_2{}^i(v_2,v_3)+O \rightleftharpoons CO_2{}^i(v_2+2, 3\text{ or }4,v_3-1)+O$ | $2\times10^{-13}(T/300)^{1/2}$ |
| 3 | $CO_2{}^i(v_2,v_3)+CO_2{}^i \rightleftharpoons CO_2{}^i(v_2+1,v_3-1)+CO_2{}^i(v_2=1)$ | $3.6\times10^{-13}\exp(-1660/T+176948/T^2)$ |
| 4 | $CO_2{}^i(v_2,v_3)+O_2 \rightleftharpoons CO_2{}^i(v_2+1,v_3-1)+O_2(1)$ | $3\times10^{-15}$ |
| 5a | $CO_2{}^i(v_2)+CO_2{}^i \rightleftharpoons CO_2{}^i(v_2-1)+CO_2{}^i(v_2=1)$ | See Table 6 in Funke et al. (2012) |
| 5b | $CO_2{}^i(v_2)+CO_2{}^j(v_2') \rightleftharpoons CO_2{}^i(v_2-1)+CO_2{}^j(v_2'+1)$ | $i$ =1; $j$ =2–4; $2.35\times10^{-11}$ |
| 6 | $CO_2{}^i(v_2,v_3)+N_2 \rightleftharpoons CO_2{}^i(v_2,v_3-1)+N_2(1)$ | $5.0\times10^{-13}(300/T)^{1/2}$ for $v_2$=0, 1, 3, 5, 7 $7.0\times10^{-13}(300/T)^{1/2}$ for $v_2$=2, 4, 6 |
| 7 | $CO_2{}^i(v_1v_2lv_3) + N_2 \rightleftharpoons CO_2{}^i(v_1'v_2'l',v_3) + N_2$ | $5.4\times10^{-13}$ for $v_2=v_2'=0$ or $v_2$ & $v_2' \neq 0$ $8.1\times10^{-13}$ for all other cases |

[†] Rate coefficient for the forward sense of the process in $cm^3s^{-1}$. [‡] This rate is taken as $10^{-15}$ $cm^3s^{-1}$ for temperatures lower than 150 K (see Funke et al., 2012). $T$ is temperature in K. $i$ and $j$ are different $CO_2$ isotopologues. $i$ =1–6 except as noted. $v_2$ denotes equivalent $2v_1+v_2$ states, e.g., $v_2$=2 is the triad (10002, 02201, 10001).

For the calculations of the non-LTE cooling rates, a collisional scheme and collisional rates are required. Although the collisional rates affecting the $CO_2$ $v_2$ levels are an input parameter for the parameterization, here we have used, for the calculations of the reference cooling rates and for testing the parameterization, the collisional rates described in Funke et al. (2012). They have been recently revised and used in the non-LTE retrieval of temperature from SABER and MIPAS measurements (García-

Comas et al., 2008; García-Comas et al., 2023). The most relevant collisional processes concerning the populations of the levels emitting in the different $\nu_2$ bands described above, and their rates, are listed in Table 1 for easier reference. We should note that these rates and their temperature dependencies are different from those used in the previous parameterization of Fomichev et al. (1998). The values are in general of very similar magnitude, except the $k_{CO_2-O}$ rate (process 1c in Table 1) that has been





considered here with its upper limit. That is, about a factor of two larger than in the parameterization of Fomichev et al. (1998).

This rate coefficient is not well known with uncertainties of the order of a factor of two (see, e.g., García-Comas et al., 2008). While laboratory measurements are in the range of 1.5 to $2 \cdot 10^{-12} \, \mathrm{cm}^3 \mathrm{s}^{-1}$ the values derived from atmospheric observations are close to $6 \cdot 10^{-12} \, \mathrm{cm}^3 \mathrm{s}^{-1}$. Although this rate can be chosen when using this parametarization, we have optimised it for the high value (see Table 1), as this value has been used in the most recent non-LTE retrievals of temperature from SABER and MIPAS measurements. The effects of using half of this value on the cooling rates are discussed in Sec. 6.2. In the comparisons

shown in the next sections, however, we used consistently the collisional rates in Table 1 for the two parameterizations.

The cooling rates near $15 \, \mu$m change very little with the illumination conditions. However, those cooling rates (or more strictly speaking, the flux divergence) of the $CO_2 \, \nu_2$ bands computed by GRANADA under daytime conditions might be affected by some emission from the relaxation and/or redistribution of the solar energy absorbed in the near-infrared bands (see, e.g. López-Puertas et al., 1990). As this absorption/heating is taken into account by the NIR solar heating parameterization

(see Sec. 8), all the non-LTE cooling rates computed here have been performed under nighttime conditions.

The results for the accurate, line-by-line non-LTE cooling rates computed for the six reference $p$-$T$ profiles and the eight $CO_2$ vmrs are shown in Fig. 5 from the stratosphere up to the lower thermosphere, and in Fig. C1 for the upper part of the parameterization, i.e., above 80 km. We observe that the altitude distribution of the cooling rates depends very much on the temperature profile. This is the major difficulty in building the parameterization. A general common feature is the maximum

near the stratopause because at these altitudes the non-LTE cooling rates do not differ significantly from those in LTE and these are mainly driven by the high temperature of this region. Above the stratopause, the total non-LTE cooling rates depend very much on the contributions of the different bands, e.g the $\nu_2$ fundamental band of the major isotopologue (FB), the contributions of the first and hot bands (Hots) and those of the $\nu_2$ bands of the five minor isotopologues. These contributions are shown in Fig. 6 for the contemporary $CO_2$ vmr profile (#3) and the six $p$-$T$ profiles. The non-LTE cooling rates generally decrease with

altitude above the stratopause, reaching a minimum near the mesopause for several $p$-$T$ profiles, see, e.g., the TRO and MLS atmospheres in Fig. 6. The cooling can be even negative, e.g. heating, for the very cold sub-arctic summer (SAS) mesopause, where heating can be of several $\mathrm{K \, day}^{-1}$ (see bottom/left panel of Fig. 6. Exceptional cases are the winter atmospheres (mid-latitude winter, MLW, and sub-arctic winter, SAW) where the mesopause is warmer and the cooling rates are large in this region. Above the mesopause, the cooling rate rapidly increases following the enhancement of the kinetic temperature. Above

about 130 km, the cooling rates decline because of the depletion of the $CO_2$ vmr (see Fig. 7). Note the significant contribution of the hot bands in the lower thermosphere (120–150 km), essentially due to the first hot band of the major isotopologue at these altitudes, which is about 10% of the total cooling (see Fig. 7).

The dependence of the non-LTE cooling rates on the $CO_2$ abundances is illustrated in Fig. 5. We observe that in general, the cooling rate correlates very well with the $CO_2$ abundance, although that correlation is not always linearly and generally

depends on altitude. This is true also for the cases where we have net heating for low $CO_2$ vmr, e.g. the subarctic summer (SAS) atmosphere between about 75 km and 95 km. For the MLE and MLS atmospheres, the cooling rate near ∼90 km changes from net cooling to net heating for the largest $CO_2$ vmrs. Further, the very small cooling for the TRO $p$-$T$ profile near 70 km remains





**Figure 5.** The non-LTE cooling rates for the reference atmospheres shown for altitudes up to the lower thermosphere. The cooling rates extended to the thermosphere are shown in Fig. C1. Note the different x-scales.



**Figure 6.** Contributions of the different $CO_2$ bands to the cooling rates for the six $p-T$ references atmospheres and the $CO_2$ vmr #3 profile. Note the different x-scales.





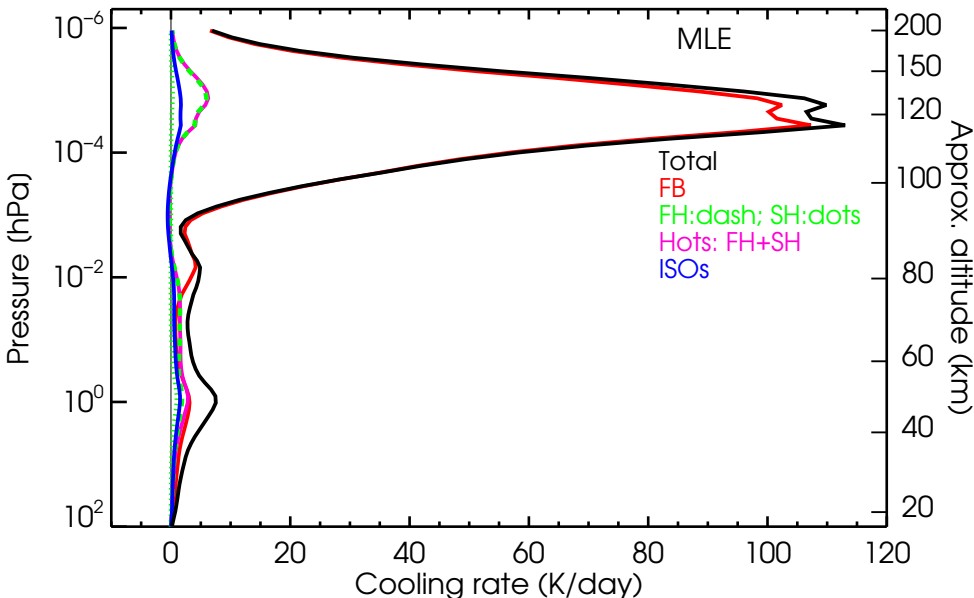

**Figure 7.** Contributions of the different $CO_2$ bands to the cooling rates for the MLE $p$-$T$ atmosphere. That is, as the top/left panel of Fig. 6 but for the entire altitude range.

very small even when the $CO_2$ vmr varies in a factor of 20. In the lower thermosphere, e.g. above around $\sim$110 km, however, the dependency of the cooling rate on the $CO_2$ is very close to being linear (see Fig. C1).

A comparison of the non-LTE and LTE cooling rates for the six $p$-$T$ reference atmospheres for $CO_2$ vmrs #3 (current vmr) and #6 (four times the preindustrial value) are shown in Figs. 8 and D1. This comparison is useful from a physical point of view and it is required to establish the boundaries of the different atmospheric regions of the parameterization (see Sec. 5). We first observe (Fig. 8) that the altitude of the departure of the cooling rate from LTE to non-LTE (considered as the altitude where the non-LTE–LTE difference is larger than 5%) depends on the temperature profiles and ranges from pressures of $5.2 \cdot 10^{-2}$ hPa

($\sim$72.5 km) for the sub-arctic summer (SAS) atmosphere to $1.2 \cdot 10^{-2}$ hPa ($\sim$78.7 km) for the tropical (TRO) atmosphere. A similar figure but for the higher $CO_2$ profile #6 is shown in Fig. D1. An overview of the altitude/pressure level of the deviation from LTE of the cooling rate is shown in Fig. 9 for the six $p$-$T$ profiles and the eight $CO_2$ vmr profiles. We see that the lower altitude (higher pressures) occur for the sub-arctic summer (SAS) and sub-arctic winter (SAW) reference atmospheres. It is also evident that this altitude increases with the $CO_2$ vmr, except for the SAS and SAW cases for which it is nearly independent of

the $CO_2$ vmr. That is expected as, for a more abundant $CO_2$ atmosphere, the 15 $\mu$m bands become optically thicker and fewer collisions are sufficient for keeping the emitting levels in LTE. Fig. 9 suggests that for higher $CO_2$ vmrs, the LTE to non-LTE transition region (see Fig. 11) could be placed at higher altitudes. However, as the parameterization is intended to cover the full range of $CO_2$ vmr profiles, we have to be conservative as placed it at the lowest altitude for any $p$-$T$ or $CO_2$ vmr profile. This has been taken at $x = 9.875$ (p $= 5.14 \times 10^{-2}$ hPa, $z \approx 70$ km) which, except for the SAW atmosphere and the lowest $CO_2$

profile, is fulfilled for all $p$-$T$ and $CO_2$ vmr profiles.







**Figure 8.** Non-LTE minus LTE cooling rates differences for the six $p-T$ references atmospheres and the $CO_2$ vmr #3 profile. Differences for the higher $CO_2$ vmr #6 profile are shown in Fig. D1. The '*' symbol indicates the pressure level (in hPa) where the non-LTE–LTE difference reaches 5%. Note the different x-scales.



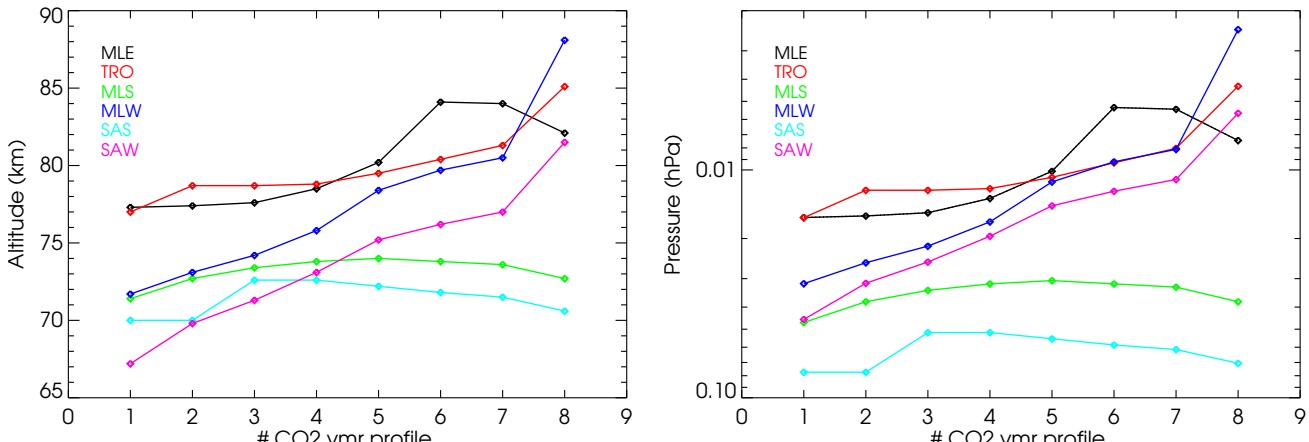

**Figure 9.** Altitude (left panel) and pressure level (right panel) of the deviation ($\geq 5\%$) of the non-LTE cooling rates from LTE values for the six $p$–$T$ references atmospheres and the eight $CO_2$ vmr profiles.

For completeness, Fig. D2 shows an example of the comparison of non-LTE and LTE cooling rates including the thermosphere for the six $p$-$T$ profiles and the current $CO_2$ values. This shows the enormous difference between LTE and non-LTE cooling rates (being non-LTE values much smaller) in the thermosphere.

The atomic oxygen concentration is an input to the parameterization and plays a crucial pointing the $CO_2$ infrared cooling. 220 The region where it is important also plays a crucial role when establishing the different non-LTE regions of the parameterization (Fomichev et al., 1998). To identify the atmospheric regions where it is important, we have perturbed the $k_{CO_2-O}$ collisional rate by a factor of two. We recall that perturbing this rate is equivalent to perturbing the $O(^3P)$ concentration. Fig. 10 shows this effect for the six $p$-$T$ profiles considered for the current $CO_2$. Generally, it is most important above around $10^{-3}$ hPa ($\sim$95 km). However, for the sub-arctic summer (SAS) and sub-arctic winter (SAW) atmospheres it is also important down to 225 $5\cdot10^{-3}$ hPa ($\sim$85 km). The fact that its importance starts being significant at different atmospheric levels for the different $p$-$T$ profiles poses an additional difficulty in the development of the parameterization.

## 5   The parameterization

Essentially, we follow here the parameterization developed by Fomichev et al. (1998). A brief description of the method including the most important features and equations is given in this section.

The atmosphere is divided into five different regions (see Fig. 11) where different approaches are used for calculating the cooling rates. These regions are qualitatively the same regions defined by Fomichev et al. (1998) but their altitude extensions (except for the LTE region) have been significantly revised, mainly as a consequence of the ample range of $CO_2$ abundances for which this parameterization is developed. In fact, their upper boundaries have been moved upwards (except for LTE), resulting in the following ranges: LTE from the lower boundary at $x = 0$ up to $x_{b,1} = 9.875$ ($z \approx 70$ km); NLTE1: from $x_{b,1}$ to $x_{b,2} = 12.625$





**Figure 10.** Effect of the $k_{CO_2-O}$ collisional rate (or, equivalently, the O($^3P$) concentration) on the non-LTE cooling rates for the six $p-T$ references atmospheres and the $CO_2$ vmr #3 profile. Note the different x-scales.





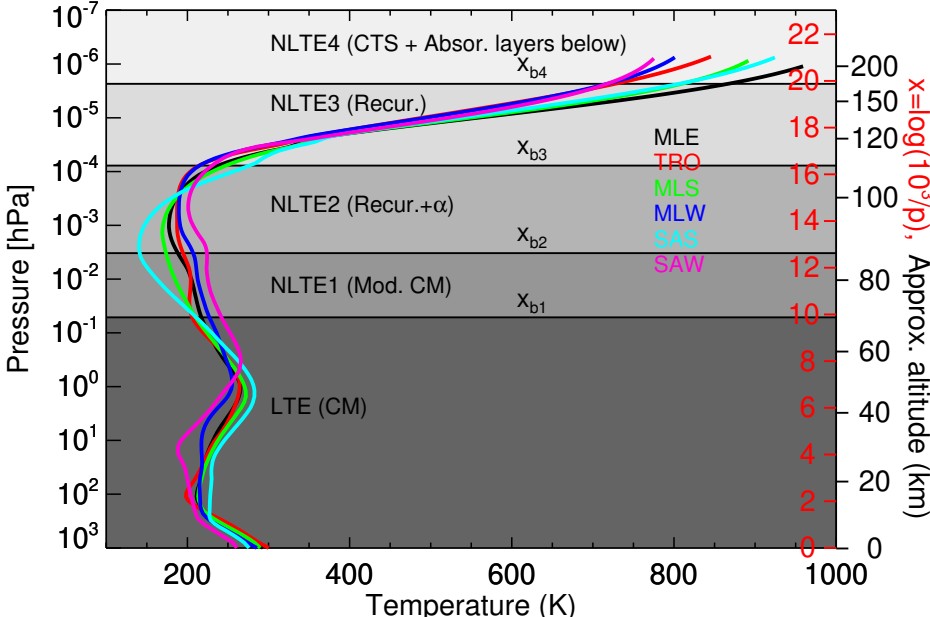

**Figure 11.** Atmospheric regions considered in the parameterization. $x_{b,i}$ represents the boundaries of the layers; $x_{b,1} = 9.875$, $x_{b,2} = 12.625$; $x_{b,3} = 16.375$, and $x_{b,4} = 19.875$. The temperature profiles used in the reference calculations are also shown as a reference.

($z \approx 70$–$87$ km); NLTE2: from $x_{b,2}$ to $x_{b,3} = 16.375$ ($z \approx 87$–$109$ km); NLTE3: from $x_{b,3}$ to $x_{b,4} = 19.875$ ($z \approx 109$–$180$ km); and NLTE4: for $x > 19.875$ ($z \gtrsim 180$ km).

The lowermost (LTE) and the uppermost (NLTE4) regions are the most straightforward and also the regions where the errors are in general smaller. The most difficult parts are the transition regions from LTE to non-LTE, where: (i) several bands contribute to the cooling with different source functions and their relative contributions depend very much on the actual temperature profiles (see Fig. 6); (ii) the exchange of radiation between layers is significant and different for the different bands. Further, although most of the radiative excitation at a given layer is produced by the absorption of photons travelling from below, the absorption of photons travelling downwards can also contribute significantly. This is the case, for example, for the stronger fundamental band near the mesopause. (iii) The cooling above around 90 km depends also on the collisions with atomic oxygen. This effect can be accurately taken into account in the upper non-LTE regions where all bands become optically thin, however it is very difficult to represent it properly between around the mesopause and a few tens km above, where the atomic oxygen concentration varies largely and the exchange of radiation between layers is still important.

## 5.1 The LTE region

The parameterization in the LTE region is based on the Curtis matrix method. The cooling rate $\epsilon_i^t(\nu)$ at a given pressure level $x_i$, in a spectral region $\nu$, and for a particular temperature profile $t$, is given by

$$\epsilon_i^t(\nu) = \sum_{j=0}^{j_{CM}} \mathcal{A}_{i,j}^t(\nu)\, \varphi_j^t(\nu) \tag{1}$$



where the indices $i, j$ refer to pressure levels $x_i$ and $x_j$, and the sum is extended over the pressure levels $x_j$ ranging from the lower boundary, $x_j = 0$, until $x_{j_{\mathrm{CM}}} = 13.875$. This upper boundary of the Curtis matrix, $x_{j_{\mathrm{CM}}} = 13.875$, has been selected to minimize the error in the lowest non-LTE region, NLTE1 (see more details in Sec. 5.2 below). $\mathcal{A}_{i,j}^t(\nu)$ is the modified Curtis matrix (slightly different from its usual definition, see, e.g. López-Puertas and Taylor, 2001). The last term in Eq. 1 represents the exponential part of the Planck function and is given by

$$\varphi_j^t(\nu) = \exp(-h\nu/(k_B T_j^t)) \tag{2}$$

where $h$ is the Planck constant, $k_B$ is the Boltzmann constant, and $T_j^t$ is the temperature of the $p$-$T$ profile $t$ at level $x_j$. The cooling rate is calculated in the spectral range from 540 to 800 $\mathrm{cm}^{-1}$ divided into frequency intervals, $\nu$, 10 $\mathrm{cm}^{-1}$ wide. Those cooling rate profiles have been calculated for each of the six $p$-$T$ reference atmospheres and the eight $CO_2$ vmrs profiles.

In the parameterization, the Curtis matrix is expressed with an explicit temperature dependence by

$$\mathcal{A}_{i,j}^t(\nu) = \mathbf{a}_{i,j}^t(\nu) + \mathbf{b}_{i,j}^t(\nu)\,\varphi_i^t(\nu),$$

where the matrix coefficients $\mathbf{a}_{i,j}^t(\nu)$ and $\mathbf{b}_{i,j}^t(\nu)$ are given by

$$\mathbf{a}_{i,j}^t(\nu) = \mathcal{A}_{i,j}^t(\nu)\,\frac{S_0(\nu)}{S_0(\nu) + [S_1(\nu) + S_2(\nu)]\,\varphi_i^t(\nu)},$$

$$\mathbf{b}_{i,j}^t(\nu) = \mathcal{A}_{i,j}^t(\nu)\,\frac{S_1(\nu) + S_2(\nu)}{S_0(\nu) + [S_1(\nu) + S_2(\nu)]\,\varphi_i^t(\nu)},$$

and $S_0(\nu)$, $S_1(\nu)$, and $S_2(\nu)$ are the band strengths of the fundamental, first hot and second hot bands, respectively, in the $\nu$-interval. In this way, the temperature dependence, mainly caused by the band strength of the first and second hot bands, is carried out in $\varphi_i^t(\nu)$. Those matrix coefficients are calculated for each spectral interval $\nu$. We obtain the coefficients for the entire spectral region of the $CO_2$ 15 $\mu$m bands by summing over all the $\nu$-intervals and weighting with the $\nu$-dependency of the $\varphi_i^t(\nu)/\varphi_i^t(\nu_0)$ factor,

$$\mathbf{a}_{i,j}^t = \frac{\sum_\nu \mathbf{a}_{i,j}^t(\nu)\,\varphi_j^t(\nu)}{\varphi_j^t(\nu_0)}$$

$$\mathbf{b}_{i,j}^t = \frac{\sum_\nu \mathbf{b}_{i,j}^t(\nu)\,\varphi_j^t(\nu)\,\varphi_i^t(\nu)}{\varphi_j^t(\nu_0)\,\varphi_i^t(\nu_0)}$$

with $\nu_0 = 667.3799\ \mathrm{cm}^{-1}$ being the frequency of the fundamental band of the major isotopologue.

Next, we define global $\mathbf{a}_{i,j}$ and $\mathbf{b}_{i,j}$ matrix coefficients, to be used for any input temperature profile, as weighted averages of the $\mathbf{a}_{i,j}^t$ and $\mathbf{b}_{i,j}^t$ for the six reference $p$-$T$ profiles. We introduce a set of normalized weights $\xi_i^t$, altitude-dependent, for each temperature profile so that:

$$\mathbf{a}_{i,j} = \sum_t \xi_i^t \mathbf{a}_{i,j}^t \quad \text{and} \quad \mathbf{b}_{i,j} = \sum_t \xi_i^t \mathbf{b}_{i,j}^t. \tag{3}$$

In this way, the cooling rate at a pressure level $x_i$, $\epsilon_i$, for a given input temperature profile $T_{inp}$, is calculated in the parameterization by:

$$\epsilon_i = \sum_j \left[\mathbf{a}_{i,j} + \mathbf{b}_{i,j}\,\varphi_i^{T_{inp}}(\nu_0)\right]\varphi_j^{T_{inp}}(\nu_0). \tag{4}$$





The weights $\xi_i^t$ are obtained by minimising the cost function $\chi(x_i)$ at each pressure level $x_i$, given by the sum of the square of the differences of the reference line-by-line LTE cooling rates, $\epsilon_{\mathrm{ref}}^t$, (see Sec. 4.2) and those computed by the parameterization, $\epsilon_{\mathrm{par},i}^t$, for each $p$-$T$ profile $t$ by using Eq. (4), e.g.,

$$\chi(x_i) = \sum_t \eta^t \left\{ \epsilon_{\mathrm{ref},i}^t - \epsilon_{\mathrm{par},i}^t \right\}^2,$$

or, in more detail, by

$$\chi(x_i) = \sum_t \eta^t \left\{ \epsilon_{\mathrm{ref},i}^t - \sum_j \sum_{t'} \xi_i^{t'} \left[ \mathbf{a}_{i,j}^{t'} + \mathbf{b}_{i,j}^{t'} \varphi_i^t(\nu_0) \right] \varphi_j^t(\nu_0) \right\}^2.$$

The normalized coefficients $\eta^t$ were introduced originally for considering different fractions of the area of the Earth ascribed to each $p$-$T$ reference profile. Thus, in the previous parameterization they were taken equal to 0.05 for the subarctic (winter and summer) profiles, 0.1 for the mid-latitude (winter and summer) profiles, 0.4 for the tropical profile, and 0.3 for the mid-

latitude equinox $p$-$T$ profile. In this study, we have explored different options including the original coefficients and a uniform weighting for the six $p$-$T$ profiles and we found a smaller $\chi$ for the latter, e.g., $\eta = 1/6$ for all profiles. Hence, that was included in this version.

In that way, we have parameterized the cooling rates as a function of temperature. The cooling rates depend also on the $CO_2$ vmr profiles (see Fig. 5). The parameterization incorporates the dependence on the $CO_2$ abundance by calculating $\mathbf{a}_{i,j}$ and $\mathbf{b}_{i,j}$

for a generic $CO_2$ profile by assuming a linear interpolation in $\log(\mathbf{a}_{i,j}/\mathrm{VMR}(x_i))$ and $\log(\mathbf{b}_{i,j}/\mathrm{VMR}(x_i))$ from the adjacent $CO_2$ vmr profiles. Thus, the $\mathbf{a}_{i,j}$ and $\mathbf{b}_{i,j}$ coefficients of Eq. 4 have been calculated (and are provided) for the eight $CO_2$ vrms shown in Fig. 3.

### 5.2   The NLTE1 region: The transition from LTE to non-LTE

This region is difficult to parameterize because we have several bands contributing to the cooling (see Fig. 6) and their relative

contributions depend significantly on both the temperature structure and the $CO_2$ vmr profile. Note that, at certain levels, the cooling induced by the weaker hot bands is larger than that of the stronger fundamental band. We should also note that the contribution of the first hot bands at high altitudes, $\sim$110–150 km is not negligible (5–10%, Fig. 7). This contribution is accounted for in the parameterization by implicitly assuming that it is produced by the fundamental band of the main isotopologue (see below and Sec 4.2).

The lower boundary of this region, e.g. the LTE-to-NLTE1 transition, occurs at altitudes from $\sim$70 km up to $\sim$85 km (0.08 hPa to 0.004 hPa, see Fig. 9), taking place a few kilometers lower for the subarctic summer atmosphere and the lowest $CO_2$ VMR. This transition region occurs at higher altitudes for larger $CO_2$ vmrs, i.e., the atmosphere becomes optically thicker and fewer collisions are enough to keep the levels in LTE. However, since we need to represent also the low $CO_2$ vmrs, we decided to conservatively set up this region at rather low altitudes, $x_{b,1} = 9.875$ ($\sim$70 km), the same used in the previous

parameterization.

The upper limit of this region was set up in the previous parameterization at the pressure levels where collisions with $O(^3P)$ start affecting significantly the cooling rates. Again, that pressure level depends on the temperature profile (and also on the





$O(^3P)$ concentration), being lower, at $\sim 0.004\,$hPa ($x \approx 12.4$), for the subarctic summer and winter conditions (see Fig. 10). Here, we have taken the upper boundary of $x_{b,2} = 12.625$ ($\approx 87\,$km), slightly higher than the 12.5 value assumed in the original parameterization.

In this region we followed, as in Fomichev et al. (1998), the matrix approach discussed in Sec. 5.1 above. Thus the Eq. 4 was used but with corrected $\mathbf{a}'^{t}_{i,j}$ and $\mathbf{b}'^{t}_{i,j}$ coefficients that account for the non-LTE corrections. For each $p$-$T$ profile, $t$, we define:

$$\mathbf{a}'^{t}_{i,j} = \mathbf{a}^{t}_{i,j} \left[ \epsilon^{t}_{(\mathrm{ref,nlte}),i} / \epsilon^{t}_{(\mathrm{ref,lte}),i} \right] \quad \text{and}$$

$$\mathbf{b}'^{t}_{i,j} = \mathbf{b}^{t}_{i,j} \left[ \epsilon^{t}_{(\mathrm{ref,nlte}),i} / \epsilon^{t}_{(\mathrm{ref,lte}),i} \right],$$

where $\epsilon^{t}_{(\mathrm{ref,lte})}$ and $\epsilon^{t}_{(\mathrm{ref,nlte})}$ are the reference LTE and non-LTE cooling rates, respectively. Then, the general $\mathbf{a}'_{i,j}$ and $\mathbf{b}'_{i,j}$ coefficients were calculated by following the same procedure as for the LTE region. That is, by weighting the $p$-$T$-specific $\mathbf{a}'^{t}_{i,j}$ and $\mathbf{b}'^{t}_{i,j}$ coefficients with a set of altitude-dependent weights $\xi'^{t}_{i}$ and minimizing the total cost function $\chi(x_i)$ (see Sec. 5.1). In this way, we obtain:

$$\mathbf{a}'_{i,j} = \sum_{t} \xi'^{t}_{i} \, \mathbf{a}^{t}_{i,j} \left[ \epsilon^{t}_{(\mathrm{ref,nlte}),i} / \epsilon^{t}_{(\mathrm{ref,lte}),i} \right] \quad \text{and}$$

$$\mathbf{b}'_{i,j} = \sum_{t} \xi'^{t}_{i} \, \mathbf{b}^{t}_{i,j} \left[ \epsilon^{t}_{(\mathrm{ref,nlte}),i} / \epsilon^{t}_{(\mathrm{ref,lte}),i} \right],$$

and the cooling rates are computed by using Eq. 4 but replacing $\mathbf{a}_{i,j}$ and $\mathbf{b}_{i,j}$ by $\mathbf{a}'_{i,j}$ and $\mathbf{b}'_{i,j}$, respectively, i.e.,

$$\epsilon_i = \sum_{j} \left[ \mathbf{a}'_{i,j} + \mathbf{b}'_{i,j} \, \varphi^{T_{inp}}_i(\nu_0) \right] \varphi^{T_{inp}}_j(\nu_0). \tag{5}$$

This procedure, while producing a perfect match for a single atmosphere by construction, generates irregularities for other atmospheres, e.g., when using $\mathbf{a}'^{t}_{i,j}$ for atmosphere $t'$ at points where $\epsilon^{t}_{(\mathrm{ref,lte}),i}$ is close to zero. We observed that the irregularities were significantly mitigated by reducing the dimensions of the Curtis matrix from $83 \times 83$ to $55 \times 55$, where $i = 55$ corresponds to $x_{CM} = 13.875$ (p $= 9.422 \times 10^{-4}\,$hPa, z$\approx 94\,$km). That is, by placing $x_{CM}$ slightly above the $x_{b2}$ boundary. Errors induced in the LTE cooling rates by the matrix reduction are negligible (smaller than $0.05\,\mathrm{K\,day}^{-1}$ at the upper boundary).

## 5.3 The NLTE2 and NLTE3 regions: The recurrence formula with/without correction

The parameterization in the NLTE2, NLTE3 and NLTE4 regions is based on the recurrence formula proposed by Kutepov and Fomichev (1993). This approach is valid when the cooling rate is dominated by the fundamental band and also the absorption of radiation coming from the layers above the layer at work can be neglected (Kutepov and Fomichev, 1993; Fomichev et al., 1998). Those conditions are fulfilled to a large degree in the NLTE3 region. In the layers below, i.e., in the NLTE2 region, however, that formula is not accurate and requires a correction term that accounts for the absorption of radiation coming from the layers above and for the cooling of bands other than the fundamental of the main isotopologue. This formula is also the basis for the calculation of the cooling rate in the NLTE4 region (see Sec. 5.4) but is simplified because the exchange of photons of this region with the layers below can be neglected.



We should emphasise that the dependence of the cooling rate on the $CO_2$ vmr in these regions is mainly twofold. On one
hand, its direct dependence (see Eq. 6 below), and also through the escape function which depends on the $CO_2$ column above
a given layer (see Fig. 12).

We discuss below the boundaries of the NLTE2 and NLTE3 layers and the expressions that are used for the cooling rates in
these regions. First, we describe the recurrence formulation, then the correction that is applied in the NLTE2 region.

The NLTE2 region is limited from below by the layers where the cooling rate obtained by the corrected recurrence formula is
better than that given by the non-LTE-corrected Curtis matrix approach. This layer has been chosen at $x_{b,2} = 12.625$ ($\approx 87$ km),
which is very similar to the value in the original parameterization of $x_{b,2} = 12.5$ ($\approx 85$ km). Its upper limit is set up at the layers
where the recurrence formula does not need to be corrected to yield an accurate estimation of the cooling rate. In this work, it
has been set up at $x_{b,3} = 16.375$ ($\approx 109$ km), which is significantly higher than the value of $x_{b,3} = 14$ ($\approx 93$ km) in the previous
parameterization. The main reason is that for the higher $CO_2$ vmr's used here, the atmosphere becomes optically thicker; hence,
the absorption of radiation of the layer above needs to be taken into account also at lower pressures.

The cooling rates in the NLTE2 and NLTE3 regions are calculated by:

$$\epsilon(x_i) = \kappa_F \, \frac{\mathrm{VMR}(x_i)\,[1-\lambda(x_i)]}{M(x_i)} \, \tilde{\epsilon}(x_i), \tag{6}$$

where $\kappa_F = 2.55520997 \times 10^{11}$ is a constant that depends on the Einstein coefficient of the fundamental band ($A$), on $\nu_0$ and on
the units of $\epsilon$ (Fomichev et al., 1998)[1], $\mathrm{VMR}(x)$ is the $CO_2$ vmr, $M(x)$ is the mean molecular weight, $\lambda(x_i) = A\,/\,[A+l_t(x_i)]$,
$l_t(x_i) = k_{N2}\,[N_2] + k_{O2}\,[O_2] + k_O\,[O]$, $k_{N2}$, $k_{O2}$ and $k_O$ are the collisional rate constants with $N_2$, $O_2$ and $O(^3P)$ (see Table
1), and $[N_2]$, $[O_2]$ and $[O(^3P)]$ are the concentrations of the respective species. Note that the collisional rates depend on $x_i$
through their temperature dependencies.

$\tilde{\epsilon}$ at level $x_i$, $\tilde{\epsilon}(x_i)$, is obtained by the recurrence formula

$$[1 - \lambda(x_i)\,(1-D_i)]\,\tilde{\epsilon}(x_i) = [1 - \lambda(x_{i-1})\,(1-D_{i-1})]\,\tilde{\epsilon}(x_{i-1}) + D_{i-1}\,\varphi_{i-1} - D_i\,\varphi_i \tag{7}$$

starting from the lower boundary at $x_i = x_{b2}$, where, using Eq. (6),

$$\tilde{\epsilon}(x_{b2}) = \frac{M(x_{b2})}{\kappa_F \, \mathrm{VMR}(x_{b2})\,[1-\lambda(x_{b2})]} \, \epsilon(x_{b2}) \tag{8}$$

and $\epsilon(x_{b2})$ is obtained by Eq. (5). The $D_i$ coefficients above are given by

$$D_i = (d_{i-1} + 3\,d_i)/4 \quad \text{and} \quad D_{i-1} = (3\,d_{i-1} + d_i)/4 \tag{9}$$

where

$$d_i = \begin{cases} \alpha(x_i, u)\,L(u) & \text{if } x_{b2} \leq x_i \leq x_{b3}. \\ L(u) & \text{if } x_i \geq x_{b3}. \end{cases} \tag{10}$$

---

[1]Note that this constant has been changed from its value of $2.63187 \times 10^{11}$ in Fomichev et al. (1998) to the actual value used in the distributed code and
quoted here of $2.55520997 \times 10^{11}$.





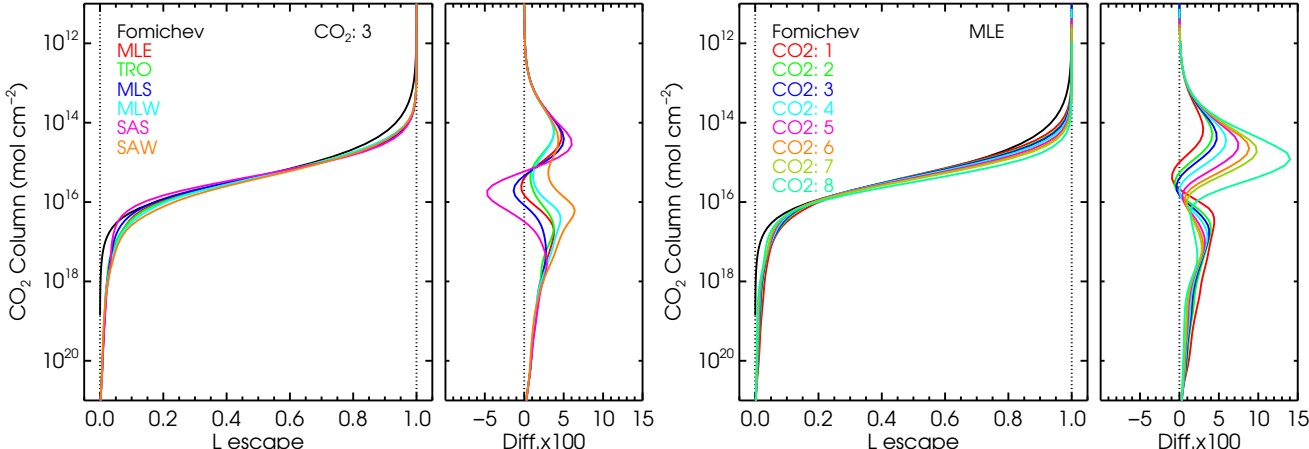

**Figure 12.** The escape probability function $L(u)$ represented as function of the $CO_2$ column at a given layer, $u(x)$, for several $p$-$T$ and $CO_2$ vmr profiles. They have been calculated with the GRANADA code. The right panels show the differences with respect to the values used in the previous parameterization (a single profile for all atmospheres). The $CO_2$ columns as a function of altitude are shown in Fig. E1.

$L(u)$ is the escape function which mainly depends on the $CO_2$ column, $u$, above a given level $x_i$. However, the temperature of the layers above affects this function as it influences the line shape of the $CO_2$ lines and hence the probability of photons escaping to space. Fig. 12 shows the calculations of $L(u)$ as a function of the $CO_2$ column for the six $p$-$T$ reference atmospheres and a single $CO_2$ vmr profile (#3); and for the MLE atmosphere and the eight $CO_2$ vmr profiles. In our calculations, we have
used for $L(u)$ the average of this function for the six $p$-$T$ reference atmospheres.

The $\alpha(x_i, u)$ parameter entering in Eq. 10 for the NLTE2 region has been computed by minimizing the following cost function at each point $x_i$:

$$\chi(x_i) = \sum_t \eta^t \left[ \epsilon_{\mathrm{ref}}^t(x_i) - \epsilon_{\mathrm{par}}^t(\alpha, x_i) \right]^2 .$$

After performing some sensitivity tests, we used uniform weighting for the different reference atmospheres ($\eta^t = 1/6$ for all
atmospheres), rather than the area-weighting used in the previous parameterization. Other tests were performed to determine the optimal upper boundary for the $\alpha$ correction: extending the region upwards reduces the error in the x = 16-19 region, but results in a spurious jump at the uppermost boundary, which is avoided with a lower $x_{b3} = 16.375$. It is worth noting that $\alpha$ above x = 14.5 goes below unity, thus decreasing the escape in the region. For the fit of the optimal $\alpha$, the parameterized value of $\epsilon(x_{b2})$ is considered as starting point rather than the reference value.
In the NLTE3 region, we used the same method as in region NLTE2, except that no correction for the $L(u)$ function is applied, i.e., $\alpha(x_i, u) = 1$.

### 5.4 The NLTE4 region

The recurrence formula used in the NLTE2 and NLTE3 regions is also valid in the uppermost region, NLTE4, but, in this case, as the $CO_2$ bands are so optically thin, the exchange of radiation within the layers of this region can be neglected and the





recurrence formula is reduced to a simpler expression. Namely, a cooling-to-space term and an additional term that accounts
for the absorption of the radiation emitted by the layers below the boundary of this NLTE4 region. Thus, in this region, the
cooling rate is calculated by using Eq. (6) but with a simple expression for $\tilde{\epsilon}(x_i)$, $\tilde{\epsilon}(x_i) = \Phi(x_{b3}) - \varphi(x_i)$, that gives a smooth
transition to the cooling to space approximation, e.g.,

$$\epsilon(x_i) = \kappa_F \frac{\text{VMR}(x_i)\left[1 - \lambda(x_i)\right]}{M(x_i)} \left[\Phi(x_{b3}) - \varphi(x_i)\right], \tag{11}$$

where $\Phi(x_{b3})$ is obtained from the boundary condition

$$\Phi(x_{b3}) = \tilde{\epsilon}(x_{b3}) + \varphi(x_{b3}) \tag{12}$$

and using the recurrence formula in Eq. (7).

## 6    Testing the parameterization

The parameterization has been tested against the accurate line-by-line cooling rates calculation for the references atmospheres
(the six $p$-$T$ profiles and the eight $CO_2$ VMR profiles) (next section), and for intermediate $CO_2$ vmrs and for the $k_{CO_2-O}$
collisional rate (Sec. 6.2). Further, it has been verified for measured temperature profiles that exhibit a large variability (Sec. 7).

### 6.1    Accuracy of the parameterization for the reference atmospheres

In this section, we discuss the accuracy of the current parameterization for the assumed references atmospheres. The non-LTE
models used in both parameterizations are different. Hence we expect some differences not just caused by the parameteriza-
tion itself but possibly also by the non-LTE models. Figure 13 shows the cooling rates of this parameterization compared to
those of the previous parameterization and to those obtained by the accurate, line-by-line (reference) non-LTE model, for a
contemporary $CO_2$ vmr profile (#3 in Fig. 3) and the six $p$–$T$ profiles. The comparison for lower $CO_2$ profiles are shown in
Figs. F1, F2, and F3, and in Figs. F4–F7 for high $CO_2$ vmrs. We should clarify that to make the comparison meaningful the
three sets of cooling rates shown here include the same updated collisional rates (Table 1). Note, however, that these rates are
different from those used in the previous parameterization (Fomichev et al., 1998). The new parameterization also supports the
previous collisional rates but it has ben optimized for the new ones, see Table 1. As expected, larger differences are obtained in
the region between $10^{-2}$ hPa ($\sim$80 km) and $2 \cdot 10^{-5}$ hPa ($\sim$120 km) and are more marked for the sub-arctic summer (SAS) and
sub-arctic winter (SAW) atmospheres.

The differences are more clearly illustrated in Figs. 14 and 15, where we show the mean and the RMS (root mean square) of
the differences for the eight $CO_2$ VMR profiles. The improvement of the new parameterization is noticeable (compare blue and
red lines). In general, the cooling rates of the current parameterization are more accurate than in the previous one for most of the
regions and temperature structures. We observe that the 'errors' (e.g., the differences with respect to the line-by-line reference
non-LTE cooling rates) of the new parameterization (red curves) are very small overall. They are below $\sim$0.5 K day$^{-1}$ for
the current and lower $CO_2$ abundances (see Fig. 14). For higher $CO_2$ concentrations, between about two and three times the



pre-industrial values, the largest errors are $\sim$1–2 K day$^{-1}$, and are located near 100–120 km (see Fig. 14 and top/left panel in Fig. 15). The quoted values refer to the mean of the differences although they are larger for the individual $p$-$T$ atmospheres. The spread of these values is larger in the region of $10^{-2}$ hPa ($\sim$80 km) to $10^{-4}$ hPa ($\sim$105 km), where the RMS reach values between $-2$ K day$^{-1}$ to $+2$ K day$^{-1}$ (Fig. 14).

     For the very high $CO_2$ concentrations (four, five and ten times the pre-industrial abundances) the errors are also very small,
below $\sim$1 K day$^{-1}$ for most regions and conditions; except in the 107–140 km region where we found maximum positive bias of $\sim$4 K day$^{-1}$, $\sim$5 K day$^{-1}$, and $\sim$16 K day$^{-1}$ for the 4$\times$, 5$\times$ and 10$\times$ the pre-industrial $CO_2$ vmr profiles (see Fig. 15). Those maximum errors in the cooling rates for the different $CO_2$ vmrs are however comparable when expressed in relative terms, all about 1.2%. It is also notable the significantly large spread (large RMS) in the region of $\sim$80–120 km; clearly, the region which is more difficult to parameterize, particularly for such a large range of $CO_2$ abundances.

That increase of the differences of the new parameterization with respect to the reference calculations for the very high $CO_2$ vmrs near 110 km, seems to be related to the transition region from NLTE2 to NLTE3 (see Fig. 11). It looks like the cooling in the lower part of the NLTE3 region requires also the $\alpha$ correction for high $CO_2$ vmrs. This suggests that for higher $CO_2$ vmrs the parameterization would be more accurate if this transition altitude is risen. Such a rise, however, would worsen the cooling below this boundary. This manifests the difficulty of obtaining very accurate cooling rates for a large range of $CO_2$ vmr with
this method.







**Figure 13.** Comparison of the cooling rates of the current and previous parameterizations with respect to reference non-LTE cooling rates for the present-day $CO_2$ vmr profile #3 and the six $p$-$T$ reference atmospheres. Note that the latter are hardly visible in the left panels of the figures. See Figs. F1 and F2 for lower $CO_2$ concentrations, and Figs. F3–F7 for higher $CO_2$ vmrs.



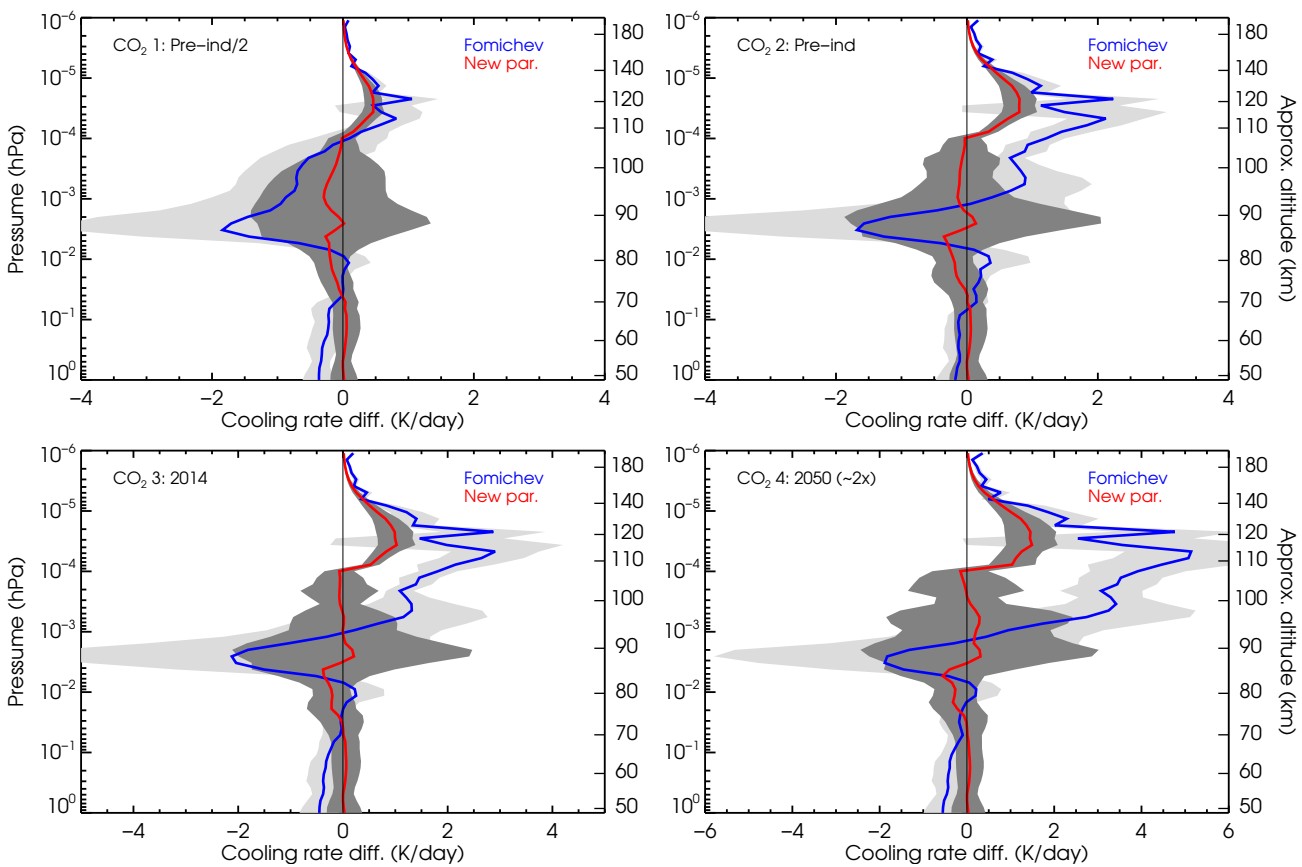

**Figure 14.** Global mean of the cooling rates differences of the current and previous parameterizations with respect to the reference line-by-line (LBL) cooling rates for the four lowest $CO_2$ vmr profiles. The shaded areas show the standard deviation of the differences between the six $p$-$T$ atmospheres. Detailed comparisons for each $p$-$T$ profile are shown in Fig. 13 above, and Figs. F1, F2, and F3 in F.





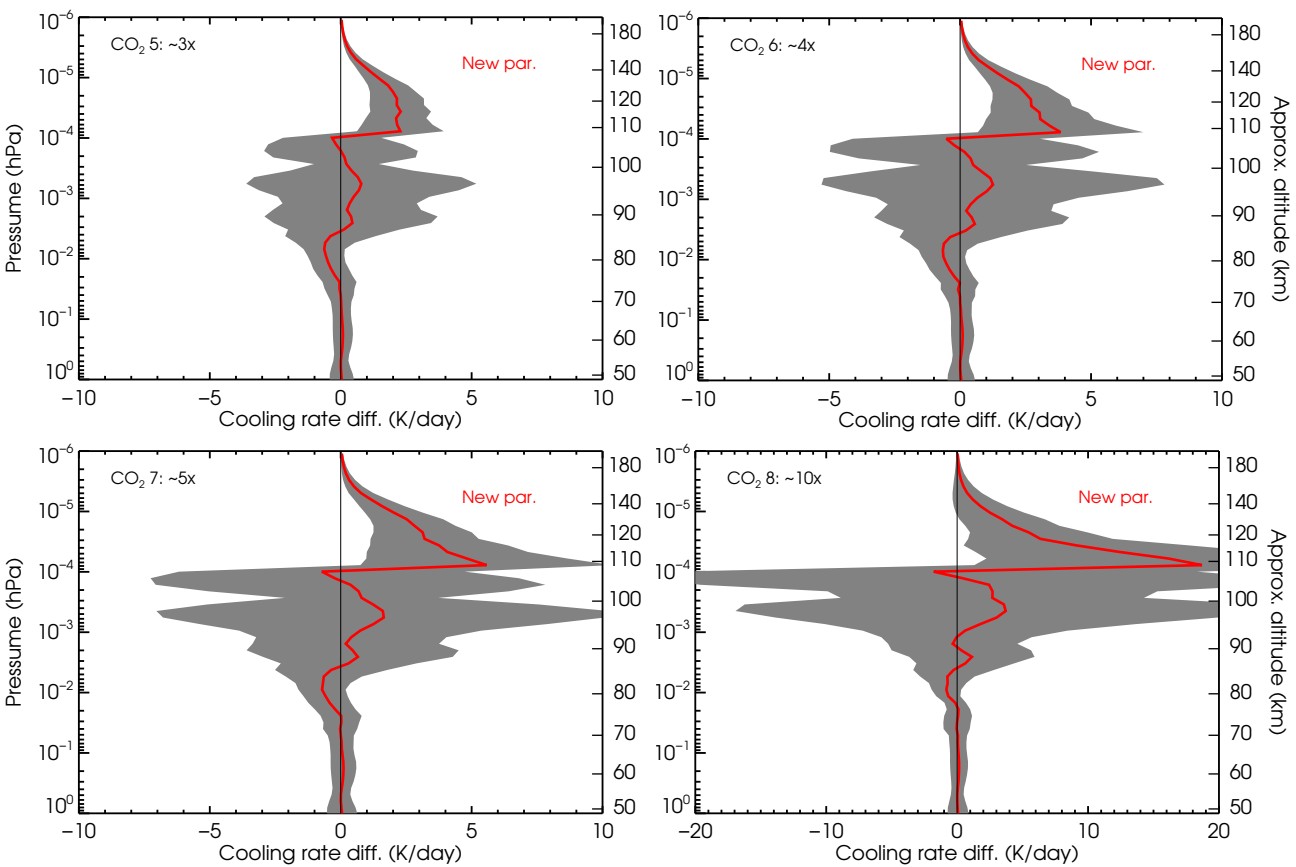

**Figure 15.** Global mean of the cooling rates differences of the current parameterization with respect to the reference non-LTE cooling rates for the four highest $CO_2$ vmr profiles (#5–8) considered in this work. The shaded areas show the differences (standard deviation) spread between the six p-T atmospheres. Detailed comparisons for each $p - T$ profile are shown in Figs. F4–F7.





## 6.2 Assessment of the cooling rates for intermediate $CO_2$ vmrs and for the $k_{CO_2-O}$ collisional rate

The aim of the parameterization is to be used for any $CO_2$ vmr input profile in the range of the profiles #1 and #8 of Fig. 3 and any plausible value for the $k_{CO_2-O}$ rates discussed in Sec. 4.2. In this section we demonstrate that the parameterization also works for $CO_2$ vmrs that fall between the reference profiles used for its development and also when using different $k_{CO_2-O}$

values. In particular we show results for the intermediate $CO_2$ vmr profiles #9, #10 and #11 (see Fig. 3) and the $k_{CO_2-O}$ collisional rate used in the reference calculations divided by two.

Figure 16 shows the results of the calculation for the intermediate $CO_2$ vmr profile #9, which is between the current $CO_2$ vmr value and that projected for 2050 (two times the pre-industrial value). We can observe similar features to the calculations for the adjacent reference $CO_2$ vmr profile #3 (see Fig. 13), although the differences are slightly larger because we are using

a larger $CO_2$ vmr profile. The distinctions are more clearly seen in Fig. 17 where we show the mean and the RMS of the differences for the six $p$-$T$ profiles. The patterns in the differences, as well as their values and spreads, are very similar to those described above in Sec. 6 for the $CO_2$ vmr reference profiles. The major differences appear between 105 and 140 km, reaching maximum values of 1, 2 and 10 K day$^{-1}$ for the vmr profiles #9, 10 and 11, respectively. Again, as we observe, the new parameterization is more accurate at practically all altitude levels. Further, the maximum values of the standard deviations

of the differences for the various $p$-$T$ profiles also resemble very much those discussed before, reaching maximum values of about 2 K day$^{-1}$, 3 K day$^{-1}$, and almost 10 K day$^{-1}$ for the respective $CO_2$ vmr profiles.

As the $CO_2(v_2)$-$O(^3P)$ collisional rate, $k_{CO_2-O}$, is still uncertain nowadays by almost a factor of two (García-Comas et al., 2012) and we intend that this parameterization be also used for rates different of the nominal value, we have tested its accuracy for its lowest likely value. Fig. 18 shows the results of decreasing the collisional rate $k_{CO_2-O}$ by a factor of two for the $CO_2$

vmr profile #3 (current $CO_2$) and the six $p$-$T$ profiles. The errors incurred when using this rate are slightly larger than for the nominal rate. We see that for the reduced rate, the differences are generally below 1 K day$^{-1}$, but can have values up to 2 K day$^{-1}$ near 90 km for the mid-latitudes summer and mid-latitudes winter atmospheres; and between ~80 km and 100 km for the sub-arctic summer conditions. The improvement with respect to the previous parameterization is not that large for this case (see Fig. 19); only below 70 km and near 90 km, mainly caused by the significant difference incurred by the previous

parameterization for the sub-arctic winter atmosphere (see bottom/right panel in Fig. 18). The smaller differences between both versions of the parameterizations for the reduced $k_{CO_2-O}$ are likely caused because the previous one was optimized for this lower rate.





**Figure 16.** Comparison of the cooling rates of the current and previous parameterizations with respect to accurate cooling rates for the intermediate $CO_2$ vmr profile #9 (see Fig. 3) and the six $p$-$T$ reference atmospheres.





**Figure 17.** Mean of the cooling rates differences of the current parameterization with respect to the reference cooling rates for the intermediate $CO_2$ vmr profiles (#9–11). The shaded areas show the differences (standard deviation) spread between the six $p$-$T$ atmospheres. Note the different scales of the $x$-axis. Detailed comparisons for each $p$-$T$ profile are shown in Figs. 16, F8 and F9.





**Figure 18.** Testing the effect of the $CO_2(v_2)$-$O(^3P)$ collisional rate on the parameterization. Comparison of the cooling rates of the current and previous parameterizations with respect to the reference cooling rates for the current $CO_2$ vmr abundance (profile #3, see Fig. 3) and the six $p$-$T$ reference atmospheres when reducing the collisional rate by a factor of two.





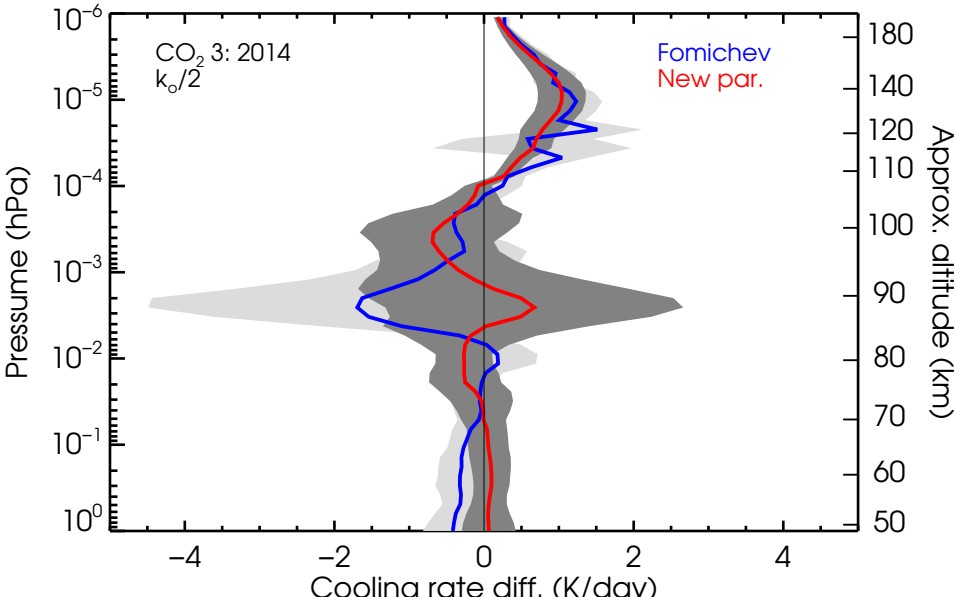

**Figure 19.** Global mean of the cooling rates differences of the current parameterization with respect to the accurate (reference) cooling rates for the results in Fig. 18. The shaded areas show the differences (standard deviation) spread between the six $p$-$T$ atmospheres.

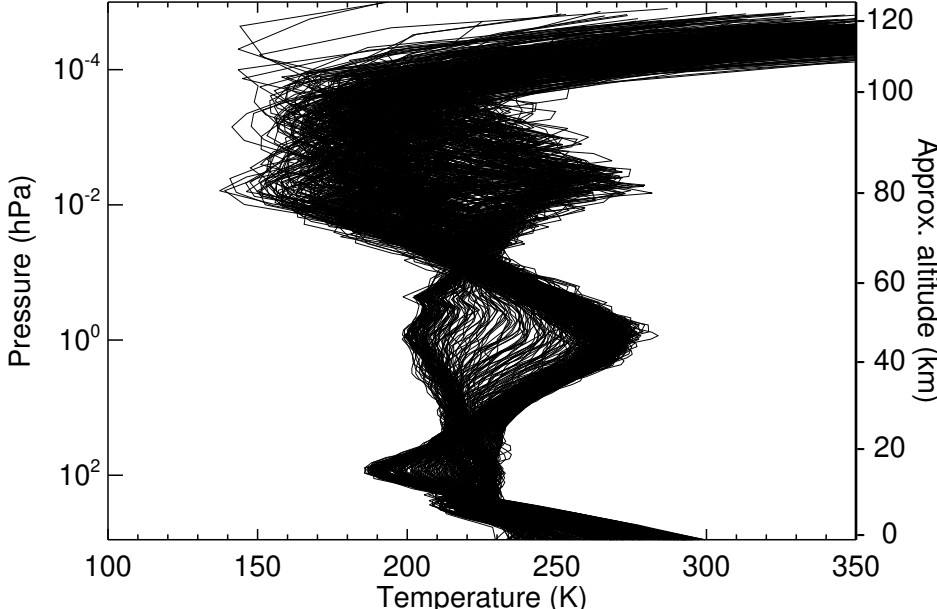

**Figure 20.** An example of the MIPAS nighttime temperature profiles (15 February 2009) used for verifying the parameterization accuracy. Note the large variability of the temperature profiles.




## 7 Testing the parameterization for measured temperatures

### 7.1 Solstice and equinox conditions

We have compared the cooling rates estimated by the parameterization with those calculated accurately by the line-by-line (LBL) model for realistic, e.g. measured, temperature profiles that present a large variability and very changing vertical structure (see e.g. Fig. 20). Specifically, we compared them for the $p$-$T$ profiles measured by the MIPAS instrument (García-Comas et al., 2023) for five full days of measurements (about 2500 profiles) with global latitude coverage and covering two days for solstice (14 January and 13 June) and two days for equinox conditions (25 March and 21 September) for 2010. Further, we

compare the results for the temperatures of 15 February 2009 when a strong stratospheric warming, followed by an elevated stratopause event, occurred in the Northern polar hemisphere. The comparison is carried out for the MIPAS measurements taken only during nighttime conditions, as the MIPAS non-LTE cooling rates for daytime, obtained simultaneously with the temperature inversion, include also the fraction of the $15\,\mu m$ cooling which is produced by the relaxation of the solar energy absorbed by $CO_2$ near-IR bands, which is not accounted for in this parameterization (see Sec. 8). The zonal mean of the

temperatures, $CO_2$ vmrs and $O(^3P)$ abundances for those conditions are shown in Figs. 21, G1 and G2, respectively.

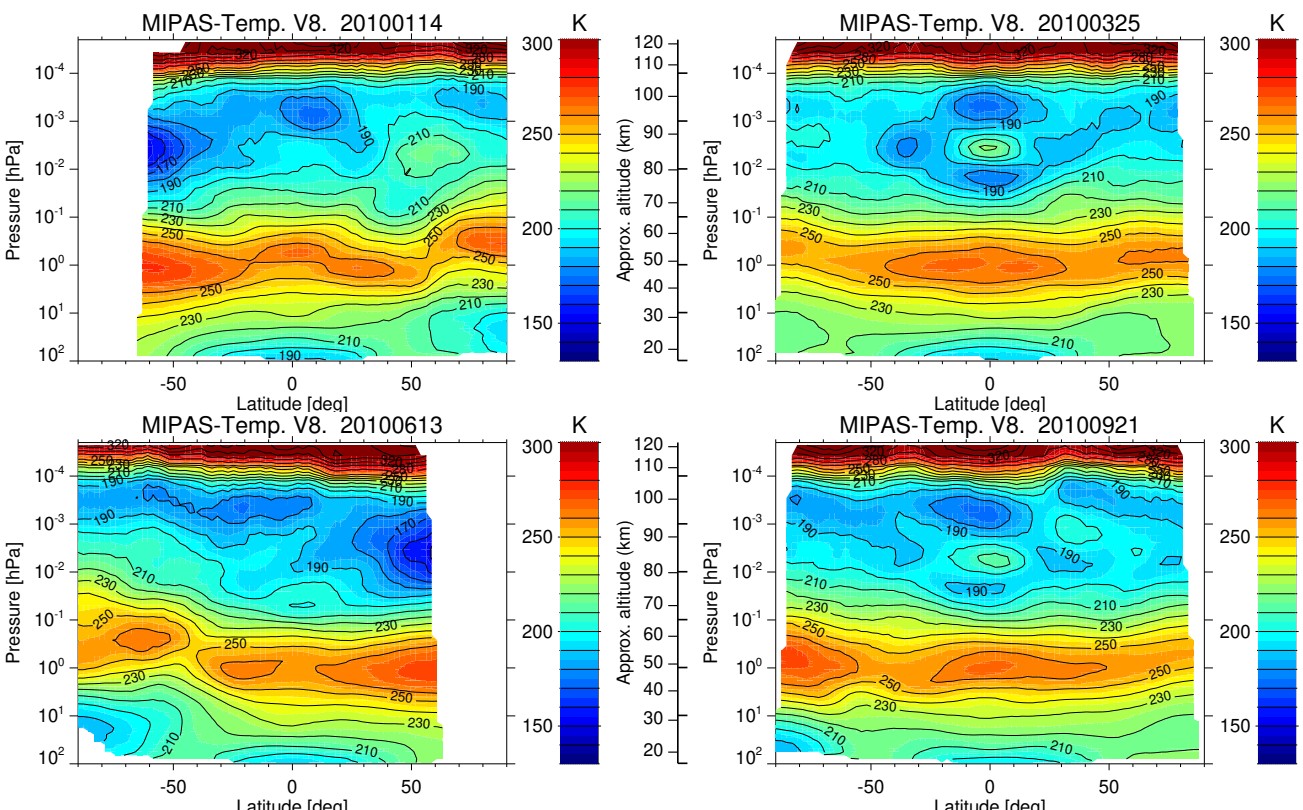

**Figure 21.** MIPAS zonal mean nighttime temperatures for 14 January 2010 (Northern winter hemisphere, top/left), 25 March 2010 (Northern spring hemisphere, top/right), 13 June 2010 (Northern summer hemisphere, bottom/left) and 21 September 2010 (Northern fall hemisphere, bottom/right).





The results are presented in Fig. 22 for the zonal mean of the differences for two days of solstice and two days of equinox conditions and in Fig. 23 as the global mean difference for all latitudes for each of the four individual days.

In general, the new parameterization is slightly more accurate. For example, the deviations of the cooling rates from the reference LBL calculations in the altitude range of 105–115 km are larger in the old parameterization (about $2\,\mathrm{K\,day}^{-1}$) than in the new one for which they are negligible. Also, the differences with respect to the reference calculations are larger in the altitude range of 80–95 km for solstice conditions, and at altitudes of 80–100 km for equinox conditions (see Fig. 23).

Overall, the errors in mean profiles of the cooling rates of the new parameterization for one day of measurements are below $0.5\,\mathrm{K\,day}^{-1}$, except in the region between $5{\cdot}10^{-3}\,\mathrm{hPa}$ and $3{\cdot}10^{-4}\,\mathrm{hPa}$ ($\sim$85–95 km), where they can reach values of $1$–$2\,\mathrm{K\,day}^{-1}$. This region is the most difficult to parameterize because several bands contribute to the cooling rate and they are very sensitive to the temperature structure of the middle atmosphere (e.g., even outside this region). Note also that this is precisely the region where the standard deviation of the differences of the cooling rate with respect to the reference ones are largest (see Fig. 24).



**Figure 22.** Zonal mean of the differences in the cooling rates of the old parameterization (left column) and the new one (right column) with respect to the LBL accurate cooling rates obtained for MIPAS temperatures for 14 January 2010 (solstice), 25 March 2010 (equinox), 13 June 2010 (Northern summer hemisphere) and 21 September 2010 (Northern fall hemisphere).



**Figure 23.** Mean of the differences in the cooling rates of the old parameterization (blue) and the new one (red) with respect to the LBL accurate cooling rates obtained for MIPAS temperatures for 14 January and 13 June 2010 (solstice, left panels), and 25 March and 21 September 2010 (equinox, right panels).





**Figure 24.** RMSs of the differences in the cooling rates of the old parameterization (in blue) and the new one (red) with respect to the LBL accurate cooling rates obtained for MIPAS temperatures for 14 January and 13 June 2010 (solstice, left panels), and 25 March and 21 September 2010 (equinox, right panels).




## 7.2 Elevated stratopause conditions

The comparison of the cooling rates estimated by the old and new parameterizations with respect to the reference LBL calcu-
lations for 15 February 2009, a day with a pronounced elevated stratopause event (see the zonal mean temperatures in Fig. 25)
are shown in Fig. 26. Similar features as for the other conditions shown above can be appreciated, except in the polar winter
region. The mean of the differences and the standard deviations for all the profiles at latitudes northernmost of 50°N are shown
in Fig. 27. The differences are significantly larger than for other latitudes in the 80–95 km altitude region. Both parameteriza-
tions underestimate the cooling in that atmospheric region. The new parameterization has, however, a better performance above
about 80 km, but in the strat-warm/elevated stratopause region (80–100 km) it still underestimates the cooling by 3–7 K day$^{-1}$
(∼10%).

It seems clear that part of this underestimation is caused by the fact that such atypical temperature profiles (see Sec. 3.1)
were not considered in the parameterization. However, its inclusion would not solve the problem as in the calculations of the
coefficients a trade-off of the weighting of the different $p$-$T$ reference atmospheres has to be chosen (see Secs. 5.1 and 5.2).
Thus, it might ameliorate the inaccuracy for these elevated-stratopause events but would worsen the accuracy for other general
situations. This manifests the difficulty/limitation of this method to provide accurate non-LTE cooling rates for all temperature
structures (gradients) that we might find in the real atmosphere.

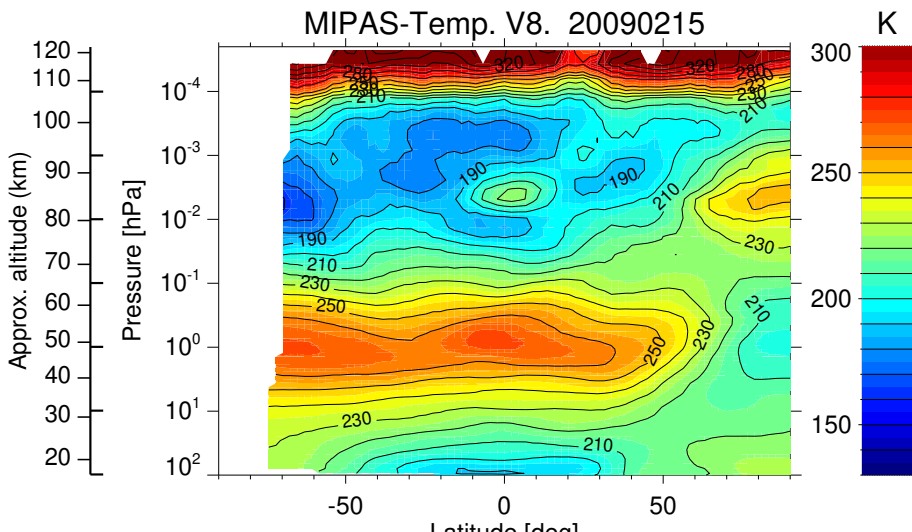

**Figure 25.** The MIPAS nighttime temperature zonal mean for 15 February 2009, used for verifying the parameterization accuracy for
stratospheric sudden warming and elevated stratopause conditions. Note the location of the stratopause in the Northern polar region.





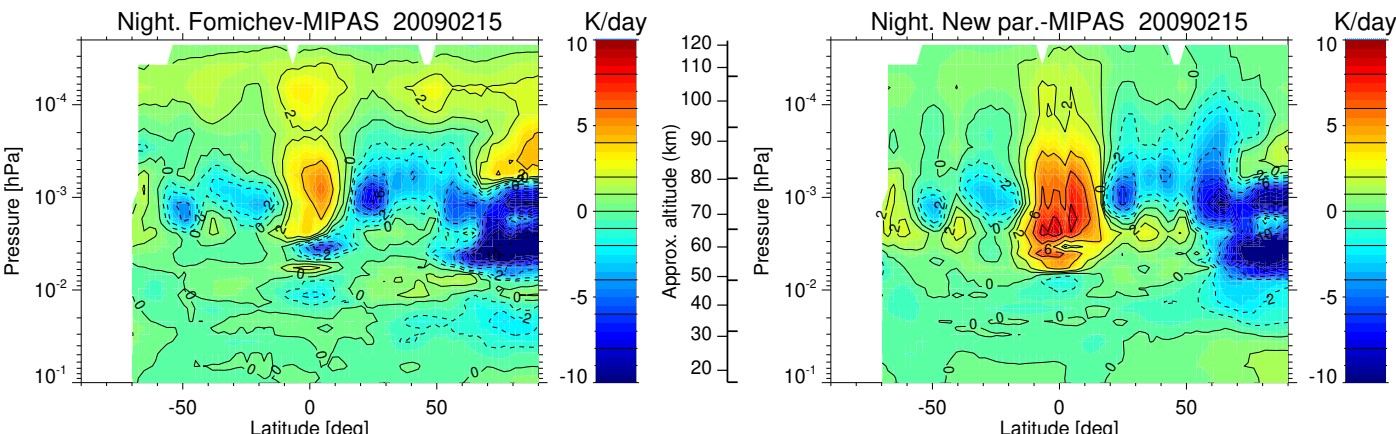

**Figure 26.** As Fig. 22 but for the MIPAS temperatures taken on 15 February 2009 when a major elevated stratopause event occurred.

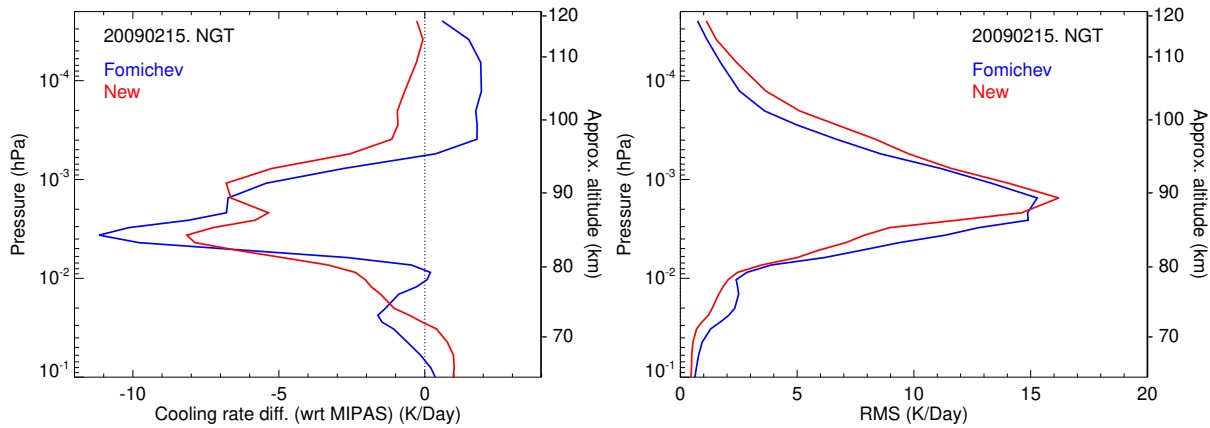

**Figure 27.** Mean (left) and RMS (right) of the differences in the cooling rates of the old parameterization (in blue) and the new one (red) with respect to the LBL accurate cooling rates obtained for MIPAS temperatures for 15 February 2009 considering only latitudes northernmost of $50°$N.

## 8    Discussion: The use of this parameterization with a previous $CO_2$ solar NIR heating rates parameterization

Some of the GCM models use the parameterization of the $CO_2$ 15 $\mu$m cooling together with that of the $CO_2$ near-infrared (NIR) heating of Ogibalov and Fomichev (2003). Hence, as we are updating the former for larger $CO_2$ abundances, and the update on the NIR heating parameterization for the large $CO_2$ abundances is out of the scope of this work, we investigate if the latter is still valid for the large $CO_2$ abundances. For that purpose, we compute $CO_2$ NIR heating rates with the parameterization of Ogibalov and Fomichev (2003) and with the GRANADA model for the large $CO_2$ concentrations for the six $p$-$T$ reference atmospheres. We should note that the non-LTE models used in both parameterizations are different and hence we expect some difference not just caused by the parameterization itself but by the differences between the underlying non-LTE model and GRANADA. The $CO_2$ NIR heating rates of GRANADA were calculated with the rate coefficients and photolysis rates described in Funke et al. (2012) but updated with those described in Jurado-Navarro et al. (2015, 2016) and also with those



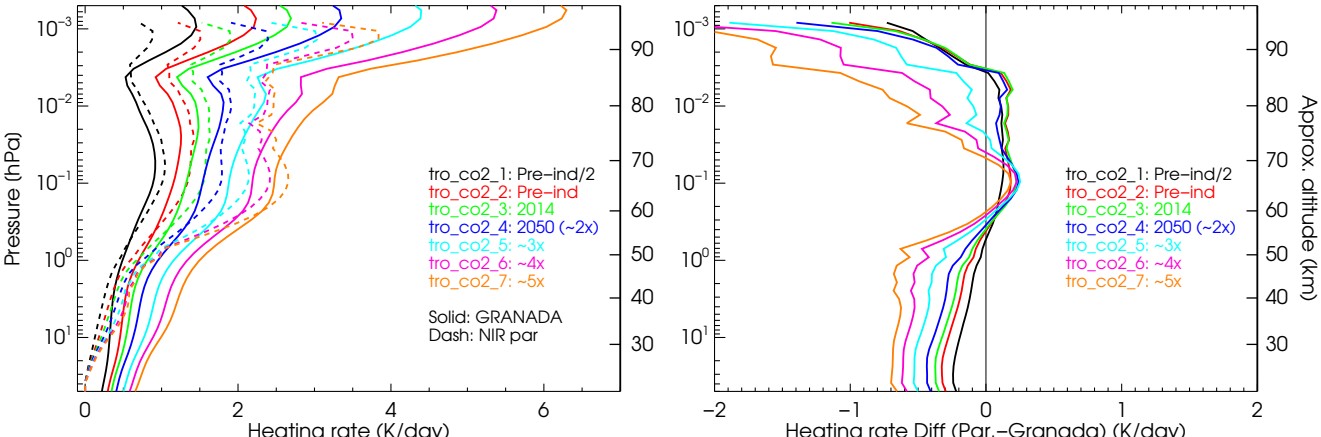

**Figure 28.** Solar NIR heating rates for the tropical atmosphere and a SZA of 44.5° computed by the solar NIR heating parameterization of Ogibalov and Fomichev (2003) and those obtained in the updated GRANADA model (see text). The right panel shows the differences in the heating rates of the parameterization minus those computed in GRANADA.

described below. In particular, the $J_{O3}$ rate used in these calculations is ∼10% smaller than in Jurado-Navarro et al. (2015) below 100 km, and thus leads to an [O($^1D$)] of ∼10% smaller below 90 km, but is very similar near 100 km. Above ∼100 km, the $J_{O2}$ coefficient used in the present calculations is about 40% smaller, than in Jurado-Navarro et al. (2015), leading to a similar reduction in [O($^1D$)]. Further, we updated the following collisional rates. The rate coefficient of $N_2 + O(^1D) \rightarrow N_2(1) + O$, has been increased by a factor of 1.08, and the collisional deactivation of $N_2(1)$ with atomic oxygen (which has an important role in the heating rates, see, e.g. López-Puertas et al., 1990), has been updated from a value of $4.5 \times 10^{-15} (T/300)^{1.5}$ cm$^3$s$^{-1}$ to $4.3 \times 10^{-15} (T/300)^{2.9}$ cm$^3$s$^{-1}$.

The results of the comparison are shown in Fig. 28 for the tropical atmosphere and an intermediate solar zenith angle (SZA) of 44.5°. The region of most importance for the $CO_2$ NIR heating rates is that comprised between 0.1 and 0.01 hPa (Fomichev et al., 2004). In this region, the differences between the algorithm of Ogibalov and Fomichev (2003) and GRANADA are in the range of +0.2 K/day to −0.5 K/day for $CO_2$ vmrs up to five times the pre-industrial $CO_2$ profile, e.g., about 10 to 15%. Hence, given that they have been computed with very different non-LTE models and the significant effect that parameters like the $CO_2$ vmr above ∼ 90 km, the collisional rate between $N_2(1)$ and O($^3P$), the O($^3P$) concentration itself, and the rate of exchange of $CO_2$ $v_3$ quanta with $N_2(1)$ have on these solar heating (see, e.g. López-Puertas et al., 1990), these differences are reasonable. Hence the new $CO_2$ cooling rate parameterization reported here can be safely used together with the $CO_2$ solar NIR heating parameterization of Ogibalov and Fomichev (2003) for $CO_2$ vmrs up to five times the preindustrial $CO_2$ profile.

# 9 Summary and Conclusions

An improved and extended parameterization of the $CO_2$ 15 $\mu$m cooling rates of the Earth's middle/upper atmosphere has been developed. It follows essentially the same method of the parameterization of Fomichev et al. (1998). The major novelty is its extended range of $CO_2$ abundances, ranging from $CO_2$ profiles with tropospheric values close to half of the pre-industrial





value to ten times that value. This extension of $CO_2$ profiles can still be safely applied to the parameterization of the $CO_2$ near-infrared heating of Ogibalov and Fomichev (2003) up to at least five times the pre-industrial $CO_2$ values, which is normally

combined with this cooling rates parameterization.

Other improvements or updates are as follows. It has an extended and finer vertical grid, increasing the number of levels from eight to 83. The $CO_2$ line list has been updated, from HITRAN 1992 to HITRAN 2016. Although the collisional rate coefficients affecting the $CO_2$ $v_1$ and $v_2$ levels are input parameters for the parameterization, in this version we have used more contemporary collisional rates, e.g., as currently used in the non-LTE retrieval of temperature from $CO_2$ $15\,\mu m$ emissions

of SABER and MIPAS measurements (García-Comas et al., 2008; García-Comas et al., 2023). The rate coefficients are in general of very similar magnitude, except for the collisional deactivation of $CO_2(v_1,v_2)$ levels by atomic oxygen, which is now larger by a factor of approximately two, e.g. close to its upper accepted limit. As a consequence of the larger range of $CO_2$ vmr profiles, the different NLTE layers for computing the cooling rates have been significantly revised. For example, it is worth mentioning that the lowermost altitude of the cooling-to-space approximation (the uppermost NLTE layer) has risen

from ~110 km up to 160–170 km.

The new parameterization has been thoroughly tested against line-by-line LTE and non-LTE cooling rates for: (i) the six $p$-$T$ reference atmospheres; (ii) the two most important input parameters (aside temperature), the $CO_2$ profiles and the collisional rate of $CO_2(v_1,v_2)$ by atomic oxygen; and (iii) for realistic measured temperature fields of the middle atmosphere (about 5000 profiles), including an episode of stratospheric warming with elevated stratopause. Further, to illustrate the improvements, such

comparisons have also been performed for the previous parameterization.

For the reference pressure-temperatures profiles, the errors of the new parameterization (mean of the differences in the cooling rates with respect to line-by-line calculations for the six $p$-$T$ atmospheres), are below $0.5\,\mathrm{K\,day^{-1}}$ for the current and lower $CO_2$ vmrs. For higher $CO_2$ concentrations, between about two and three times the preindustrial values, the largest errors are ~1–2 K day$^{-1}$, and are located near 100–120 km. For the very high $CO_2$ concentrations (from four to ten times the

pre-industrial abundances) the errors are also very small, below ~1 K day$^{-1}$, for most regions and conditions; except in the 110–120 km region where the parameterization overestimates them in a few K day$^{-1}$, ~1.5%. For these reference atmospheres, the new parameterization has a better performance for most of the atmospheric layers and temperature structures.

From the testing of the parameterization for realistic, current temperature fields of the middle atmosphere, we found that in general, the new parameterization is slightly more accurate. In particular, in the 105–115 km, the previous parameterization

overestimates the cooling rate by $1.5\,\mathrm{K\,day^{-1}}$, while the new one is very accurate. However, in the other height regions the difference is not so dramatic. The new parameterization has a better performance in the 80–95 km altitude region. Overall, the errors in the global mean profiles (about 1000 profiles) of the cooling rates of the new parameterization are below $0.5\,\mathrm{K\,day^{-1}}$, except between $5\cdot10^{-3}$ hPa and $3\cdot10^{-4}$ hPa (~85–95 km), where they can reach biases of 1–2 K day$^{-1}$. That region is the most challenging to parameterize because several $CO_2$ $15\,\mu m$ bands contribute to the cooling rate and they depend very much on

the temperature structure of the whole middle atmosphere (e.g., even outside this region).





We have further tested the parameterization against very rare and demanding situations, such as the temperature structures of stratospheric warming events with elevated stratopause. In these situations, however, the parameterization underestimates the cooling rates by 3–7 K day$^{-1}$ ($\sim$10%) at altitudes of 80–100 km.

This parameterization has some limitations (see Secs. 6 and 7.2). To apply specific approximations for the cooling rates, it

has been designed for fixed atmospheric regions where specific radiative transfer regimes prevail. Thus, its extension to a very large range of $CO_2$ abundances inevitably causes a loss of accuracy for extreme cases at specific atmospheric layers. A possible solution for future updates could be to use different extensions of the non-LTE regions (i.e. Fig. 11) for different abundances of $CO_2$. Likewise, this parameterization (like the original one) was devised for being used in GCM, that is to produce accurate global cooling rates, e.g., for all expected temperature profiles. Thus, the ability of the parameterization for computing accurate

cooling rates for specific and very different temperature gradients of the middle atmosphere is limited. The most critical region is between $5 \cdot 10^{-3}$ hPa and $3 \cdot 10^{-4}$ hPa ($\sim$85–95 km). From the comparison for realistic measurements we have found that, although the mean cooling rates are accurate, the standard deviation of the parameterized cooling rates is large ($\sim$6 K day$^{-1}$) in that region. As another example, see Sec. 7.2, we have shown that the parameterization underestimates the cooling rates in that atmospheric region (and the standard deviation is even larger) for elevated stratopause events. One may think that this is

caused because such extreme temperature profiles were not considered in the reference atmospheres. But, if so, the degree of improving the accuracy would depend on the chosen weighting factors for each reference atmosphere. Thus, its inclusion might give more accurate cooling rates for elevated stratopause episodes but would worsen the accuracy for other general situations. To conclude, parameterizations overcoming those limitations but retaining the goodness of this approach are highly desirable to be developed in the future.





*Code availability.*   The code will be made publicly available to the community.

## Appendix A: N$_2$, O$_2$ and O($^3P$) abundances

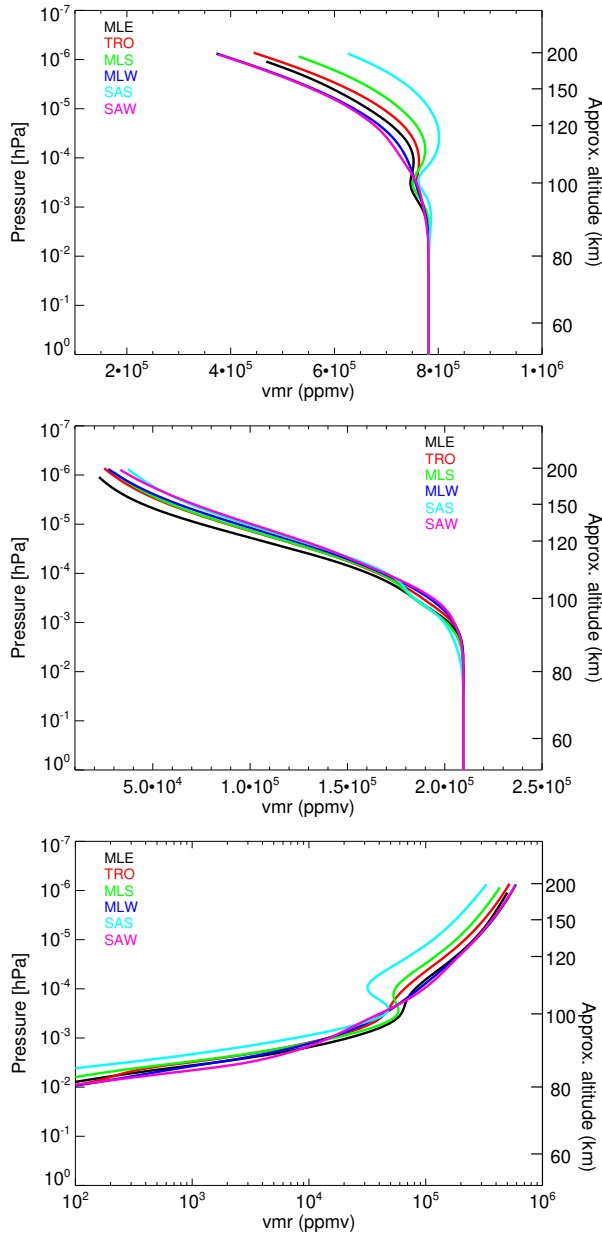

**Figure A1.** N$_2$ (top), O$_2$ (middle) and O($^3P$) (bottom) volume mixing ratio profiles for the respective atmospheric conditions used in the references calculations.



## Appendix B: LTE cooling rates for the reference atmospheres



**Figure B1.** The LTE cooling rates for the reference atmospheres shown up to the lower thermosphere. Note the different x-scales. The cooling rates extended up to the thermosphere are shown in Fig. B2 below.





**Figure B2.** The LTE cooling rates for the reference atmospheres. As in Fig. B1 but covering the thermosphere.



## Appendix C: Non-LTE cooling rates for the reference atmospheres for the entire altitude range.



**Figure C1.** Non-LTE cooling rates for the reference atmospheres covering the thermosphere. Those for the lower altitudes are shown above in Fig. 5. The cooling rates for the $CO_2$ volume mixing ratio profiles used to test the parameterization are also shown (dashed lines).



## Appendix D:  Non-LTE–LTE cooling rate differences.



**Figure D1.** Non-LTE–LTE cooling rates differences for the six $p-T$ references atmospheres (as in Fig. 8) but for the $CO_2$ vmr profile #6 ($4\times$ the pre-industrial values). The '*' symbol indicates the pressure level (in hPa) where the non-LTE–LTE difference reaches 5%. Note the different x-scales.



**Figure D2.** Non-LTE–LTE cooling rates differences for the six $p$-$T$ references atmospheres and the $CO_2$ vmr profile #3 (as in Fig. 8) but including the thermosphere.





**Appendix E: The $CO_2$ column amounts.**

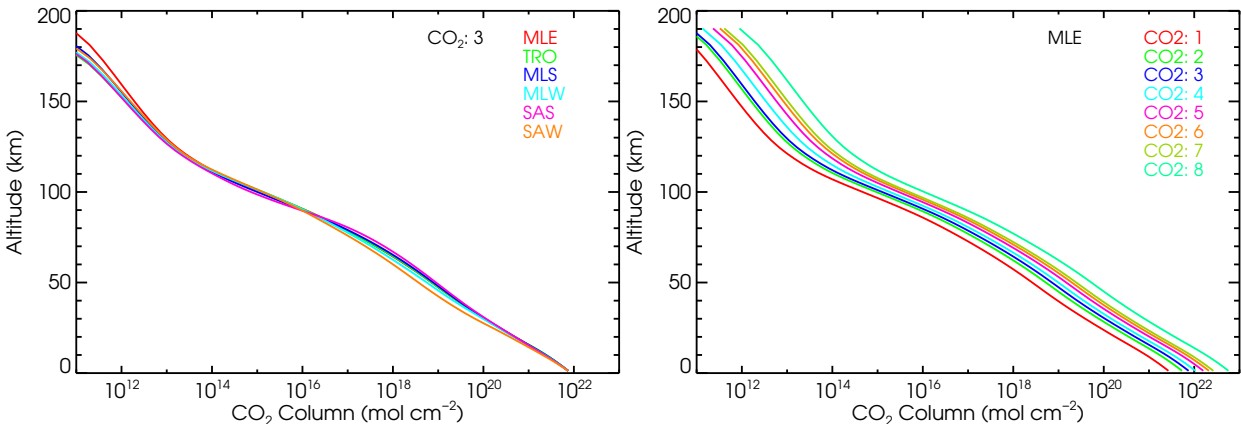

**Figure E1.** The $CO_2$ column amount $u$ as a function of altitude $z$ for several $p$-$T$ and the current $CO_2$ vmr profile, #3 (left), and for the eight $CO_2$ vmr profiles considered and the mid-latitude equinox (MLE) $p$-$T$ profile. They have been used in the representation of the escape probability function $L(u)$ in Fig. 12.



**Appendix F: Cooling rate differences of the current and previous parameterizations with respect to accurate cooling rates for the reference atmospheres**

We show in this section a comparison of the cooling rates of the current and previous parameterizations with respect to accurate cooling rates for several $CO_2$ vmr profiles for the six $p$-$T$ reference atmospheres. Note that similar figures for the $CO_2$ vmr profile #3 are shown in the main text (Fig. 13).



**Figure F1.** Comparison of the cooling rates of the current and previous parameterizations with respect to accurate cooling rates for the $CO_2$ vmr profile #1 for the six $p$-$T$ reference atmospheres.





**Figure F2.** Comparison of the cooling rates of the current and previous parameterizations with respect to accurate cooling rates for the $CO_2$ vmr profile #2 for the six $p$-$T$ reference atmospheres.





**Figure F3.** Comparison of the cooling rates of the current and previous parameterizations with respect to accurate cooling rates for the $CO_2$ vmr profile #4 for the six $p$-$T$ reference atmospheres.



**Figure F4.** Comparison of the cooling rates of the current parameterization with respect to accurate cooling rates for the $CO_2$ vmr profile #5 for the six $p$-$T$ reference atmospheres. The results of the previous parameterization are not shown because this $CO_2$ profile is beyond its applicability limits.




**Figure F5.** Comparison of the cooling rates of the current parameterization with respect to accurate cooling rates for the $CO_2$ vmr profile #6 for the six $p$-$T$ reference atmospheres. The results of the previous parameterization are not shown because this $CO_2$ profile is beyond its applicability limits.





**Figure F6.** Comparison of the cooling rates of the current parameterization with respect to accurate cooling rates for the $CO_2$ vmr profile #7 for the six $p$-$T$ reference atmospheres. The results of the previous parameterization are not shown because this $CO_2$ profile is beyond its applicability limits.



**Figure F7.** Comparison of the cooling rates of the current parameterization with respect to accurate cooling rates for the $CO_2$ vmr profile #8 for the six $p$-$T$ reference atmospheres. The results of the previous parameterization are not shown because this $CO_2$ profile is beyond its applicability limits.







**Figure F8.** As Fig. 16 but for the higher intermediate $CO_2$ vmr profile #10 (see Fig. 3).






**Figure F9.** As Fig. 16 but for the higher intermediate $CO_2$ vmr profile #11 (see Fig. 3). Note that the results of the previous parameterization are not shown as the $CO_2$ profile used here is beyond its applicability limits.



**Appendix G: Quantities used in the calculation of the accurate and parameterized cooling rates for the MIPAS temperatures**

We show in this appendix the $CO_2$ vmr and $O(^3P)$ abundances used in the calculations of the accurate and parameterized cooling rates. The LBL accurate cooling rates are also shown.

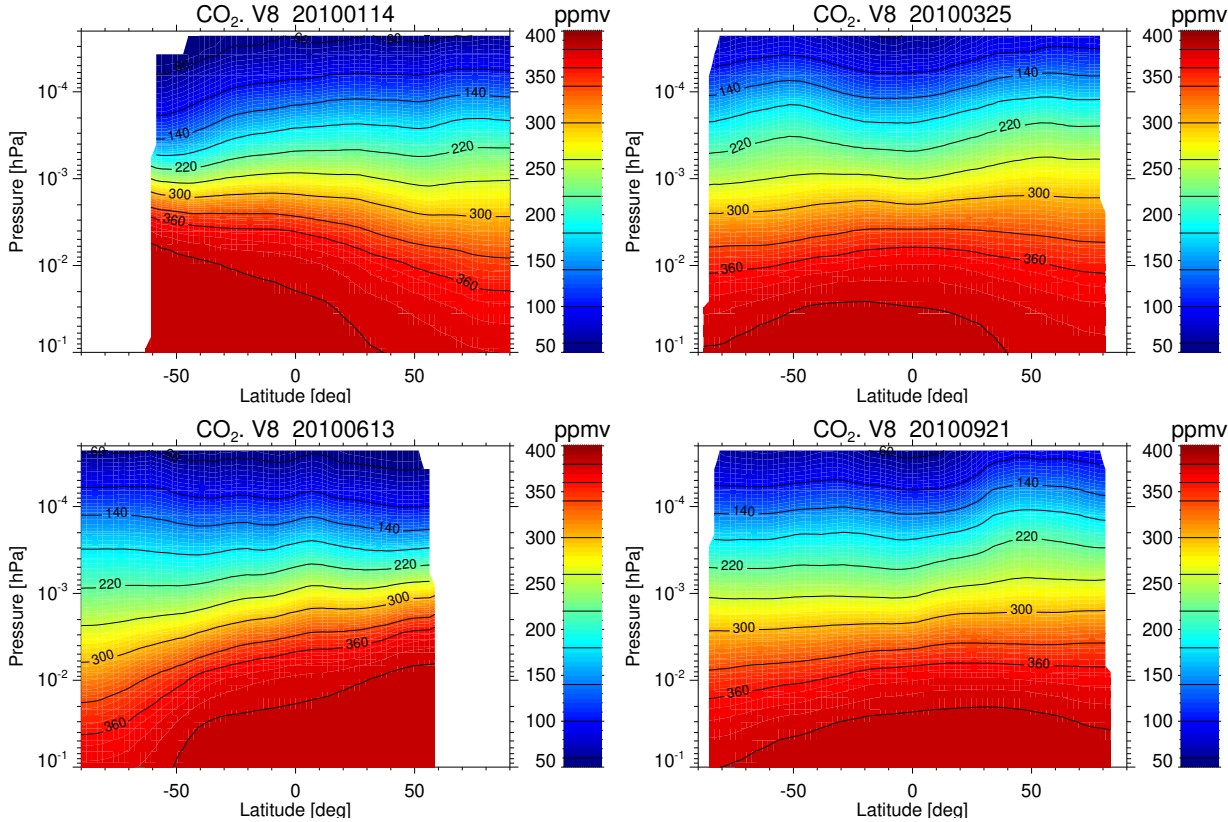

**Figure G1.** Zonal mean nighttime $CO_2$ vmrs used in the calculation of the cooling rates.



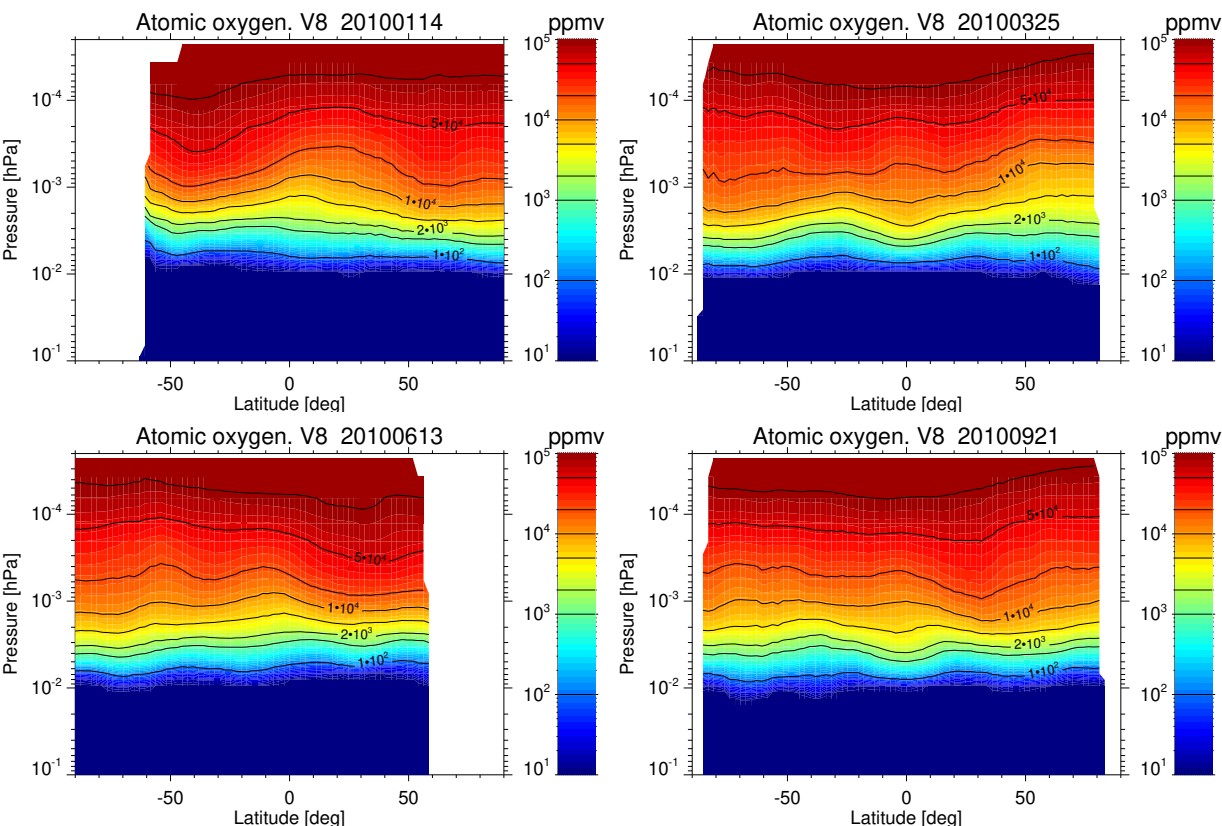

**Figure G2.** Zonal mean nighttime O($^3P$) vmr used in the calculation of the cooling rates.



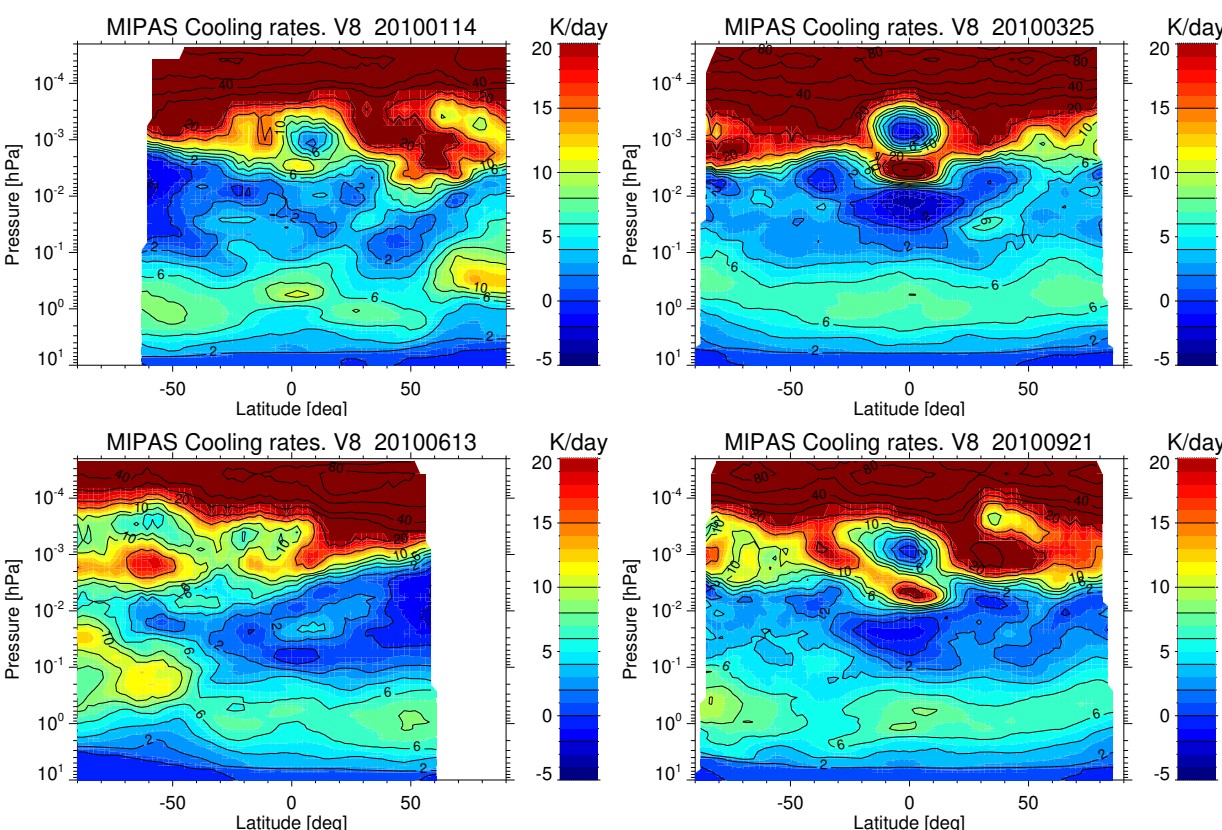

**Figure G3.** Zonal mean LBL accurate cooling rates obtained for the MIPAS nighttime temperature fields considered in this work.



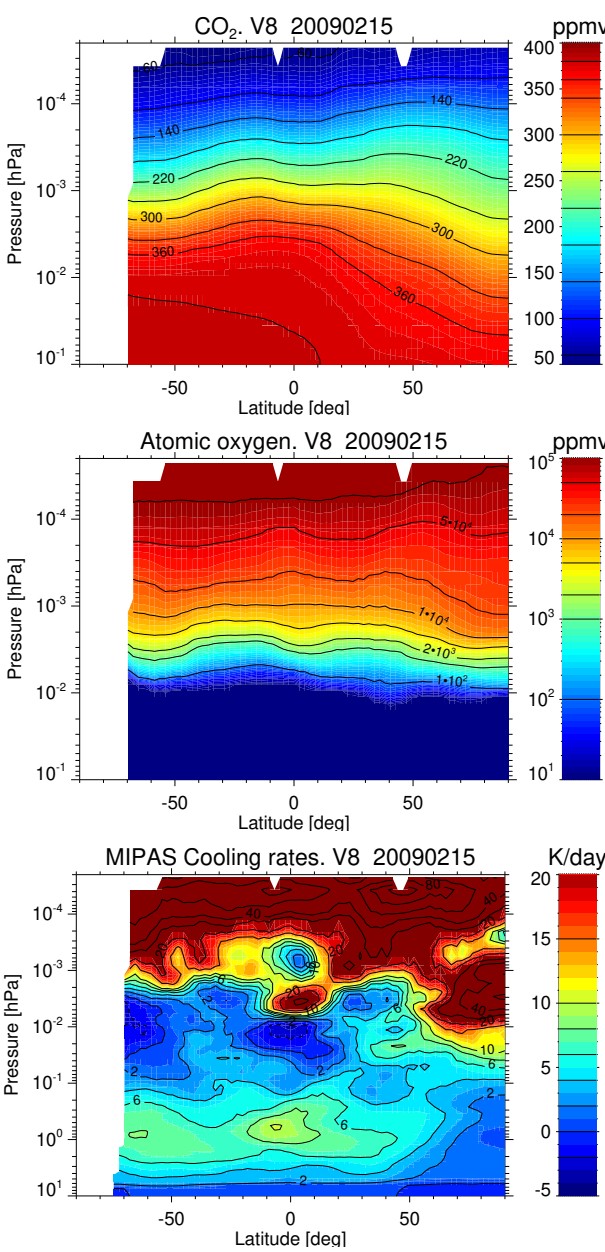

**Figure G4.** Zonal mean of the $CO_2$ vmr (top), $O(^3P)$ abundance (middle) and the LBL accurate cooling rates (bottom) for the elevated stratopause conditions (15 February 2009).



*Author contributions.* MLP performed the LTE and non-LTE reference calculations, participate in the adaptation of the original parametarization, wrote the manuscript and had the final editorial responsibility for this paper. FF led (together with BF) the adaptation of the original parametarization of Fomichev et al (1998), wrote the code of the new parameterization, performed all the tests of the parameterization and calculated the cooling rates of the previous and new parameterizations presented here. VF made a critical contribution for the adaptation of its original algorithm and to its update. BF co-led the adaptation of the original parametarization and designed the accuracy tests. DM provides the CO2 abundances and advise on the development of the parameterization for its use and implementation in climate models. All the authors contributed to the discussions, and provided text and comments.

*Competing interests.* The authors declare that they have no conflict of interest.

*Acknowledgements.* The IAA team acknowledges financial support from the Agencia Estatal de Investigación, MCIN/AEI/10.13039/501100011033, through grants PID2019-110689RB-I00, PID2022-141216NB-I00 and CEX2021-001131-S.





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
