# Peer review of "An improved and extended parameterization of the $CO_2$ 15 $\mu m$ cooling in the middle/upper atmosphere ( $CO2\_cool\_fort-1.0$ )"

_EGUsphere, 2023_

## Community Comment (CC1)

**Comments to the manuscript by Lopez-Puertas et al "An improved and extended parameterization of …"**

**Alexander Kutepov**
**The Catholic University of America**
**Washigton, DC**
**Email: kutepov@cua.edu**

OVERVIEW

The paper presents the revised Fomichev et al. (1998) matrix parameterization of the CO2 15 µm cooling rates of the Earth's middle and upper terrestrial atmosphere for both local thermodynamic equilibrium (LTE) and non-LTE layers. It is essentially based on the same approach as the Fomichev et al. (1998) parameterization. The main improvement is an extended range of CO2 abundances: whereas Fomichev et al. (1998) routine covered the range of CO2 concentrations with tropospheric values from 150 to 720 ppm the parameterization presented in the paper goes up to 3000 ppm of tropospheric CO2. Another minor improvement is the finer altitude grid of revised parameterization.

GENERAL COMMENTS

This is a large manuscript with very large numbers of plots both in main body and appendixes. Although it is well written and structured, many tests of a new routine repeat one another and, therefore, look excessive.

The work takes us back to the late 70s - early 80s of the last century, when new for that time techniques of the approximate 15-micron CO2 cooling calculations (both for LTE and non-LTE conditions) were developed, see (Kutepov, 1978) and (Akmaev and Shved, 1982). The revised version of the Fomichev et al, 1998 (hereafter F98) routine described in the manuscript does not represent any innovation and does not suggest any new option for the GCM users and developers.

This statement may be explained as follows. Any physical parameterization for GCM must be able to react as realistic as possible at steady changing local physical state of the atmosphere in the modeling process. This is particularly true for the infra-red radiation and its effects (local cooling or heating), which are critically important for adequate modeling of energy balance. It is well known that instantaneous p-T distributions in modern GCMs of middle and upper atmosphere exhibit very strong variability, caused by the superposition of tidal and gravity waves of different amplitude and vertical scales. Same is true for the atmospheric p-T distributions measured in both ground and space experiments. Therefore, the parameterization of the 15-micron CO2 cooling, which is a main radiative cooling mechanism of these layers, must properly react to this variability.

However, the matrix parameterizations of the 15-micron cooling are unable to provide adequate reaction to strongly disturbed T distributions. This was well known already 30 years ago for those (I was among them) who worked on developing the first version of the F98

parameterization. Demonstration of large parameterization errors for wavy T profiles was not included in the paper by Fomichev et al, 1993, however, it contained at least the warning addressed to its users *"It is recommended for the calculation of the radiative cooling for smooth temperature profiles, namely for profiles undisturbed by micro- and meso-scale motions."* When the updated 1993 routine was released by Fomichev et al, 1998 nothing changed, the revised routine was still unable to treat wavy distributions, see (Kutepov and Feofilov, 2023, hereafter KF23). Again, as in the 1993 paper, the accuracy of F98 was demonstrated exceptionally for smooth T profiles. However, the authors dropped from the paper text the warning for its users cited above. Since that time the F98 routine has been widely used in GCMs of middle and upper atmosphere for any T distributions.

It is also worth noting here that 15-micron $CO_2$ radiative cooling is strongly non-linear in respect to the temperature variations. Therefore, the zonal mean cooling, which is enthusiastically discussed in this manuscript, is not equal to that calculated for the zonal mean temperature. The authors, however, pay no attention to this fact!

Now back to the current manuscript. The authors invested large efforts to update the F98 routine. They declare that the revised routine (hereafter L-P.23) allows calculations of 15-mucron cooling with higher accuracy for current $CO_2$ vmr. It calculates with reasonable accuracy also the cooling for much higher $CO_2$ concentrations. Nevertheless, again the accuracy of L-P.23 is demonstrated only for very smoothed "standard" T distributions.

Meanwhile, the authors are aware about large errors F98 routine has for disturbed T profiles. D. Marsh was the co-investigator of a recent NASA grant (Kutepov, 2021), where I was the PI. He and his team received funding for testing the KF23 algorithm for calculating 15-micron non-LTE cooling, comparisons of this routine with F98 parameterization, and for installing KF23 routine in the WACCM model. This study showed that F98 causes very large cooling errors (up to 25 K/day) on wavy profiles. These errors are discussed in KF23.

Meanwhile, the authors of the manuscript are quite honest when, describing the main motivation of their study, they write: *"In our case we have the option of developing a completely new parameterization, to adapt other $CO_2$ parameterizations (as those cited above), or to extent and improve the parameterization of Fomichev et al. (1998). Attending mainly to practical reasons of promptness, we opted for the later."* The keyword here is "*promptness*".

This "*promptness*" looks somewhat strange after 25 years of no interest of the authors to the problem of fast and accurate calculation of radiative cooling for the Earth's GCMs. Knowing about the drawbacks of F98 routine and other matrix parameterizations we spent these years developing accurate non-LTE radiative transfer techniques, which are free of these drawbacks and are fast enough to be applied in GCMs. The results of this long-term study are summarized in KF23.

Finally, what kind of new product these authors developed with the main motivation "to be prompt"?

Again, I do not need my own judgement. It is enough to cite what the manuscript authors write when they describe large errors, much lagers than 0.5-1.0 K/day reported for standard profiles, which they observed for all profiles at latitudes northernmost of 50 N for a single non-standard situation with a pronounced elevated stratopause event:

*"**Both parameterizations underestimate the cooling in that atmospheric region. The new parameterization has, however, a better performance above about 80 km, but in the strat-warm/elevated stratopause region (80–100 km) it still underestimates the cooling by 3–7 K Day−1 (~10%)**".*

This looks like a confession that in a non-standard situation the new l-P.23 routine works no better than F98.

And further: ***"It seems clear that part of this underestimation is caused by the fact that such atypical temperature profiles (see Sec. 3.1) were not considered in the parameterization. However, its inclusion would not solve the problem as in the calculations of the coefficients a trade-off of the weighting of the different p -T reference atmospheres have to be chosen (see Secs. 5.1 and 5.2). Thus, it might ameliorate the inaccuracy for these elevated-stratopause events but would worsen the accuracy for other general situations. This manifests the difficulty/limitation of this method to provide accurate non-LTE cooling rates for all temperature structures (gradients) that we might find in the real atmosphere."***

I absolutely agree with this statement: this approach for parametrizing the 15-micron non-LTE cooling in the middle and upper atmospheric layers, which was applied in previous routine versions in 1993, in F98, and repeated in L-P.23 **is a deadened approach**.

In KF23 we discuss in detail its drawbacks. Briefly: it is impossible to adequately estimate the non-LTE cooling in the very variable atmosphere by dividing it in several altitude regions, where different techniques or expressions for cooling calculation (although linked in various ways) are used. Only exact algorithms, which rigorously describe the radiative energy exchange between various altitude layers and the non-LTE radiative field coupling with atmospheric heat reservoir may satisfy the current cooling accuracy requirements.

***It is my opinion that this paper in its current shape must not be published. It does not provide any significant improvement compared to previous work(s).***

To be published the manuscript requires significant major revision:

(1) it must demonstrate how revised routine works for T profiles disturbed by the strong waves. If the routine fails on the wavy profiles, but the authors still recommend it for further usage in GCMs, then (2) they need to justify that these errors have negligible or no effects on the GCM model results.

SPECIFIC COMMENTS

1. ***Why exact methods of the non-LTE CO2 cooling calculation cannot be applied in GCM?***

The manuscript authors write: ***The computation of the cooling under those non-LTE conditions requires the solution of the radiative transfer equation (RTE) which is a non-local problem and requires a large amount of CPU time. Therefore, the solving the RTE in general circulation models (GCMs) or climate models that extend in height above the stratopause is***

*impractical and efficient parameterizations of the CO2 infrared cooling have been developed and implemented 30 in such models.*

I disagree with this statement. It does not matter whether the LTE approach is applied or the non-LTE problem is considered, in both cases the calculation of cooling requires the solution of RTE, and is, therefore, the non-local problem.

Moreover, the computing costs of exact RTE solution in modern algorithms are not the main problem, which makes "***the solving the RTE in general circulation models GCMs impractical".*** The authors mean here not just RTE but the entire non-LTE problem. Inversion of large matrices to get the populations in the developed in 1950s CM (Curtis, 1956) matrix algorithm, which the members of this team utilize since 1980s, is most computational costly part of the non-LTE problem solution. The authors either are not aware of or ignore dramatic progress in the developing the non-LTE techniques, see, for instance, (Hubeny and Mihalas, 2015) and (Frish, 2021). Large matrix inversion costs make CM technique usage impractical in GCMs. We discussed this in the papers by Kutepov et al, 1998, Gusev and Kutepov, 2003, and in more details in KF23.

    *2.* **How the 15-micron cooling maximum in the lower thermosphere is formed?**

In the manuscript**: *Above the mesopause, the cooling rate rapidly increases following the enhancement of the kinetic temperature. Above about 130 km, the cooling rates decline because of the depletion of the CO2 vmr (see Fig. 7).***

Are the authors sure about this? I recall that in early publications of the 1980s about the 15-micron cooling it was demonstrated that cooling declines at higher altitudes even for constant CO2 vmr. Main problem at these altitudes is the rapid decrease of atmospheric pressure and the CO(v2) collisional quenching rate, which disconnects the 15-micron radiation from the atmospheric heat reservoir. If the authors do not agree with my comment, could they demonstrate that cooling stops decaying above 130 km when CO2 vmr is constant?

    *3.* **Why the CO2-O quenching rate coefficient ~ 6.0e-12 cm3 s −1 was used as the basic one for revision of the F98 parameterization?**

The authors write that they used for updating the F98 parameterization the quenching rate coefficients which are very similar to those used for its development "***… except the k CO2−O rate (process 1c in Table 1) that has been considered here with its upper limit. That is, about a factor of two larger than in the parameterization of Fomichev et al. (1998). This rate coefficient is not well known with uncertainties of the order of a factor of two (see, e.g., García-Comas et al., 2008). While laboratory measurements are in the range of 1.5 to 2·10−12 cm3 s −1 the values derived from atmospheric observations are close to 6·10−12 cm3 s −1. ….."***

First, measured in laboratory and retrieved from the atmospheric observation values of k differ not by a factor 2, but by 3-4. Additionally, if one accounts for the k retrievals by Feofilov et al, 2012, which involved the ground-based lidar temperature measurements, then tis factor will be 3-6.

Table 1 shows that the authors selected k ~ 6.0e-12 cm3 s −1 for updating the parameterization. Although k is supposed to be an input parameter in both F98 and L-P.23 routines, however, the previous one was optimized in the transition region from LTE to non-LTE for k ~ 3.0e-12 cm3 s −1, whereas new one is optimized for 6.0e-12 cm3 s −1. The authors tell in manuscript that this causes additional differences between F98 and L-P.23 in the transition area even when both apply the same rate coefficient, and then explain the reason why they selected higher rate for optimizing L-P.23 as *"… we have optimized it for the high value (see Table 1), as this value has been used in the most recent non-LTE retrievals of temperature from SABER and MIPAS measurements"*.

It is not enough, however, to say higher k was selected because this rate provides more reasonable T retrievals from space observation. As the authors know to validate these T measurements current GCMs apply twice lower rate 3.0e-12 cm3 s −1. If the authors recommend, which follows from the text, to use L-P.23 with the most high k, then can they demonstrate how this affects the GCM (for instance the WACCM model) runs compared to those with twice lower k? Does this provide better fitting of measured temperatures?

**4.* **Testing the parameterization for measured temperatures**

The Fig.21 of manuscript shows an example of the MIPAS nighttime temperature profiles (15 February 2009) used for verifying the parameterization accuracy. The authors note large variability of the measured temperature profiles. These individual profiles are the good inputs for the revised parameterization to show how it works for strongly disturbed T profiles.

The authors have obviously performed these tests, but they do not show these results. Instead, they write "*The results are presented in Fig. 22 for the zonal mean of the differences for two days of solstice and two days of equinox conditions and in Fig. 23 as the global mean difference for all latitudes for each of the four individual days*." Why only these mean values are shown? Obviously, the averaging smashes the errors obtained for individual profiles, for which we observed in our study of F98 parameterization presented in KF23 errors up to 25 K/day. Meanwhile these large errors in our study have generally concentrated in the altitude region around 90 km, exactly where the RMSs in Fig.25 of manuscript are maximized reaching 8-9 K/day. In our paper we explain why F98 works badly in this transition region. L-P.23 has the same problems and is nothing better.

**5.* **The accuracy of L-P.23 for smooth T profiles**

I wrote above that L-B.23 will not be any better than the F98 for disturbed T profiles. But how about the smoothed standard T profiles? The accuracy of cooling calculations with L-P.23 less than 0.5 K/Day for preindustrial CO2 for smooth profiles is also questionable.

(1) The non-LTE model, which is used in this study for the reference calculations to optimize L-P.23 and then to check its accuracy, includes only 18 15-micron bands. From my point of view this model itself is not accurate. In the routine we suggest in KF23, which utilizes the exact solution of the non-LTE problem, we use the same bands to calculate nighttime cooling for 400 ppm of CO2 with an error not higher than 1 K/day for any T profile, including disturbed by strong waves. For smooth profiles this error reduces to ~ 0.2

K/Day. These errors were estimated by comparing KF23 routine with our reference model, which comprises 60 vibration levels of 5 CO2 isotopic species and hundreds of bands. So, the exact algorithm compared to the exact algorithm showed 0.2 K/day error when a reduced set of bands was used. I doubt that the error of 0.5 K/day the authors report for the L-P.23 routine for smooth profiles after comparing it with a very simplified "reference model" is true. It must be higher.

(2) The authors write that contribution of the heating due to the absorption of solar radiation in the near infrared CO2 bands at day time is negligible compared to the 15-micron cooling. However, 2-3 K/day (for current CO2) do not look negligible for the routine, which accuracy is declared to be about ~0.5 K/day. We tested in detail the (Ogibalov and Fomichev, 2003) parameterization of this heating, found this warming increasing for some wavy T profiles, and made sure this parameterization cannot guarantee the error of the KF23 routine at daytime to be lower than 1 K/day. As a result, we extended our KF23 daytime model up to 26 CO2 vibrational levels and 56 bands to satisfy this requirement.

(3) The authors say nothing about how they account for the cooling effect of the micro-scale sub-grid T disturbances. Kutepov et al., 2007 and Kutepov et al., 2013 showed that these temperature fluctuations cause near the mesopause an additional cooling up to 3 K day-1. I draw the authors' attention to the results shown by Kutepov et al, 2013 in Fig.1.5. It demonstrates one of the runs of the Leibniz Institute Middle Atmosphere (LIMA) model with the 15-micron cooling modified to account for the sub-grid T disturbances. It is shown that very minor variation of cooling (not higher than 2-3 K/day) lead to significant changes of the monthly and zonal mean temperatures for July 2005.

I am sure that errors of L-P.23 routine are much higher than 2-3 K/day and can be hardly reduced due to the deficiencies of the methodology applied. These errors will obviously have a strong impact on the GCM results.

**6. The code availability**

It seems the manuscript was submitted as the GMD "Development and technical paper". If it is correct, then "**The code should be made available, and a model availability paragraph must be included".** The code is, however, not available.

Once the code is available, I will demonstrate that its errors are much higher than reported in the manuscript.

**References**

Akmaev, R. A. and Shved, G. M.: Parameterization of the radiative flux divergence in the 15 μm CO2 band in the 30–75 km layer, Journal
of Atmospheric and Terrestrial Physics, 44, 993–1004, https://doi.org/10.1016/0021-9169(82)90064-2, 1982.

Curtis, A. R. and Goody, R. M.: Thermal Radiation in the Upper Atmosphere, Proceedings of the Royal Society of London Series A, 236, 193–206, https://doi.org/10.1098/rspa.1956.0128, 1956.

Feofilov, A. G., Kutepov, A. A., She, C. Y., Smith, A. K., Pesnell, W. D., and Goldberg, R. A.: $CO_2(v_2)$-O quenching rate coefficient derived from coincidental SABER/TIMED and Fort Collins lidar observations of the mesosphere and lower thermosphere, Atmospheric Chemistry & Physics, 12, 9013–9023, https://doi.org/10.5194/acp-12-9013-2012, 2012.

Fomichev, V. I., Kutepov, A. A., Akmaev, R. A., and Shved, G. M.: Parameterization of the 15-micron $CO_2$ band cooling in the middle atmosphere (15-115 km), Journal of Atmospheric and Terrestrial Physics, 55, 7–18, https://doi.org/10.1016/0021-9169(93)90149-S, 1993.

Fomichev, V. I., Blanchet, J.-P., and Turner, D. S.: Matrix parameterization of the 15 μm $CO_2$ band cooling in the middle and upper atmosphere for variable $CO_2$ concentration, Journal of Geophysical Research: Atmospheres, 103, 11 505–11 528, https://doi.org/10.1029/98jd00799, 1998.

Frisch, H.: Radiative Transfer. An Introduction to Exact and Asymptotic Methods, Springer, 2022.

Gusev, O. A. and Kutepov, A. A.: Non-LTE Gas in Planetary Atmospheres, in: Stellar Atmosphere Modeling, edited by Hubeny, I., Mihalas, D., and Werner, K., vol. 288 of Astronomical Society of the Pacific Conference Series, p. 318, 2003.

Hubeny, I. and Mihalas, D.: Theory of Stellar Atmospheres, Princeton University Press, 2015.

Kutepov, A. A.: Parametrization of the radiant energy influx in the $CO_2$ 15 microns band for earth's atmosphere in the spoilage layer of local thermodynamic equilibrium, Akademiia Nauk SSSR Fizika Atmosfery i Okeana, 14, 216–218, 1978.

Kutepov, A. A., Gusev, O. A., and Ogibalov, V. P.: Solution of the non-LTE problem for molecular gas in planetary atmospheres: superiority of accelerated lambda iteration., Journal of Quantitative Spectroscopy and Radiative Transfer, 60, 199–220, https://doi.org/10.1016/S0022-4073(97)00167-2, 1998.

Kutepov, A. A., Feofilov, A. G., Medvedev, A. S., Pauldrach, A. W. A., and Hartogh, P.: Small-scale temperature fluctuations associated with gravity waves cause additional radiative cooling of mesopause the region, Geophysical Research Letters, 34, L24807, https://doi.org/10.1029/2007GL032392, 2007

Kutepov, A. A., Feofilov, A. G., Medvedev, A. S., Berger, U., Kaufmann, M., and Pauldrach, A. W. A.: Infra-red Radiative Cooling/Heating of the Mesosphere and Lower Thermosphere Due to the Small-Scale Temperature Fluctuations Associated with Gravity Waves, pp. 429–442, Springer Netherlands, Dordrecht, https://doi.org/10.1007/978-94-007-4348-9_23, 2013.

Kutepov, A. A, (PI), Study of IR emissions of $CO_2$ and OH in the mesosphere and lower thermosphere using SABER/TIMED observations, NASA Award Number NNX17AD38G, 2021

Kutepov, A. A, and Feofilov A. G.: New Routine NLTE15µmCool-E v1.0 for Calculating the non-LTE CO2 15 µm Cooling in GCMs of Earth's atmosphere, Geophysical Model Development (discussion), https://doi.org/10.5194/gmd-2023-115, 2023.

Ogibalov, V. P. and Fomichev, V. I.: Parameterization of solar heating by the near IR CO 2 bands in the mesosphere, Advances in Space Research, 32, 759–764, https://doi.org/10.1016/S0273-1177(03)80069-8, 2003.

---

## Author Comment (AC3)

**Response to Reviewer 1**

We would like to thank the referee for the helpful review and the constructive comments. His/her comments are given below in black and our responses in blue.

**General comments:**

1) In my opinion, the manuscript tries to cover too many topics despite the fact that the title states that this is an update to existing parameterization. Well, it doesn't hurt to recall the main milestones of the previous work, but the manuscript in its current form is "unfocused". If it is intended to be a text-book chapter on non-LTE in the CO2 bands, then the same information could be found in the first authors' book (Lopez-Puertas and Taylor, 2001). If it is supposed to be a technical paper then it should put more stress on technical aspects like the accuracy, interfacing, and so on. In addition, the text is overloaded with figures. I counted more than a hundred (!) of individual panels in the main part that consists of 41 pages. This is definitely an overkill and one can tell the same story in a much more concise way. I suggest the authors to shorten the manuscript and to keep only the essential parts. For example, all the inputs can be gathered on one two-panel plot, where the left-hand side would be occupied by the reference temperatures and the right-hand side would show the profiles of CO2 (only one is needed) and atomic oxygen, which is crucial for the estimate of the cooling rate. The first figure is not informative at all and can be easily omitted. The large number of panels in figures like Fig. 5 does not add much to the understanding of the principle of the radiative cooling or its calculation, and so on. Overall, I would reduce the number of non-essential figures to a minimum and put more efforts to a description of the routine itself, its implementation to a GCM and its limitations.

**Reply:** The point is well taken. We propose the following actions.
- About the text, we will retain only the major points driving the CO2 coolings, e.g their dependence on the pT structure, on the CO2 and O vmrs, the different bands contributing to them and how these contributions and their deviation from LTE depend on the p-T and CO2 and O(3P) vmr profiles. These parts are essential to understand the difficulty and the rationale of the parameterization. Further, note that we are extending the CO2 vmr range and this implies changes in the LTE to non-LTE switching regions, e.g. in the boundaries of the previous version of the parameterization. We think these changes have to be justified and explained to the reader. The part of the paper which is somehow redundant with Lopez-Puertas and Taylor (2001) is not that much, only part of Sect. 4.2 where we describe the reference non-LTE cooling rates. Here we focus more on the details of the input parameters (collisional rates, etc.) for the non-LTE model, which we think are important, for example, if compared to other parameterizations. In any case, the point is taken and we will try to reduce as much as possible the text of Sec. 4.2.

- About the figures, following your suggestions, we propose the following actions:
   a) To move all figures of the Appendices to Supplementary information.
   b) To remove Fig. 1.
   c) To combine Figs. 2 and 3 in one 2-panel fig. The referee also proposes to include one Fig for O. How can it be done for the 3 parameters and only 2 panels? Further, combining CO2 and O in one figure is difficult because of the different (linear and log) scales. We propose a 3-panel 1-col fig. for p-T, CO2 and O.
   d) Fig. 5. Sorry but we do not agree with the referee on this point. This is an essential figure to discuss the points mentioned above as a reference for the line-by-line accurate cooling rates. Nevertheless, it has been reduced to keep only 4 essential panels.
   e) Fig. 6 Contributions of the different bands. It has been reduced to only 4 panels (as Fig. 5).

f) Fig. 7 Contributions of bands in the lower thermosphere (1 panel). It will be moved to the Appendix

g) Figs. 8 and 9. Comparison of non-LTE and LTE cooling rates. Fig. 8 will be reduced to four panels, but not moved to the appendix because the other referee suggested that we should describe the non-LTE-LTE differences in order to show the relevance of the non-LTE errors of the parameterization.
Fig. 9, however, will be moved to the Appendix.

h) Fig. 10. The effects of the K(CO2-O) collisional rate. We will show only 4 panels, consistent with previous figures.

i) Fig. 12 (escape probability functions). It will be moved to the Supplement.

j) Figs. 13, 14 and 15 compare the previous and current par. They are essential and are kept in the main text.

k) Figs. 16 and 17, and 18 and 19 show the accuracy of the parameterization for intermediate CO2 vmr profiles and for the K(CO2-O)/2. We will keep in the main text only Figs. 17 and 19 (the summaries) and move Figs. 16 and 18 to the Appendix.

l) Fig. 20: MIPAS temperature profiles. It will be kept and likely modified (see below).

m) Fig. 21. MIPAS zonal mean temperature maps for 4 days. Will be moved to the Appendix.

n) Fig. 22. Zonal mean of the differences between the previous and new par. for the MIPAS temperatures (8 panels). We suggest keeping in the main body only 4 panels (solstice (January) and equinox (March) and moving the other 2 panels to the Appendix.

o) Figs. 25, 26, and 27 show the performance of the parameterizations for the elevated stratopause conditions. Fig. 25 will be moved to the Appendix. The others will be kept.

Overall, the number of Figs. in the main text will be reduced from 28 to 19 (and some of them with a reduced number of panels); and the number of Figs. in the appendix will be reduced from 20 to 8. 21 figures will be moved to a Supplement.

About "... put more efforts on the description of the routine itself, its implementation to a GCM and its limitations", the description and its limitations are already discussed in detail. Its implementation is described in the Readme file of its distribution and, as suggested by the other referee, we will add a paragraph about the altitude region where it should be used. That is, the parameterisation is specifically developed for the CO2 15 μm **non-LTE region**, and, although it also works for the LTE region, it should be used with caution in that region. We recommend that the users utilize their radiation scheme in the LTE region and this parameterization in the non-LTE region (above~50, 60 km).

2) Regarding the routine, I did not see any reference or repository for it provided along with the manuscript. As far as I understand, this is now a requirement for EGUsphere journals, so I wonder when and how did the authors plan to publish the routine. Besides a general conformity with the journal's rules, it would be advisable to get a self-consistent compilable code with the cooling rate parameterization routine called from the main code with some standard atmospheric profile, which could be replaced with a test profile by the user. In the next paragraph, I explain the current problem I see with the accuracy of the updated parameterization, and I guess the GCM-modelers would have liked to test it themselves prior to implementing it to the model. Summarizing this point, the authors must provide a ready-to-use code and provide an instruction for its compiling and running both as a standalone routine and as a part of the test code, that might be useful not only for GCM-modelers, but for other researchers wanting to get a quick estimate of non-LTE cooling profile for a given atmosphere.

**Reply:** The referee is fully correct. We were not aware of that and thought of providing it during the review process. We will provide it in its final version as a Fortran 90 code jointly with the revised version of the manuscript. In the meantime, the parameterization is now available provisionally as a Python routine, please see the reply to the Editor comment 1 (https://zenodo.org/doi/10.5281/zenodo.10547026)

We thank the referee for the advice and recommendations. We have already taken them into account and are providing input and output files for the 6 reference atmospheres, for two CO2 concentrations (#3 (current) and #8 (10x preindustrial values)).
We have a modeller in our team and we will incorporate all of his comments about its implementation in the WACCM model in the documentation of the code. Further, the referee made a very good point to provide a version that could be used by any researcher and not only in GCMs. We will facilitate its use for this purpose.

3) The authors show that the routine works fairly well for the multitude of atmospheric profiles they use in the tests. This is quite impressive given a large variability of CO2 concentrations used in the study. Still, I cannot consider these tests to be fully representative because the real atmospheric temperature profiles are far from those shown in Fig. 2 of the manuscript in terms of vertical variability. Moreover, they are even worse than the ones shown in Fig. 20 because the averaging kernels of MIPAS instrument in the middle atmosphere are broad (e.g. Dinelli et al., 2021) and, therefore, this instrument cannot capture any wave shorter than 5-7 km. In addition, MIPAS is a limb measuring instrument, so it washes out horizontal inhomogeneities and this further reduces its vertical resolution. At the same time, the gravity waves (GWs) are known to be composed of several waves of different wavelengths and their amplitude increases with height up to the top of the "wave-turbopause" layer at 90-110km (e.g. Offerman et al., 2007; Ge et al., 2022), which overlaps with the non-LTE-affected area. I believe, the MIPAS profiles used in the study do not represent a full picture of a gravity-wave perturbed temperature profiles, so the tests performed with them show only a partial truth. Moreover, large RMS values presented in Fig. 24 even for these, partially smoothed, profiles make me think that the real accuracy of the parameterization is far from 1−2 K/day as it is stated in the abstract (I do not consider the specific case of elevated stratopause, which is addressed; I'm talking about the accuracy of "regular" profiles). If the tests I suggest below do not disprove my suspicions, this wouldn't cancel the suggested parameterization. But, the GCM modelers will have a clear picture of what they will use and act accordingly if their models produce lifelike GW-perturbed temperature profiles.

**Reply:** Overall we agree with the referee in testing a few more cases, although we are quite confident that the results will not change significantly (see below). However, let us first clarify a couple of points about the MIPAS temperatures used. It is true that they have a limited vertical resolution and do not fully recover the actual pT altitude structure. However, they are also affected by the noise of the instrument, which frequently maps in the profiles as a vertical variability larger than the real one. The given reference about MIPAS (Dinelli et al., 2021) describes the retrieved temperatures from the MIPAS **nominal** mode. The MIPAS data we used in the manuscript are those retrieved from the middle atmosphere (MA) mode, which has a better vertical sampling and finer AKs than the nominal mode ones (see García-Comas et al., 2023).
Hence MIPAS profiles (eg Fig. 20) are not that far away from real ones, we think.
We are including below a couple of Figs. showing some profiles extracted from MIPAS data for 15 February 2009 (Fig. 25). They clearly show wave structures and the very unusual pT profiles during the elevated stratopause.

[Figure]

We concur with the referee that the values of 1-2 K/day given for the accuracy are for the reference p-T profiles. When dealing with individual p-T profiles the errors are larger, as demonstrated in its application to MIPAS data, particularly for polar summer and polar winter conditions. For that reason, we did the test with the MIPAS individual temperature profiles and even showed the case of elevated stratopause conditions. However, those errors for MIPAS can be considered representative.

In general, we think the most proper way of expressing its accuracy is by a mean error value (bias), which will inform us about its accuracy in a global sense, and by the RMS (not the standard deviation), as an appropriate estimate of its error for individual profiles as assessed from a significant sample.

We will clarify in the abstract that the values now given for the accuracy are for the reference atmospheres, and will include the RMSs as an estimate of the individual profiles for the different atmospheric conditions. Further, we will also assess similarly its accuracy for the specific lidar cases described below.

4) Here is the test I believe is necessary to conduct in order to properly estimate the accuracy of the suggested parameterization. Instead of using the MIPAS profiles and showing the averaged effects, I ask the authors to use the individual temperature profile coming from a ground-based lidar (e.g. Spitsbergen (78°N, 15°E), ALOMAR at Andøya (69°N, 16°E), Kühlungsborn (54°N, 12°E), Boulder (40°N, 105°W), Fort Collins (41°N, 105°W), Logan (42°N, 112°W), Arecibo (18°N, 293°E), Cerro Pachon (30°S, 71°W), or McMurdo (78°S, 167°E).) Since these lidars do not cover the whole atmospheric profile, in the lower atmosphere one can take a corresponding smooth profile from the reanalysis or from any other model. Among the profiles provided by the lidar teams, one has to pick up the ones, which clearly show the GW-signature (at least, the wave of +/- 10 K amplitude at 100km), and perform the exact and parameterized calculations of radiative cooling rate for these profiles. It would be also interesting to see how different wavelengths affect the accuracy, because the non-LTE effects are different in the case of a short-and large-scale perturbations. I have to stress here that due to a nonlinear nature of temperature perturbation effects on non-LTE cooling rates, these results cannot be replaced with the ones obtained for averages.

**Reply:** We will check several of the lidar profiles suggested by the referee. We do not know now (we will check) how different they are from the variability shown above for MIPAS. If they are significantly different, we will assess the accuracy of the parameterization for a statistically representative ensemble of lidar p-T profiles in addition to the MIPAS profiles. We will assess its accuracy for the specific lidar cases in a similar way as described above.

**Minor comments and technical corrections**

Lines 4 and 41: this statement is too general. In fact, there are ways of including the non-LTE calculations to GCMs. For example, this was done for Martian GCM (Hartogh et al., 2005) and I don't see why this can't be done for Earth. The corresponding manuscript is still in discussion in the same journal, so it can't be referenced, but the authors claim that the same approach works for the Earth's atmosphere.

**Reply:** We agree that there are different ways of including non-LTE in GCMs but generally they are all very CPU time consuming. "The corresponding manuscript … " Does the referee mean the manuscript submitted recently by Kutepov et al.? We will include the mentioned reference and a citation to the manuscript recently submitted to this journal in the revised version.

Line 11: since the atomic oxygen is a variable component and the reaction 1c is directly linked to it, it would be good to have it as a variable parameter, see the second general comment

**Reply:** The user will have the option of including both, the atomic oxygen concentration profile and the collisional rate K1c. This is specified in the documentation of the code provided now.

Line 17: the measured profiles themselves is not a ground truth because of the vertical resolution, see the third general comment. Moreover, one cannot average the results for individual runs and present this difference as an error, because the GCMs accumulate the error due to non-linearity of the processes.
**Reply: First point**: We agree it is not the ground truth but we think it is not thus far (see plots above). We will also analyse lidar pT profiles and test the parameterization for them.
**2nd point:** We agree. Still, that average gives us a measure of the "error" (systematic) in a global sense (eg the cooling at high altitudes in the previous version is always about 2K/Day larger than in the line-by-line calculations). But will also provide the RMS as an estimate of its error for individual profiles as assessed from a significant sample.

We do not understand what the referee means by "... because the GCMs accumulate the error due to non-linearity of the processes." Does he/she refer to the fact that the cooling is proportional to exp(-kE/T) and not to T? We can only assess cooling rate errors induced by the parameterization, but not non-linear effects on modelled quantities such as temperature and winds.

Line 58: Figure 1 is not informative, see the first general comment
**Reply:** OK. It will be removed.

Lines 120-150: I guess, the readers will be confused here. It looks like GRANADA non-LTE code cannot work in LTE mode whereas normally it requires one line or one flag to calculate LTE populations. Could you, please, explain?
**Reply:** GRANADA computes non-LTE populations and non-LTE cooling/heating rates. We do not need a code for computing LTE populations and GRANADA has not been specifically designed/coded for computing LTE cooling rates. We already clarified in the manuscript that we use KOPRA for LTE cooling rates using a Curtis Matrix approach.

Line 125: I didn't find any mention of line mixing effects here. Are they not important?
**Reply:** KOPRA does include line mixing and it was used in these calculations. We will include a sentence stating it.

Line 135: I didn't get the phrase about the "oscillations in RFM results". Could you, please, clarify?
**Reply:** We refer to the minor oscillations in the RFM cooling rate profile, e.g. around 50 km, 60 km, 65 km, and 85 km.

Line 145: And what about the results without this additional iteration?
**Reply:** In short, the cooling rates would be less accurate**.** Without that final iteration (for specified accuracies for the vibrational temperatures), the non-LTE populations, e.g. the vibrational temperatures do not change significantly (see Funke et al. 2012). However, we did not describe in that work the non-LTE cooling rates. To properly calculate the cooling rates, in addition to the specific procedure described in that work, a final iteration to compute the radiative field in all bands is required in order to properly account for the overlapping between the different bands. The effect of not including it is to have slightly less accurate cooling rates.

Line 171: I recall that the difference is about 2-3K, and I wouldn't call this a small change, so one should not neglect the daytime vs nighttime differences. This is addressed in Sect. 8, so the wording has to be changed here
**Reply:** It is not that large, about 1-2 K at most (see e.g. Fig. 3 in López-Puertas et al., 1990) which in relative terms (the day/night differences are larger above about 90 km) are very small.
Our more recent calculations of day/night differences of the CO2 15 μm cooling rates are smaller than 1K/day for all pT profiles, at any altitude and for CO2 vmrs up to 5 times the pre-industrial

value, except for MLS and SAS pT profiles near 105 km and CO2 vmr 4x and 5x the pre-industrial value.
We already state that this point is addressed in Sec. 8 (line 175).

Lines 200-215: all these descriptions are correct, but I doubt that they add something to the understanding of the routine or its accuracy, see the first general comment on the scope and focus of the manuscript.
**Reply:** We already state the reason why including this discussion,  lines 201-202: *"This comparison is useful from a physical point of view and it is required to establish the boundaries of the different atmospheric regions of the parameterization".* We have moved Figs. 8 and 9 to the Appendix and will try to lighten the text in that section.

Lines 220-225 (see also the comment to lines 437-462): if we assume that the k(CO2-O) used in the models is equal to 3 x 1e-12 cm3 s-1, then this experiment shows the effects of quadrupling the atomic oxygen mixing ratio. In addition, the reader might be misled by switching from atomic oxygen to k(CO2-O) and back. It would be enough to present this numerical experiment as the one performed for doubled (or quadrupled) atomic oxygen.
**Reply:** That is what we did, just to half the k(CO2-O) nominal value of ~6e-12 cm3 s-1. We will clarify this in the text and change "perturbed the kCO2−O collisional rate by a factor of two" to "divided the kCO2−O collisional rate by a factor of two".

Lines 248-395: in fact, this deserves a general comment, but I cannot suggest to modify the general approach, it is what it is, and it works for smooth profiles for a current atmosphere. But what if the atmosphere changes some day and the LTE/NLTE1,2,3,4 regions become different? Let's imagine that a modeler wants to calculate cooling rates for an Earth-like atmosphere, but with a different vertical temperature structure. How can he/she tell the limits of applicability of this parameterization?
**Reply:** Very likely, it will not work with high accuracy for those situations. This parameterization has not been designed from its origin for any planetary or p-T temperature profile. It has been specifically tailored for the current-like Earth's atmosphere and projected high CO2 vmr in the future.

Line 304: this is just one of the examples of the problem outlined in the previous comment. The error introduced by this "implicit assumption" depends on the atmospheric profile and is hidden deep in this module. It is small for the current atmospheres, but it is not guaranteed that it will remain small for some unusual temperature profile.
**Reply:** That particular aspect is not an issue because the atmosphere is in the optically thin regime at those altitudes and the parameterization is very accurate there. Look at all the tests performed and see how the errors tend to vanish at high altitudes.

Lines 437-462: this is another potential candidate for a general comment. The problem is that there are three values of this rate coefficient, k(CO2-O): the first one is measured in the lab and is about 1.5−2.0 x 1e-12 cm3 s-1 (e.g. Castle et al., 2012), the second one is about 6.0-9.0 x 1e-12 cm3 s-1 retrieved from space-borne observations of 15-micron CO2 emissions, and the third one, 3.0 x 1e-12 cm3 s-1 was suggested for using in the GCMs. This paradox has not been resolved over the decades of its study, but it shouldn't matter for the parameterization itself, because it is supposed to calculate the cooling rate with any given k(CO2-O).
**Reply:** Exactly, we fully agree with the referee. The point we want to stress is that the parameterization has been optimized for k~6.0e-12 cm3 s-1, and we would like to assess the errors when it is used for half of this value. This is important to tell the reader that if using a rate half of the value for which it has been optimized, it does not lose significant accuracy.

Line 455: how do you explain that the errors obtained for a smaller k(CO2-O) are larger? Usually, the larger the quenching rate, the larger the cooling rate, and the magnitude of this error is linked to cooling, as in the case of enhanced CO2.

**Reply:** The very likely reason is that the coefficient has been calculated, and it is therefore optimized, for the larger k(CO2-O) rate. In any case, the differences are very small.

Line 462: Fig. 20 is barely readable. Please, provide only a couple of profiles coming from a different source (see general comment 4) and show the errors for the corresponding cooling rate profile on the right-hand side panel.

**Reply:** The intention of showing this figure is not to distinguish between the different pT profiles but to show that the parameterization has been tested for an ample range of temperature structures, ranging from polar-summer-like profiles to polar-winter-like profiles and for elevated p-T profiles. That is, it is a very demanding test.
We have shown above a few profiles extracted from Fig. 20. We will show a few pT profiles like those in the Figures above together with the cooling rates estimated by the "exact" model and by the parameterization. We will make clear that they are examples but should not be taken as representative of the performance of the parameterization.

References
García-Comas, M., Funke, B., López-Puertas, M., Glatthor, N., Grabowski, U., Kellmann, S., Kiefer, M., Linden, A., Martínez-Mondéjar, B., Stiller, G. P., & von Clarmann, T. (2023). Version 8 IMK–IAA MIPAS temperatures from 12–15 μm spectra: Middle and Upper Atmosphere modes.*Atmospheric Measurement Techniques*, *16*(21), 5357–5386. https://doi.org/10.5194/amt-16-5357-2023

---

## Author Comment (AC4)

**Response to Reviewer 2**

We would like to thank the referee for the helpful review and the constructive comments. His/her comments are given below in black and our responses in blue.

**General comments:**

As the developer of an LTE radiation scheme used in a GCM, my review is from the point of view of a potential user of the code who would want to improve representation of the upper atmosphere in their code. There is certainly a strong case for a revised non-LTE parameterisation that can handle much larger concentrations of CO2, and therefore the contribution from this paper is welcome. But as I am not an expert on non-LTE effects I can't comment on the scientific details of this particular parameterisation. Naturally I would expect the code to be available at the time of submission to GMD and I expect the authors to rectify this.
**Reply:** We appreciate your valuable suggestions even if not being an expert on non-LTE. The referee is fully correct about the availability of the code. We were not aware of the need for its availability during the review process and thought of providing it during the review process. We will provide it in its final version as a Fortran 90 code with the revised version of the manuscript. In the meantime, the parameterization is now available provisionally as a Python routine, please see the reply to the Editor comment 1 (https://zenodo.org/doi/10.5281/zenodo.10547026).

The algorithm is presented as a complete radiation code, but the LTE part described in section 5.1 is quite crude for tropospheric radiative transfer since it doesn't represent scattering or cloud overlap and heterogeneity. I imagine that most GCM developers (like myself) would want to use their existing radiation scheme in place of the algorithm described in section 5.1 for the LTE region from 0 to 70 km, and then to add on NLTE regions 1-4 described in section 5.2-5.4. It would therefore be helpful to describe how to do this:
**Reply:** We did not make it clear in the text. It is not intended to use this radiation scheme in the LTE region, at least not below 50 km. It was not our purpose to develop the parameterization for that region. Note that it includes only CO2 and hence it is not designed to be used in the LTE region where other species contribute to the cooling/heating.
We will clarify this in the text and the documentation of the code.
The suggestion is that users use their own LTE radiation scheme at low altitudes, in the LTE region and then switch to the non-LTE region with the cooling rates provided by this parameterization. A safe switching region could be between 50 and 60 km. Still, this parameterization can be used with a reasonable accuracy down to the lower stratosphere, although it should be kept in mind that accounts only for CO2 cooling/heating.

Can I simply replace the cooling rates from my code with the cooling rates from the non-LTE parameterisation above 70 km.
**Reply:** Exactly, or from even above 50 km. A switching region is advisable to be used.

Do I need to provide any upwelling fluxes at 70 km to feed the treatment of the regions above?
**Reply:** No, the parameterization calculates internally the entire cooling rate profile (with its 'own' upwelling flux from the LTE region). However, the input file (with pressure, CO2, T, etc. ) should cover the entire atmospheric range, e.g., from the surface up to at least $x=log(1e3/p(hPa))=13.5$, i.e. $p=1e-3$ hPa (or ~92 km) in order to properly calculate internally the upwelling flux. If some layers at the bottom of the atmosphere are omitted, the parameterization still works but the cooling rates are not fully correct. Hence that is not recommended.
A surface temperature is also an option to be imputed. If not given, it will be taken as the value of the lower altitude of the input file.

*Again, all of this is already documented in the Readme file of the version in the repository.*

The description of how one region is coupled to the one above is not clearly described in the text.
**Reply:** *That's correct. We are giving recommendations above the altitude range for the use of the parameterization, as mentioned above. But the user only needs to consider two regions: the lower one, in LTE, and then this parameterization on the region above.*

Is the vertical resolution of the parameterisation fixed or can I run it on my own vertical grid?
**Reply:** *The user can provide its own vertical grid in a specified input file which should contain the pressure (hPa), temperature (K), VMRs of $CO_2$, O, $O_2$, and $N_2$. Then, the parameterization interpolates those parameters onto its internal grid (described in the manuscript). This is detailed now in the readme file of the code.*

Do I need to simulate all regions or can I omit the uppermost 1, 2 or 3 regions if I am not interested in modelling temperatures above a particular height?
**Reply:** *Again this is an interesting question. In principle, the users should input the parameterization from the surface up to the upper boundary of interest. This upper boundary cannot be lower than ~92 km (1e-3 mb). However, it is recommended to extend the upper limit up to ~120 km since exchange or radiation takes place between the upper mesosphere and the lower thermosphere.*

Has the parameterisation been coded in Fortran or C (needed for a GCM) or in an interpreted language like Python? What is its computational cost? How large is the dataset that needs to be read in (e.g. the various look-up tables that are used to populate the Curtis matrices)?
**Reply:** *The current (provisional) version in the repository is in Python but we will make it available in Fortran with the revised version of the manuscript.*
*That version of Python is very slow. We will provide the computational cost when translated to Fortran.*
*There will not be an external dataset for reading. The Fortran routine will contain all the needed coefficients as parameter variables.*

SPECIFIC COMMENTS
1. Abstract: it would help to mention the magnitude of LTE heating rate errors (i.e. LTE minus non-LTE), in order to put into perspective the magnitude of the errors in the parameterisation of non-LTE effects (i.e. parameterised non-LTE minus accurate non-LTE).
**Reply:** *We agree. They very much depend on the altitude but in general they are way smaller. We will include a general statement about these differences.*

2. There is an excessive number of figures, and they are sometimes referenced out of order. My suggestion is to select one (or at most two) of the test atmospheres and then when you have a 6-panel figure with a separate panel for each atmosphere, reduce it to just one panel in the main body of the text, and then combine figures together when it is useful to have panels next to each other. For example, combine Figs. 6a and 7 into a 2-panel figure, since they show the same thing for the same atmosphere, just on a different scale. Then put the other five atmospheres in Supplementary Material (not in appendices) and refer to them sparingly. I suggest that Fig. 1 is removed as it is not needed, and Fig. 2 is replaced by Fig. 11 (the latter shows the same as the former but with more information). There are plenty of other opportunities for the authors to cut down the number of figures.
**Reply:** *We agree. The other referee has made a similar comment. The reason for showing the results for the six p-T profiles is that frequently, the effects of the discussed parameter depend very much on the temperature profile. Nevertheless, we take the suggestion. We will make a drastic reduction in the number of figures/panels in the main text. They will be reduced from 28*

to 19 (and some of them with a reduced number of panels); and the number of Figs. in the appendix will be reduced from 20 to 8. 21 figures will be moved to a Supplement.

Please see the response to the first general point of the other referee for all the detailed actions taken regarding this point. Further, Figs. 2 and 11 are not merged because, if we do, we will not see the differences in the temperatures in the mesosphere and lower thermosphere, precisely the region where the major difficulty of the parameterization resides. About "and refer to them sparingly", we will try to make the reading smooth but in general we prefer to point the reader to the figure in question so he/she can better understand the argument. He/she always cannot look at the figures if unnecessary.

3. Lines 102, 200 and elsewhere - please refer to profiles #3 and #6 as "present day" and "4x pre-industrial" so the reader doesn't need to flip back to remind themselves what these numbers refer to, and similarly for the other hash-numbered profiles. Also, Fig. 9 could state the factor multiplying the pre-industrial figure surface value for each point somewhere on the figure.
**Reply:** Definitely, we will do that. It will make the reading much easier. About Fig. 9, it will be moved to the Appendix but we will include in the caption the factor multiplying the pre-industrial figure surface value for each point and will make a reference to Fig. 2 (the CO2 vmr profiles).

4. Line 219: "pointing" -> "role in determining"?
**Reply:** Thank you. That was a leftover of a previous version. It will be corrected.

5. The equations are largely taken from Fomichev et al. (1998), but it would be useful to improve clarity in several places. For example, Equation 1 is simply a matrix-vector multiplication - why not present it as such? The equations for "a" and "b" on lines 264 and 265 are presented as if they define the element the Curtis matrix (via the equation on line 261), but they themselves contains the elements of the Curtis matrix on the right hand side as a scaling for the terms involving the band strengths. So one is left wondering how the actual value of the elements of the Curtis matrix are determined.
**Reply:** "Equation 1 is simply a matrix-vector multiplication - why not present it as such? " Yes, writing it in a matrix formulation will be more concise (and clean), but we prefer to write it explicitly, particularly because the summation goes over different altitude ranges and it is easily written in this form. If using the matrix notation we would need to define different matrices (symbols) for the different altitude ranges.

The Curtis matrix is actually embedded in $\mathcal{A}_{i,j}^t(\nu)$ , see lines 253-254.
**a** and **b** are scaled Curtis matrices.

6. Line 404: remind the reader that by "both" parameterisations you mean Fomichev's original one, and the one in this paper.
**Reply:** Sure. We will do that.

---

## Author Comment (AC5)

**Response to the Community comment by Alexander Kutepov.**

We are responding below to the comments by our colleague. His comments are listed in black with our responses in blue.

**GENERAL COMMENTS**
This is a large manuscript with very large numbers of plots both in main body and appendixes. Although it is well written and structured, many tests of a new routine repeat one another and, therefore, look excessive.
**Reply**. The point is accepted. The other two referees made similar comments. The number of figures in the main text and the Appendix will be substantially reduced (see reply to referee 1 for the details). The number of Figs. in the main text will be reduced from 28 to 19 (and some of them with a reduced number of panels); and the number of Figs. in the appendix will be reduced from 20 to 8. 21 figures will be moved to a Supplement.
Nevertheless, we would like to state clearly in the manuscript the goodness and limitations of the parameterization; hence all the tests will be retained but the corresponding figures moved to a Supplement.

The work takes us back to the late 70s - early 80s of the last century, when new for that time techniques of the approximate 15-micron CO2 cooling calculations (both for LTE and non-LTE conditions) were developed, see (Kutepov, 1978) and (Akmaev and Shved, 1982). The revised version of the Fomichev et al, 1998 (hereafter F98) routine described in the manuscript does not represent any innovation and does not suggest any new option for the GCM users and developers.
**Reply**. We recognize in the manuscript that this is based on the same approach of Fomichev et al, 1998. Our major aim was to extend it to higher CO2 vmrs and once we did it, to improve it also in other aspects, as detailed in the manuscript. We think it is a new option, as GCM models can now be run for higher CO2 vmrs and it is more accurate.

This statement may be explained as follows. Any physical parameterization for GCM must be able to react as realistic as possible at steady changing local physical state of the atmosphere in the modeling process. This is particularly true for the infra-red radiation and its effects (local cooling or heating), which are critically important for adequate modeling of energy balance. It is well known that instantaneous p-T distributions in modern GCMs of middle and upper atmosphere exhibit very strong variability, caused by the superposition of tidal and gravity waves of different amplitude and vertical scales. Same is true for the atmospheric p-T distributions measured in both ground and space experiments. Therefore, the parameterization of the 15-micron CO2 cooling, which is a main radiative cooling mechanism of these layers, must properly react to this variability.

However, the matrix parameterizations of the 15-micron cooling are unable to provide adequate reaction to strongly disturbed T distributions. This was well known already 30 years ago for those (I was among them) who worked on developing the first version of the F98 parameterization. Demonstration of large parameterization errors for wavy T profiles was not included in the paper by Fomichev et al, 1993, however, it contained at least the warning addressed to its users ***"It is recommended for the calculation of the radiative cooling for smooth temperature profiles, namely for profiles undisturbed by micro- and meso-scale motions."*** When the updated 1993 routine was released by Fomichev et al, 1998 nothing changed, the revised routine was still unable to treat wavy distributions, see (Kutepov and Feofilov, 2023, hereafter KF23). Again, as in the 1993 paper, the accuracy of F98 was demonstrated exceptionally for smooth T profiles. However, the authors dropped from the paper text the warning for its users cited above. Since that time the F98 routine has been widely used in GCMs of middle and upper atmosphere for any T distributions.

It is also worth noting here that 15-micron CO2 radiative cooling is strongly non-linear in respect to the temperature variations. Therefore, the zonal mean cooling, which is enthusiastically discussed in this manuscript, is not equal to that calculated for the zonal mean temperature. The authors, however, pay no attention to this fact!

**Reply**. Precisely for the reasons mentioned above, we tested the parameterizations with realistic pT profiles as those derived from MIPAS. Rather than giving a recommendation we assess its performance and give errors so the users are aware of them. One important point: The individual pT profiles do show a larger vertical structure because they are affected by the instrument noise. They are not smoothed profiles. For that reason, we wanted to show Fig. 20. We have included a couple of Figs. in the Reply to Referee 1 where we can see the large vertical variability of the tested pT profiles.

Let us clarify also the point about "zonal mean cooling" and the zonal mean temperature. We did NOT calculate the cooling for the zonal mean temperatures. We calculate the cooling for each individual p-T profile. Only when showing the differences between the "exact line-by-line" model cooling rates and those from the parameterization they were zonally averaged. For that reason, we also show the RMS. As Referee 1 also pointed out, we agree that we should not consider the mean as the error of the parameterisation. We will still provide the mean values (bias), which will inform us about its accuracy in a global sense (eg the cooling at high altitudes in the previous version is always about 2K/Day larger than in the line-by-line calculations), and by the RMS (not the standard deviation), as an appropriate estimate of its error for individual profiles. Those RMS values will be brought up in the abstract.

Also, note the study was done altitude by altitude, not averaging over altitudes. Of course, the parameterization is not perfect and one would wish to run the full non-LTE model for each p-T profile in each grid of the model for each run. Here we offer a fast parameterisation with reasonable (well-assessed "errors"). Let us the modeller choose what they prefer.

Now back to the current manuscript. The authors invested large efforts to update the F98 routine. They declare that the revised routine (hereafter L-P.23) allows calculations of 15-mucron cooling with higher accuracy for current CO2 vmr. It calculates with reasonable accuracy also the cooling for much higher CO2 concentrations. Nevertheless, again the accuracy of L-P.23 is demonstrated only for very smoothed "standard" T distributions.

**Reply**. We have shown both, the accuracy for the reference (smooth) atmospheres and also for the MIPAS pT atmospheres, which show many "wavy" profiles and very uncommon nearly isothermal profiles (see previous point and the Figs included in the reply to Referee 1).

As suggested by Referee 1, we will include also tests for lidar pT profiles.

Meanwhile, the authors are aware about large errors F98 routine has for disturbed T profiles. D. Marsh was the co-investigator of a recent NASA grant (Kutepov, 2021), where I was the PI. He and his team received funding for testing the KF23 algorithm for calculating 15-micron non-LTE cooling, comparisons of this routine with F98 parameterization, and for installing KF23 routine in the WACCM model. This study showed that F98 causes very large cooling errors (up to 25 K/day) on wavy profiles. These errors are discussed in KF23.

**Reply**. We agree that for very wavy individual profiles, the cooling rates might differ significantly from the "accurate" calculations in the upper mesosphere. Fig. 22 gives us the error when averaging over narrow latitude bands (remember they are the mean of the differences of the "accurate" and "parameterized" cooling rates (not the differences of the cooling rates calculated for the mean temperatures), showing that they could be significant (up to 2-3 K/day) as discussed in the text; and Fig. 24 shows the RMS, which near the mesopause they are very significant. Thus we are properly assessing the parameterization accuracy. As suggested by Referee 1, these values will be brought up in the abstract.

One point to be made is that there might be room for multiple non-LTE parameterizations and, for climate simulations, the errors introduced due to small-scale temperature variability may well be tolerated by modelling groups if it comes at a reduced numerical cost. The KF23

parameterization adds significant numerical cost in the running of a GCMs and therefore may not be suitable for centennial-scale ensemble simulations. In limited testing by the software engineer at NCAR it was not insignificant and the option to call it only every other timestep needed to be introduced to reduce the cost. We should point out that KF2023 is about 300 times slower than F98.

Meanwhile, the authors of the manuscript are quite honest when, describing the main motivation of their study, they write: ***"In our case we have the option of developing a completely new parameterization, to adapt other CO2 parameterizations (as those cited above), or to extent and improve the parameterization of Fomichev et al. (1998). Attending mainly to practical reasons of promptness, we opted for the later."*** The keyword here is "***promptness".***

This "***promptness"*** looks somewhat strange after 25 years of no interest of the authors to the problem of fast and accurate calculation of radiative cooling for the Earth's GCMs. Knowing about the drawbacks of F98 routine and other matrix parameterizations we spent these years developing accurate non-LTE radiative transfer techniques, which are free of these drawbacks and are fast enough to be applied in GCMs. The results of this long-term study are summarized in KF23.

**Reply**. The recent interest in improving/extending the parameterization, as you have done, is that a) climate simulations conducted for CMIP have only just begun to include models that resolve the stratosphere and mesosphere, and b) the standard climate sensitivity metric includes a 4xCO2 experiment which as incorporated into the DECK for CMIP6. This work paves the way for moving the F98 code to modern Fortran or C++. It is unlikely that, as modellers increase the model height of their model top for climate simulations, they would want to incorporate code that relies on the compatibility of compilers to 30+-year-old coding standards or does not cover the full CO2 range of the DECK.

Finally, what kind of new product these authors developed with the main motivation "to be prompt"?

**Reply**. To make available a fast and efficient algorithm capable of coping with the very large CO2 VMRs. Some GCM modellers urged us to extend it to large CO2 vmrs.

Again, I do not need my own judgement. It is enough to cite what the manuscript authors write when they describe large errors, much lagers than 0.5-1.0 K/day reported for standard profiles, which they observed for all profiles at latitudes northernmost of 50 N for a single non-standard situation with a pronounced elevated stratopause event: "***Both parameterizations underestimate the cooling in that atmospheric region. The new parameterization has, however, a better performance above about 80 km, but in the strat-warm/elevated stratopause region (80–100 km) it still underestimates the cooling by 3–7 K Day−1 (~10%)".***

**Reply**.  We do not understand this kind of comment. We just did what we scientists are supposed to do, e.g., to recognize the limitations of the parameterization. These elevated stratopause situations are the most difficult situations to handle with this parameterization, and this one of 2009 in particular, which was very strong. However, globally, they are rather unusual and limited to certain regions and times. In any case, the parameterization, although not perfect, calculates the cooling rates with errors of ~10%.

This looks like a confession that in a non-standard situation the new l-P.23 routine works no better than F98.

**Reply**.  It is better in about 2 K/Day above 5e-4 hPa and also around 4e-3 hPa (see Fig. 27 left panel); and true, it is a very unfavourable situation.

And further: *"It seems clear that part of this underestimation is caused by the fact that such atypical temperature profiles (see Sec. 3.1) were not considered in the parameterization. However, its inclusion would not solve the problem as in the calculations of the coefficients a trade-off of the weighting of the different p -T reference atmospheres have to be chosen (see Secs. 5.1 and 5.2). Thus, it might ameliorate the inaccuracy for these elevated-stratopause events but would worsen the accuracy for other general situations. This manifests the difficulty/limitation of this method to provide accurate non-LTE cooling rates for all temperature structures (gradients) that we might find in the real atmosphere."*

 I absolutely agree with this statement: this approach for parametrizing the 15-micron non-LTE cooling in the middle and upper atmospheric layers, which was applied in previous routine versions in 1993, in F98, and repeated in L-P.23 **is a deadened approach**.
**Reply**. We do not agree with this final statement. The fact that the parameterization is not very accurate (errors of ~10%) for certain atypical and unusual conditions should not be generalized for all conditions. As very often occurs, it is a trade-off between accuracy and efficiency.

In KF23 we discuss in detail its drawbacks. Briefly: it is impossible to adequately estimate the non-LTE cooling in the very variable atmosphere by dividing it in several altitude regions, where different techniques or expressions for cooling calculation (although linked in various ways) are used. Only exact algorithms, which rigorously describe the radiative energy exchange between various altitude layers and the non-LTE radiative field coupling with atmospheric heat reservoir may satisfy the current cooling accuracy requirements.
**Reply**. We think it depends on what we understand by "adequate estimate". We are giving conservative error estimates so the users can decide if it is (or not) adequate for his/her cooling rate accuracy requirements. We offer an option, not stating that this option is the best.

*It is my opinion that this paper in its current shape must not be published. It does not provide any significant improvement compared to previous work(s)*.
**Reply**. We have no comment on the first part of the sentence. We will accept the editor's decision. Regarding the second sentence, we think it is already clear in the manuscript the advantages/improvements of the parameterization.

To be published the manuscript requires significant major revision:
(1) it must demonstrate how revised routine works for T profiles disturbed by the strong waves.
**Reply**. We will show a few examples of the MIPAS pT profiles, which are rather "wavy" (see, e.g. the figures in the reply to Referee 1, but will also make clear that they cannot be considered as representative of the parameterization performance. We think that a better estimate of its accuracy is given by the RMS (not the standard deviation) obtained from a statistically significant sample. In any case, referee 1 suggested showing the results for some kind of "lidar" pT profiles and they will be included in the revised version.

If the routine fails on the wavy profiles, but the authors still recommend it for further usage in GCMs, then (2) they need to justify that these errors have negligible or no effects on the GCM model results.
**Reply**. We think that it corresponds to the GCM modellers to decide if the estimated accuracy, given in Figs. 22 and 23, fulfil or not their accuracy requirements.

**SPECIFIC COMMENTS**
**1. Why exact methods of the non-LTE CO2 cooling calculation cannot be applied in GCM?**
The manuscript authors write: *The computation of the cooling under those non-LTE conditions requires the solution of the radiative transfer equation (RTE) which is a non-local problem and requires a large amount of CPU time. Therefore, the solving the RTE in general circulation models (GCMs) or climate models that extend in height above the*

*stratopause is impractical and efficient parameterizations of the CO2 infrared cooling have been developed and implemented in such models.*

I disagree with this statement. It does not matter whether the LTE approach is applied or the non- LTE problem is considered, in both cases the calculation of cooling requires the solution of RTE, and is, therefore, the non-local problem.

Moreover, the computing costs of exact RTE solution in modern algorithms are not the main problem, which makes *"the solving the RTE in general circulation models GCMs impractical".* The authors mean here not just RTE but the entire non-LTE problem.

**Reply**. You are correct, even in LTE, it is still a non-local problem. About the second sentence, you are correct again, we meant the entire non-LTE problem. We will change the text accordingly.

Inversion of large matrices to get the populations in the developed in 1950s CM (Curtis, 1956) matrix algorithm, which the members of this team utilize since 1980s, is most computational costly part of the non-LTE problem solution. The authors either are not aware of or ignore dramatic progress in the developing the non-LTE techniques, see, for instance, (Hubeny and Mihalas, 2015) and (Frish, 2021). Large matrix inversion costs make CM technique usage impractical in GCMs. We discussed this in the papers by Kutepov et al, 1998, Gusev and Kutepov, 2003, and in more details in KF23.

**Reply**. The original algorithm was developed based on the Curtis matrix (CM) approach and so we used it here. Nevertheless, we recall that the parameterization does not invert CMs. Coefficients are computed based on pre-calculated CMs but these are not inverted within the parameterization. On the other hand, in the revised manuscript, we will cite the recent fast non-LTE model of KF23 that uses another non-LTE technique.

**2. How the 15-micron cooling maximum in the lower thermosphere is formed?**

In the manuscript: Above the mesopause, the cooling rate rapidly increases following the enhancement of the kinetic temperature. Above about 130 km, the cooling rates decline because of the depletion of the CO2 vmr (see Fig. 7).

Are the authors sure about this? I recall that in early publications of the 1980s about the 15-micron cooling it was demonstrated that cooling declines at higher altitudes even for constant CO2 vmr. Main problem at these altitudes is the rapid decrease of atmospheric pressure and the CO(v2) collisional quenching rate, which disconnects the 15-micron radiation from the atmospheric heat reservoir. If the authors do not agree with my comment, could they demonstrate that cooling stops decaying above 130 km when CO2 vmr is constant?

**Reply**. We should have said here "mainly" due to the CO2 vmr fall. As you well know, in this region we can assume the cool-to-space approximation, in which (see, e.g. Eq. 9.1 in Lopez-Puertas and Taylor (2001)) the cooling rate is proportional (when expressed in K/Day) to the CO2 vmr, to the [O] concentration (density) and to temperature through exp(-E/kT). As the altitude increases, the CO2 VMR decreases, and so does [O] (density) but T increases. So what we see at the end is the total effect of the three quantities. We will correct this in the revised version.

**3. Why the CO2-O quenching rate coefficient ~ 6.0e-12 cm3 s −1 was used as the basic one for revision of the F98 parameterization?**

The authors write that they used for updating the F98 parameterization the quenching rate coefficients which are very similar to those used for its development *"... except the k CO2−O rate (process 1c in Table 1) that has been considered here with its upper limit. That is, about a factor of two larger than in the parameterization of Fomichev et al. (1998). This rate coefficient is not well known with uncertainties of the order of a factor of two (see, e.g., García-Comas et al., 2008). While laboratory measurements are in the range of 1.5 to 2·10−12 cm3 s −1 the values derived from atmospheric observations are close to 6·10−12 cm3 s −1. ...."*

First, measured in laboratory and retrieved from the atmospheric observation values of k differ not by a factor 2, but by 3-4. Additionally, if one accounts for the k retrievals by Feofilov et al, 2012, which involved the ground-based lidar temperature measurements, then this factor will be 3-6.

**Reply**. That is correct, taking the entire variation (equivalent to 3 sigma), we have roughly a change from 1.5 to 6, e.g. a factor of 4, which would be +/- a factor of 2 for a 3-sigma error. We believe the higher range derived by Feofilov et al, 2012 also accounts for the errors in the measured temperatures and an estimated uncertainty in [O], which was not measured; hence its larger uncertainty.

Table 1 shows that the authors selected k ~ 6.0e-12 cm3 s −1 for updating the parameterization. Although k is supposed to be an input parameter in both F98 and L-P.23 routines, however, the previous one was optimized in the transition region from LTE to non-LTE for k ~ 3.0e-12 cm3 s −1, whereas new one is optimized for 6.0e-12 cm3 s −1. The authors tell in manuscript that this causes additional differences between F98 and L-P.23 in the transition area even when both apply the same rate coefficient, and then explain the reason why they selected higher rate for optimizing L-P.23 as ***"... we have optimized it for the high value (see Table 1), as this value has been used in the most recent non-LTE retrievals of temperature from SABER and MIPAS measurements"***.

It is not enough, however, to say higher k was selected because this rate provides more reasonable T retrievals from space observation. As the authors know to validate these T measurements current GCMs apply twice lower rate 3.0e-12 cm3 s −1. If the authors recommend, which follows from the text, to use L-P.23 with the most high k, then can they demonstrate how this affects the GCM (for instance the WACCM model) runs compared to those with twice lower k? Does this provide better fitting of measured temperatures?

**Reply**. Of course, it is questionable which value of K to use to optimize the parameterization (e.g. ~3e-12 or ~6e-12, as the lower laboratory values (~1.5e-12) do not fit well with atmospheric measurements). We could have chosen 3e-12 instead of 6e-12. But we did not, as explained above and in the manuscript, because the two major temperature databases of the upper mesosphere/lower thermosphere, SABER and MIPAS, (covering several years, more than 20 years in the case of SABER) both use the larger rate. And the teams responsible for those retrievals made that decision based on the temperature validation performed for both datasets E.g., García-Comas et al (2023) have shown that they obtain a better agreement with temperature measurements of independent "non-LTE" free instruments when this large rate is used. You mentioned the work of Feofilov et al, 2012, who also retrieved from SABER and lidars temperature, and they obtained a k value of 6.5e-12, even larger than that used here.

The argument of using a lower value of ~3e-12 because that is being used in GCM models to reproduce the SABER and MIPAS T measurements we think is less substantiated and weaker. First, it is not clear that the models can reproduce the measured temperature field with that rate, see, e.g. Fig. 1 in Smith (2012). Secondly, the temperature computed by GCMs depends not only on this rate but also on many other factors like the parameterisation of the GWs, the chemistry (related mainly to O3 and O), vertical descent, etc. Further, they should also explain not only the temperature fields but also, for example, the CO2 and CO observations. A good example of the difficulty of e.g. WACCM in simulating those measured temperature fields can be found in this recent work:

https://essopenarchive.org/users/568957/articles/657910-a-novel-gravity-wave-transport-parametrization-for-global-chemistry-climate-models-description-and-validation.

In any case, the errors incurred by the parameterization if using the intermediate K rate are not that much larger. For this reason, we performed such an assessment.

**4. Testing the parameterization for measured temperatures**

The Fig.21 of manuscript shows an example of the MIPAS nighttime temperature profiles (15 February 2009) used for verifying the parameterization accuracy. The authors note large

variability of the measured temperature profiles. These individual profiles are the good inputs for the revised parameterization to show how it works for strongly disturbed T profiles. The authors have obviously performed these tests, but they do not show these results. Instead, they write ***"The results are presented in Fig. 22 for the zonal mean of the differences for two days of solstice and two days of equinox conditions and in Fig. 23 as the global mean difference for all latitudes for each of the four individual days."*** Why only these mean values are shown? Obviously, the averaging smashes the errors obtained for individual profiles, for which we observed in our study of F98 parameterization presented in KF23 errors up to 25 K/day. Meanwhile these large errors in our study have generally concentrated in the altitude region around 90 km, exactly where the RMSs in Fig.25 of manuscript are maximized reaching 8-9 K/day. In our paper we explain why F98 works badly in this transition region. L-P.23 has the same problems and is nothing better.

**Reply**. We think we do appropriately present the results. On one hand, by the zonal mean of the differences as shown in Fig. 22, where the cooling rate differences are averaged over small latitude boxes (5º). And also by using the mean of the differences (to give an idea of "global" biases) and the RMSs. This is the standard procedure followed in the many validation studies we find in the literature when the sample is statistically significant.

**5. The accuracy of L-P.23 for smooth T profiles**
I wrote above that L-B.23 will not be any better than the F98 for disturbed T profiles. But how about the smoothed standard T profiles? The accuracy of cooling calculations with L-P.23 less than 0.5 K/Day for preindustrial CO2 for smooth profiles is also questionable.

(1) The non-LTE model, which is used in this study for the reference calculations to optimize L-P.23 and then to check its accuracy, includes only 18 15-micron bands. From my point of view this model itself is not accurate. In the routine we suggest in KF23, which utilizes the exact solution of the non-LTE problem, we use the same bands to calculate nighttime cooling for 400 ppm of CO2 with an error not higher than 1 K/day for any T profile, including disturbed by strong waves. For smooth profiles this error reduces to ~ 0.2 K/Day. These errors were estimated by comparing KF23 routine with our reference model, which comprises 60 vibration levels of 5 CO2 isotopic species and hundreds of bands. So, the exact algorithm compared to the exact algorithm showed 0.2 K/day error when a reduced set of bands was used. I doubt that the error of 0.5 K/day the authors report for the L-P.23 routine for smooth profiles after comparing it with a very simplified "reference model" is true. It must be higher.

**Reply**. Our reference non-LTE model is not simplified at all. We just dropped the contributions of bands beyond the listed 16 bands because we tested that their contribution in the non-LTE region is negligible (below 0.1 K/Day). Some of the not-included very weak bands may have some very small contributions in the LTE region, e.g. around the stratopause and below, but those contributions are not significant for the non-LTE region.

(2) The authors write that contribution of the heating due to the absorption of solar radiation in the near infrared CO2 bands at day time is negligible compared to the 15-micron cooling. However, 2-3 K/day (for current CO2) do not look negligible for the routine, which accuracy is declared to be about ~0.5 K/day. We tested in detail the (Ogibalov and Fomichev, 2003) parameterization of this heating, found this warming increasing for some wavy T profiles, and made sure this parameterization cannot guarantee the error of the KF23 routine at daytime to be lower than 1 K/day. As a result, we extended our KF23 daytime model up to 26 CO2 vibrational levels and 56 bands to satisfy this requirement.

**Reply**. It seems there is a misunderstanding here. We are NOT neglecting the cooling by the CO2 15 μm bands during the daytime. It is just that their contributions during the daytime are accounted for by the NIR heating routine. Hence, if included in this parameterization it would be included twice.
Just one precision. Our day/night differences of the CO2 15 μm cooling rates are smaller than 1K/day for all pT profiles, at any altitude and for CO2 vmrs up to 5 times the pre-industrial

value, except for MLS and SAS pT profiles near 105 km for CO2 vmr 4x and 5x the pre-industrial values.

(3) The authors say nothing about how they account for the cooling effect of the micro-scale sub-grid T disturbances. Kutepov et al., 2007 and Kutepov et al., 2013 showed that these temperature fluctuations cause near the mesopause an additional cooling up to 3 K day-1. I draw the authors' attention to the results shown by Kutepov et al, 2013 in Fig.1.5. It demonstrates one of the runs of the Leibniz Institute Middle Atmosphere (LIMA) model with the 15-micron cooling modified to account for the sub-grid T disturbances. It is shown that very minor variation of cooling (not higher than 2-3 K/day) lead to significant changes of the monthly and zonal mean temperatures for July 2005.

I am sure that errors of L-P.23 routine are much higher than 2-3 K/day and can be hardly reduced due to the deficiencies of the methodology applied. These errors will obviously have a strong impact on the GCM results.

**Reply**. Effectively, we do not include (e.g. do not provide a routine) the cooling rates induced by thermal structure at a grid smaller than the input grid (or, more properly, than the internal grid of the parameterisation). To properly account for them it would be necessary to know (or make assumptions) about how the temperature varies between grid points. We assume that the cooling induced by non-resolved GWs, propagating with a vertical wavelength of the order of/or smaller than the parameterisation grid, would be taken into account in the GCMs by using an appropriate GW parameterisation (eg. see Intro of Kutepov et al, 2013). Currently, some new parameterizations are being developed to account for these effects (see, https://essopenarchive.org/users/568957/articles/657910-a-novel-gravity-wave-transport-parametrization-for-global-chemistry-climate-models-description-and-validation).

**6. The code availability**

It seems the manuscript was submitted as the GMD "Development and technical paper". If it is correct, then **"The code should be made available, and a model availability paragraph must be included"**. The code is, however, not available.

Once the code is available, I will demonstrate that its errors are much higher than reported in the manuscript.

**Reply:** We were not aware of that and thought of providing it during the review process. We will provide it in its final version as a Fortran 90 code jointly with the revised version of the manuscript. In the meantime, the parameterization is now available provisionally as a Python routine, please see the reply to the Editor comment 1 (https://zenodo.org/doi/10.5281/zenodo.10547026).

That version of Python is very slow. We will provide the computational cost when translated into Fortran.

A model availability paragraph will be included in the section about the code availability (see Reply to the Editor-in-Chief).

**References**

García-Comas, M., Funke, B., López-Puertas, M., Glatthor, N., Grabowski, U., Kellmann, S., Kiefer, M., Linden, A., Martínez-Mondéjar, B., Stiller, G. P., & von Clarmann, T. (2023). Version 8 IMK–IAA MIPAS temperatures from 12–15-µm spectra: Middle and Upper Atmosphere modes. *Atmospheric Measurement Techniques*, *16*(21), 5357–5386. https://doi.org/10.5194/amt-16-5357-2023

Smith, A. K. (2012). Global Dynamics of the MLT. *Surveys in Geophysics*, *33*(6), 1177–1230. https://doi.org/10.1007/s10712-012-9196-9

https://essopenarchive.org/users/568957/articles/657910-a-novel-gravity-wave-transport-parametrization-for-global-chemistry-climate-models-description-and-validation).

---

## Author Response (AR1)

**Dear Editor,**

We are including below the responses to the referees and to the colleagues who commented on the manuscript.

We have taken longer than expected because we have performed another set of tests for new temperature profiles (see below). Referee 1 suggested to perform a study for lidar measurements. However, we have made it for the temperature calculated by a high-resolution version of WACMM-X. As we show below, these temperatures show a very similar vertical structure to the lidar measurements and, in addition, they cover a larger altitude range and supply the O, O2 and N2 vmr profiles. Further, this set of tests has the additional advantage of applying the parameterization just to the kind of models for which it has been mainly devised. We hope the referee agrees with us.

The second reason for the delay was that the translation of the code into Fortran took a bit longer than expected.

We hope we have responded to the referees and colleagues adequately and that they will be satisfied with the revised version.

Best regards,
Manuel López Puertas on behalf of all co-authors.

**Response to Reviewer 1**

We would like to thank the referee for the helpful review and the constructive comments. His/her comments are given below in black and our responses in blue.

**General comments:**

1) In my opinion, the manuscript tries to cover too many topics despite the fact that the title states that this is an update to existing parameterization. Well, it doesn't hurt to recall the main milestones of the previous work, but the manuscript in its current form is "unfocused". If it is intended to be a text-book chapter on non-LTE in the CO2 bands, then the same information could be found in the first authors' book (Lopez-Puertas and Taylor, 2001). If it is supposed to be a technical paper then it should put more stress on technical aspects like the accuracy, interfacing, and so on. In addition, the text is overloaded with figures. I counted more than a hundred (!) of individual panels in the main part that consists of 41 pages. This is definitely an overkill and one can tell the same story in a much more concise way. I suggest the authors to shorten the manuscript and to keep only the essential parts. For example, all the inputs can be gathered on one two-panel plot, where the left-hand side would be occupied by the reference temperatures and the right-hand side would show the profiles of CO2 (only one is needed) and atomic oxygen, which is crucial for the estimate of the cooling rate. The first figure is not informative at all and can be easily omitted. The large number of panels in figures like Fig. 5 does not add much to the understanding of the principle of the radiative cooling or its calculation, and so on. Overall, I would reduce the number of non-essential figures to a minimum and put more efforts to a description of the routine itself, its implementation to a GCM and its limitations.

**Reply:** The point is well taken. We have taken the following actions.
- About the text, we have retained only the major points driving the CO2 coolings, e.g their dependence on the pT structure, on the CO2 and O vmrs, the different band's contributions and how these contributions and their deviation from LTE depend on the p-T and CO2 and O(3P) vmr profiles. These parts are essential to understand the difficulty and the rationale of the parameterization. Further, note that we are extending the CO2 vmr range of the original parameterization and this implies changes in the extensions of the LTE and non-LTE regions, e.g. in the boundaries of the previous version of the parameterization. We think these changes have to be justified and explained to the reader. The part of the paper which is somehow redundant with  Lopez-Puertas and Taylor (2001) is not that much, only part of Sect. 4.2 where we describe the reference non-LTE cooling rates. Here we focus more on the details of the input parameters (collisional rates, etc.) for the non-LTE model, which we think are important, for example, if compared to other parameterizations. Nevertheless, we have slightly reduced the text of Sect. 4.2.

- About the figures, following your suggestions, we have taken the following actions:
    a) To move all figures of the original Appendices to a supplement.
    b) To remove Fig. 1.
    c) To combine Figs. 2 and 3 in one 2-panel fig. The referee also proposes to include one Fig for O. So we have included a 3-panel 1-col fig. for p-T, CO2 and O.
    d) Fig. 5. Sorry but we do not agree with the referee on this point. This is an essential figure to discuss the points mentioned above as a reference for the line-by-line (reference) cooling rates. Nevertheless, it has been reduced to keep only 4 essential panels.
    e) Fig. 6. Contributions of the different bands. It has been reduced to only 4 panels (as Fig. 5).
    f) Fig. 7 Contributions of bands in the lower thermosphere (1 panel). It has been moved to Appendix B.

g) Figs. 8 and 9. Comparison of non-LTE and LTE cooling rates. Fig. 8 has been reduced to four panels but has not moved to the appendix because the other referee suggested that we should describe the non-LTE--LTE differences to show the relevance of the non-LTE errors of the parameterization. Fig. 9, however, has been moved to Appendix B.

h) Fig. 10. The effects of the K(CO2-O) collisional rate. We have reduced it to 4 panels, consistent with previous figures 5 and 6.

i) Fig. 12 (escape probability functions). It has been moved to the supplement.

j) Figs. 13, 14 and 15 compare the previous and current par. Figs. 13 and 14 are essential and have been kept in the main text. Fig. 15 has been moved to the supplement.

k) Figs. 16 and 17, and 18 and 19 show the accuracy of the parameterization for intermediate CO2 vmr profiles and for the K(CO2-O)/2. Figs. 16 and 18 have been moved to Appendix B. Figs. 17 and 19, the summary of the results, has been retained in the main text.

l) Figs. 20 and 21: MIPAS temperature profiles have been moved to Appendix B.

m) Fig. 22. Zonal mean of the differences between the previous and new par. for the MIPAS temperatures (8 panels). We have reduced it to only 4 panels: solstice (January) and equinox (March). The other panels have been moved to Appendix B.

n) Figs. 23 and 24 have been merged into one figure.

o) Figs. 25, 26, and 27 show the performance of the parameterizations for the elevated stratopause conditions. Fig. 25 has been moved to Appendix B. The others have been retained but the 2 panels of Fig. 27 have been merged into one panel.

p) As suggested by this referee, 2 new figures have been included showing a comparison for a few individual p-T profiles and the summary for the comparison carried out for 225 p-T profiles of the high vertical resolution WACCM-X model.

Overall, the number of Figs. in the main text have been reduced from 28 to 18 (including the two newly added figures) and some of them with a reduced number of panels; and the number of Figs. in the appendix has been reduced from 20 to 9. 22 figures have been moved to a supplement.

About "... put more efforts on the description of the routine itself, its implementation to a GCM and its limitations", the description and its limitations are already discussed in detail. Its implementation is described in the Readme file of its distribution. Note that this has been significantly changed after the previous response. Further, we have included Appendix A with some notes and recommendations for its use, including the vertical range.
Although the parameterisation is specifically developed for the CO2 15μm non-LTE region, it also works for the LTE region, but the user should be cautious that other important cooling rates in the LTE region, such as those of O3 and H2O, are not included. We recommend that the GCM users utilise their radiation scheme in the LTE region and this parameterization in the non-LTE region (e.g., above ~50 or 60 km).

2) Regarding the routine, I did not see any reference or repository for it provided along with the manuscript. As far as I understand, this is now a requirement for EGUsphere journals, so I wonder when and how did the authors plan to publish the routine. Besides a general conformity with the journal's rules, it would be advisable to get a self-consistent compilable code with the cooling rate parameterization routine called from the main code with some standard atmospheric profile, which could be replaced with a test profile by the user. In the next paragraph, I explain the current problem I see with the accuracy of the updated parameterization, and I guess the GCM-modelers would have liked to test it themselves prior to implementing it to the model. Summarizing this point, the authors must provide a ready-to-use code and provide an instruction for its compiling and running both as a standalone routine and as a part of the test code, that might be useful not only for GCM-modelers, but for other researchers wanting to get a quick estimate of non-LTE cooling profile for a given atmosphere.

**Reply:** The referee is fully correct. Momentarily, the parameterization was made available as a Python routine at https://zenodo.org/doi/10.5281/zenodo.10547026. We have now made it available in its Fortran version at https://zenodo.org/doi/10.5281/zenodo.10849969.
We thank the referee for the advice and recommendations. We have taken them into account and are providing some input and output examples. Further, the referee made a very good point that it could be used by any researcher and not only in GCMs. We are including some instructions about how to use it in the README.md.

3) The authors show that the routine works fairly well for the multitude of atmospheric profiles they use in the tests. This is quite impressive given a large variability of CO2 concentrations used in the study. Still, I cannot consider these tests to be fully representative because the real atmospheric temperature profiles are far from those shown in Fig. 2 of the manuscript in terms of vertical variability. Moreover, they are even worse than the ones shown in Fig. 20 because the averaging kernels of MIPAS instrument in the middle atmosphere are broad (e.g. Dinelli et al., 2021) and, therefore, this instrument cannot capture any wave shorter than 5-7 km. In addition, MIPAS is a limb measuring instrument, so it washes out horizontal inhomogeneities and this further reduces its vertical resolution. At the same time, the gravity waves (GWs) are known to be composed of several waves of different wavelengths and their amplitude increases with height up to the top of the "wave-turbopause" layer at 90-110km (e.g. Offerman et al., 2007; Ge et al., 2022), which overlaps with the non-LTE-affected area. I believe, the MIPAS profiles used in the study do not represent a full picture of a gravity-wave perturbed temperature profiles, so the tests performed with them show only a partial truth. Moreover, large RMS values presented in Fig. 24 even for these, partially smoothed, profiles make me think that the real accuracy of the parameterization is far from 1−2 K/day as it is stated in the abstract (I do not consider the specific case of elevated stratopause, which is addressed; I'm talking about the accuracy of "regular" profiles). If the tests I suggest below do not disprove my suspicions, this wouldn't cancel the suggested parameterization. But, the GCM modelers will have a clear picture of what they will use and act accordingly if their models produce lifelike GW-perturbed temperature profiles.

**Reply:** Overall we agree with the referee in testing a few more cases, see below.
However, let us first clarify a couple of points about the MIPAS temperatures used. It is true that they have a limited vertical resolution and do not fully recover the actual pT altitude structure. However, they are also affected by the noise of the instrument, which frequently maps in the

[Figure]

profiles as a vertical variability larger than the real one. The given reference about MIPAS (Dinelli et al., 2021) describes the retrieved temperatures from the MIPAS **nominal** mode. The MIPAS data we used in the manuscript are those retrieved from the middle atmosphere (MA)

mode, which has a better vertical sampling and finer AKs than the nominal mode ones (see García-Comas et al., 2023). We are including above a couple of Figs. showing some profiles extracted from MIPAS data for 15 February 2009. They clearly show wave structures and very unusual pT profiles during the elevated stratopause event. Hence MIPAS profiles are not that far away from those measured by the lidar instruments and some of them are even more extreme.

We concur with the referee that the values of 1-2 K/day given for the accuracy are for the reference p-T profiles. When applied to generic wavy dividual p-T profiles the errors are larger, as demonstrated by its application to MIPAS data, particularly for polar summer and polar winter conditions. For that reason, we tested it with the MIPAS temperature profiles and even showed the case of elevated stratopause conditions.

In general, we think the most proper way of expressing its accuracy is by a mean error value (bias), which informs us about its accuracy in a global sense, and by the RMS (as assessed from a significant sample), as an appropriate estimate of the error for individual profiles.

We have clarified it in the manuscript, particularly in the abstract and conclusions. Further, we have also assessed its accuracy for the high-resolution WACCM-X output p-T profiles, which very much resemble the lidar measurements, as described below.

4) Here is the test I believe is necessary to conduct in order to properly estimate the accuracy of the suggested parameterization. Instead of using the MIPAS profiles and showing the averaged effects, I ask the authors to use the individual temperature profile coming from a ground-based lidar (e.g. Spitsbergen (78°N, 15°E), ALOMAR at Andøya (69°N, 16°E), Kühlungsborn (54°N, 12°E), Boulder (40°N, 105°W), Fort Collins (41°N, 105°W), Logan (42°N, 112°W), Arecibo (18°N, 293°E), Cerro Pachon (30°S, 71°W), or McMurdo (78°S, 167°E).) Since these lidars do not cover the whole atmospheric profile, in the lower atmosphere one can take a corresponding smooth profile from the reanalysis or from any other model. Among the profiles provided by the lidar teams, one has to pick up the ones, which clearly show the GW-signature (at least, the wave of +/- 10 K amplitude at 100km), and perform the exact and parameterized calculations of radiative cooling rate for these profiles. It would be also interesting to see how different wavelengths affect the accuracy, because the non-LTE effects are different in the case of a short-and large-scale perturbations. I have to stress here that due to a nonlinear nature of temperature perturbation effects on non-LTE cooling rates, these results cannot be replaced with the ones obtained for averages.

**Reply:** We have looked at most of the lidar temperature measurements listed above (see https://cloud.iaa.csic.es/index.php/s/BFaiqFTy8kwL5Jn). The application of the parameterization to some of those profiles requires the extension of the profiles to lower and higher altitudes and to include atomic and molecular oxygen concentration from somewhere else. It happens that we have available the output of a high-resolution version of the WACCM-X model, 0.25°x0.25° in latitude/longitude and a vertical grid of ~0.5 km in the lower levels, and extending from the surface to above 500 km. With such a fine grid the model itself can internally generate gravity waves thus providing temperature profiles with a vertical structure very similar to that measured by high vertical resolution lidars of the mesosphere and lower thermosphere. Compare, e.g. some of the plots in https://cloud.iaa.csic.es/index.php/s/BFaiqFTy8kwL5Jn) with Fig. S23 in the supplement, which shows that those pT profiles have a similar amplitude and vertical wavelength. The WACCM-x data has the advantage of covering from the surface up to >500 km and contains the O, O2 and N2 vmr profiles. Further, this is precisely the kind of GCM model for which the parameterization has been mainly devised. We have applied the parameterization to 225 profiles (Fig. S23) and show some examples of the differences for individual p-T profiles (Fig. 16 and S24-S26) and the mean and RMS for this dataset (Fig. 17). Note the values are very similar to those obtained for the MIPAS test.

**Minor comments and technical corrections**

>>> Lines 4 and 41: this statement is too general. In fact, there are ways of including the non-LTE calculations to GCMs. For example, this was done for Martian GCM (Hartogh et al., 2005) and I don't see why this can't be done for Earth. The corresponding manuscript is still in discussion in the same journal, so it can't be referenced, but the authors claim that the same approach works for the Earth's atmosphere.

**Reply:** We agree that there are different ways of including non-LTE in GCMs but generally they are all very CPU time consuming. "The corresponding manuscript … " Does the referee mean the manuscript submitted recently by Kutepov et al. 2023? We have included the mentioned reference and a citation to the manuscript by Kutepov et al. recently submitted to this journal.

>>> Line 11: since the atomic oxygen is a variable component and the reaction 1c is directly linked to it, it would be good to have it as a variable parameter, see the second general comment

**Reply:** The user will have the option of including both, the atomic oxygen concentration profile and the collisional rate K1c. This is specified in the documentation of the code provided now.

>>> Line 17: the measured profiles themselves is not a ground truth because of the vertical resolution, see the third general comment. Moreover, one cannot average the results for individual runs and present this difference as an error, because the GCMs accumulate the error due to non-linearity of the processes.

**Reply: First point**: We agree it is not the ground truth but we think it is not thus far (see plots above). We have also analysed and tested the parameterization for lidar-like pT profiles.

**2nd point:** We agree. Still, that average gives us a measure of the "error" (systematic) in a global sense (eg the cooling at high altitudes in the previous version is always about 2K/Day larger than in the line-by-line calculations). But have provided the RMS as an estimate of its error for individual profiles as assessed from a significant sample.

We do not understand what the referee means by "... because the GCMs accumulate the error due to non-linearity of the processes." Does he/she refer to the fact that the cooling is proportional to exp(-kE/T) and not to T? We can only assess cooling rate errors induced by the parameterization, but not non-linear effects on modelled quantities such as temperature and winds.

>>> Line 58: Figure 1 is not informative, see the first general comment

**Reply:** OK. It has been removed removed.

>>> Lines 120-150: I guess, the readers will be confused here. It looks like GRANADA non-LTE code cannot work in LTE mode whereas normally it requires one line or one flag to calculate LTE populations. Could you, please, explain?

**Reply:** GRANADA computes non-LTE populations and non-LTE cooling/heating rates. We do not need a code for computing LTE populations and GRANADA has not been specifically designed/coded for computing LTE cooling rates. We already clarified in the manuscript that we use KOPRA for LTE cooling rates using a Curtis Matrix approach.

>>> Line 125: I didn't find any mention of line mixing effects here. Are they not important?

**Reply:** KOPRA does include line mixing and it was used in these calculations. We have included a sentence stating it.

>>> Line 135: I didn't get the phrase about the "oscillations in RFM results". Could you, please, clarify?

**Reply:** We refer to the minor oscillations in the RFM cooling rate profile, e.g. around 50 km, 60 km, 65 km, and 85 km.

>>> Line 145: And what about the results without this additional iteration?

**Reply:** In short, the cooling rates would be less accurate. Without that final iteration (for specified accuracies for the vibrational temperatures), the non-LTE populations, e.g. the vibrational temperatures do not change significantly (see Funke et al. 2012). However, we did not describe in that work the non-LTE cooling rates. To properly calculate the cooling rates, in addition to the specific procedure described in that work, a final iteration to compute the radiative field in all bands is required in order to account for the overlapping between the different bands. The effect of not including it is to have slightly less accurate cooling rates.

>>> Line 171: I recall that the difference is about 2-3K, and I wouldn't call this a small change, so one should not neglect the daytime vs nighttime differences. This is addressed in Sect. 8, so the wording has to be changed here

**Reply:** It is not that large, it is about 1-2 K at most (see e.g. Fig. 3 in López-Puertas et al., 1990,*), which in relative terms is very small since that day/night difference is larger above ~90 km, where the cooling rate is large.
(*) Our more recent calculations of day/night differences of the $CO_2$ 15 μm cooling rates are smaller than 1K/day for all reference pT profiles, at any altitude and for $CO_2$ vmrs up to 5 times the pre-industrial value, except for MLS and SAS pT profiles near 105 km and $CO_2$ vmr 4x and 5x the pre-industrial value.
We already stated that this point is addressed in Sec. 8 (line 175). The text has been slightly changed for clarification.

>>> Lines 200-215: all these descriptions are correct, but I doubt that they add something to the understanding of the routine or its accuracy, see the first general comment on the scope and focus of the manuscript.

**Reply:** We already state the reason why including this discussion, lines 201-202: *"This comparison is useful from a physical point of view and it is required to establish the boundaries of the different atmospheric regions of the parameterization"*. Fig. 8 has been reduced and Fig. 9 has been moved to the Appendix B. We have also slightly changed the text in that section.

>>> Lines 220-225 (see also the comment to lines 437-462): if we assume that the k(CO2-O) used in the models is equal to 3 x 1e-12 cm3 s-1, then this experiment shows the effects of quadrupling the atomic oxygen mixing ratio. In addition, the reader might be misled by switching from atomic oxygen to k(CO2-O) and back. It would be enough to present this numerical experiment as the one performed for doubled (or quadrupled) atomic oxygen.

**Reply:** That is what we did, just to half the k(CO2-O) nominal value of ~6e-12 cm3 s-1. We have clarified this in the text and changed "perturbed the kCO2−O collisional rate by a factor of two" to "dividing the kCO2−O collisional rate by a factor of two".

>>> Lines 248-395: in fact, this deserves a general comment, but I cannot suggest to modify the general approach, it is what it is, and it works for smooth profiles for a current atmosphere. But what if the atmosphere changes some day and the LTE/NLTE1,2,3,4 regions become different? Let's imagine that a modeler wants to calculate cooling rates for an Earth-like atmosphere, but

with a different vertical temperature structure. How can he/she tell the limits of applicability of this parameterization?

**Reply:** Very likely, it will not work with high accuracy for those situations. This parameterization has not been designed for any planetary or p-T temperature profile or any atmospheric composition. It has been specifically tailored for the current-like Earth's atmosphere with projected high CO2 vmr in the future.

>>> Line 304: this is just one of the examples of the problem outlined in the previous comment. The error introduced by this "implicit assumption" depends on the atmospheric profile and is hidden deep in this module. It is small for the current atmospheres, but it is not guaranteed that it will remain small for some unusual temperature profile.

**Reply:** That particular aspect is not an issue because the atmosphere is in the optically thin regime at those altitudes and the parameterization is very accurate there. Look at all the tests performed and see how the errors tend to vanish at high altitudes.

>>> Lines 437-462: this is another potential candidate for a general comment. The problem is that there are three values of this rate coefficient, k(CO2-O): the first one is measured in the lab and is about 1.5−2.0 x 1e-12 cm3 s-1 (e.g. Castle et al., 2012), the second one is about 6.0-9.0 x 1e-12 cm3 s-1 retrieved from space-borne observations of 15-micron CO2 emissions, and the third one, 3.0 x 1e-12 cm3 s-1 was suggested for using in the GCMs. This paradox has not been resolved over the decades of its study, but it shouldn't matter for the parameterization itself, because it is supposed to calculate the cooling rate with any given k(CO2-O).

**Reply:** Exactly, we fully agree with the referee. The point we want to stress is that the parameterization has been optimized for k~6.0e-12 cm3 s-1, and we would like to assess the errors when it is used for half of this value. It is important to tell the reader that if using a rate half of the value for which it has been optimized, it does not significantly lose its accuracy.

>>> Line 455: how do you explain that the errors obtained for a smaller k(CO2-O) are larger? Usually, the larger the quenching rate, the larger the cooling rate, and the magnitude of this error is linked to cooling, as in the case of enhanced CO2.

**Reply:** The very likely reason is that the coefficients of the parameterizations have been calculated, and it is therefore optimized, for the larger k(CO2-O) rate. In any case, the differences are very small for both cases, for Ko and Ko/2.

>>> Line 462: Fig. 20 is barely readable. Please, provide only a couple of profiles coming from a different source (see general comment 4) and show the errors for the corresponding cooling rate profile on the right-hand side panel.

**Reply:** The intention of showing this figure is not to distinguish between the different pT profiles but to show that the parameterization has been tested for an ample range of temperature structures, ranging from polar-summer-like profiles to polar-winter-like profiles and for elevated p-T profiles. That is, it is a very demanding test!
The figure has been moved to Appendix B.

References
García-Comas, M., Funke, B., López-Puertas, M., Glatthor, N., Grabowski, U., Kellmann, S., Kiefer, M., Linden, A., Martínez-Mondéjar, B., Stiller, G. P., & von Clarmann, T. (2023). Version 8 IMK–IAA MIPAS temperatures from 12–15 μm spectra: Middle and Upper Atmosphere modes.*Atmospheric Measurement Techniques*, *16*(21), 5357–5386. https://doi.org/10.5194/amt-16-5357-2023.

**Response to Reviewer 2**

We would like to thank the referee for the helpful review and the constructive comments. His/her comments are given below in black and our responses in blue.

**General comments:**

>>> As the developer of an LTE radiation scheme used in a GCM, my review is from the point of view of a potential user of the code who would want to improve representation of the upper atmosphere in their code. There is certainly a strong case for a revised non-LTE parameterisation that can handle much larger concentrations of CO2, and therefore the contribution from this paper is welcome. But as I am not an expert on non-LTE effects I can't comment on the scientific details of this particular parameterisation. Naturally I would expect the code to be available at the time of submission to GMD and I expect the authors to rectify this.

**Reply:** We appreciate your valuable suggestions even if not being an expert on non-LTE. The referee is fully correct about the availability of the code. We were not aware of the need for its availability during the review process and thought of providing it during the review process. We provisionally made available the parameterization as a Python routine, at (https://zenodo.org/doi/10.5281/zenodo.10547026); and we are now making it available in its Fortran version at https://zenodo.org/doi/10.5281/zenodo.10849969.

>>> The algorithm is presented as a complete radiation code, but the LTE part described in section 5.1 is quite crude for tropospheric radiative transfer since it doesn't represent scattering or cloud overlap and heterogeneity. I imagine that most GCM developers (like myself) would want to use their existing radiation scheme in place of the algorithm described in section 5.1 for the LTE region from 0 to 70 km, and then to add on NLTE regions 1-4 described in section 5.2-5.4. It would therefore be helpful to describe how to do this:

**Reply:** We did not make it clear in the text. Although the parameterisation is specifically developed for the CO2 15μm non-LTE region, it also works for the LTE region, but the user should be cautious that other important cooling rates in the LTE region, such as those of O3 and H2O, are not included. We recommend that the GCM users utilise their radiation scheme in the LTE region and this parameterization in the non-LTE region (e.g., above ~50 or 60 km). We have included a README.md file and Appendix A in the paper with some notes and recommendations for its use, including the vertical range.

>>> Can I simply replace the cooling rates from my code with the cooling rates from the non-LTE parameterisation above 70 km.

**Reply:** Exactly, or from even above 50 km. A switching region is advisable to be used. See the README.md file and Appendix A.

>>> Do I need to provide any upwelling fluxes at 70 km to feed the treatment of the regions above?

**Reply:** No, the parameterization calculates internally the entire cooling rate profile (with its 'own' upwelling flux from the LTE region). However, it is recommended that the input file (with pressure, T, CO2, etc. ) covers the lower part of the atmosphere to properly calculate internally the upwelling flux. If some layers at the bottom of the atmosphere are omitted, the parameterization still works but the cooling rates might not be fully correct, although they can be corrected by including a "surface" temperature. All these details are documented now in the README.md file and Appendix A.

>>> The description of how one region is coupled to the one above is not clearly described in the text.

**Reply:** That's correct. We are now giving recommendations above the altitude range for the use of the parameterization, as mentioned in the previous point. The user only needs to consider the lower region, in LTE, and then this parameterization in the non-LTE region above it.

>>Is the vertical resolution of the parameterisation fixed or can I run it on my own vertical grid?

**Reply:** The user can provide its own vertical grid in a specified input file which should contain the pressure (hPa), temperature (K), and VMRs of $CO_2$, O, $O_2$, and $N_2$. Then, the parameterization interpolates those parameters onto its internal grid.
All these details are documented now in the README.md file and Appendix A on "Notes and recommendations for using the parameterization".

>>> Do I need to simulate all regions or can I omit the uppermost 1, 2 or 3 regions if I am not interested in modelling temperatures above a particular height?

**Reply:** In principle, the users should input the parameterization from the surface up to the upper boundary of interest. We give also recommendations for the upper boundary. All these details are documented now in the README.md file and Appendix A.

>>> Has the parameterisation been coded in Fortran or C (needed for a GCM) or in an interpreted language like Python? What is its computational cost? How large is the dataset that needs to be read in (e.g. the various look-up tables that are used to populate the Curtis matrices)?

**Reply:** It was originally written in Python and was made temporarily available in that language. The version for use is written in Fortran and available at Zenodo (see the first point above). There is no external dataset for reading. The Fortran routine contains all the needed coefficients as parameter variables. It is very fast, the routine takes only between 1.5e-5 s to 7.5e-5 s (15-75 μs) depending on the extension of the input atmosphere, the processor and the Fortran compiler. We have included a new Sec. 6.3 where we specify its performance. All of this is documented in the README file and Appendix A.

SPECIFIC COMMENTS

1. Abstract: it would help to mention the magnitude of LTE heating rate errors (i.e. LTE minus non-LTE), in order to put into perspective the magnitude of the errors in the parameterisation of non-LTE effects (i.e. parameterised non-LTE minus accurate non-LTE).

**Reply:** The "errors" of the parameterization are in general way smaller than the deviation of cooling rates from LTE, e.g. than the non-LTE-LTE differences. We have included a sentence about these differences in the abstract.

2. There is an excessive number of figures, and they are sometimes referenced out of order. My suggestion is to select one (or at most two) of the test atmospheres and then when you have a 6-panel figure with a separate panel for each atmosphere, reduce it to just one panel in the main body of the text, and then combine figures together when it is useful to have panels next to each other. For example, combine Figs. 6a and 7 into a 2-panel figure, since they show the same thing for the same atmosphere, just on a different scale. Then put the other five atmospheres in Supplementary Material (not in appendices) and refer to them sparingly. I suggest that Fig. 1 is

removed as it is not needed, and Fig. 2 is replaced by Fig. 11 (the latter shows the same as the former but with more information). There are plenty of other opportunities for the authors to cut down the number of figures.

**Reply:** We agree. The other referee has made a similar comment. The reason for showing the results for the six p-T profiles is that very frequently, the effects of the discussed parameter depend very much on the temperature profile.
Please see the response to the first general point of the other referee for all the detailed actions that we have taken. We have drastically reduced the number of figures/panels in the main text. It has been reduced from 28 to 18 (including the two newly added figures) and some of them with a reduced number of panels. Also, the number of Figs. in the appendix has been reduced from 20 to 9.  22 figures have been moved to a supplement.
About "and refer to them sparingly", we have tried to make the reading smooth but in general, we prefer to point the reader to the figure in question so he/she can better understand the argument. The reader always has the option of not looking at the figures if unnecessary.

3. Lines 102, 200 and elsewhere - please refer to profiles #3 and #6 as "present day" and "4x pre-industrial" so the reader doesn't need to flip back to remind themselves what these numbers refer to, and similarly for the other hash-numbered profiles. Also, Fig. 9 could state the factor multiplying the pre-industrial figure surface value for each point somewhere on the figure.

**Reply:** Definitely, we have done that. About Fig. 9, which has been moved to Appendix B,  now includes in the caption the multiplying factor corresponding to each CO2 vmr profile.

4. Line 219: "pointing" -> "role in determining"?

**Reply:** Thank you. That was a leftover of a previous version. It has been corrected.

5. The equations are largely taken from Fomichev et al. (1998), but it would be useful to improve clarity in several places. For example, Equation 1 is simply a matrix-vector multiplication - why not present it as such? The equations for "a" and "b" on lines 264 and 265 are presented as if they define the element the Curtis matrix (via the equation on line 261), but they themselves contains the elements of the Curtis matrix on the right hand side as a scaling for the terms involving the band strengths. So one is left wondering how the actual value of the elements of the Curtis matrix are determined.

**Reply:** "Equation 1 is simply a matrix-vector multiplication - why not present it as such? " Yes, writing it in a matrix formulation will be more concise (and clean), but we prefer to write it explicitly, particularly because the summation goes over different altitude ranges and it is easily written in this form. If using the matrix notation we would have to define new matrices (symbols) for the different altitude ranges. But you are right that since **a** and **b** are written with indices, it is not necessary to be typed in bold. We keep it for easy reading but maybe the editor will remove their bold typeface.
The Curtis matrix is actually embedded in $\mathcal{A}_{i,j}^t(\nu)$ , see lines 253-254.  **a** and **b** are scaled Curtis matrices.
We should mention that the formulation has been extended to describe also the a and b coefficients corresponding to the surface flux, e.g., a_surf and b_surf.

6. Line 404: remind the reader that by "both" parameterisations you mean Fomichev's original one, and the one in this paper.
**Reply:** Thank you. We have done it.

**Response to the comment by Alexander Kutepov.**

We are responding below to the comments by our colleague. His comments are listed in black with our responses in blue.

**GENERAL COMMENTS**
This is a large manuscript with very large numbers of plots both in main body and appendixes. Although it is well written and structured, many tests of a new routine repeat one another and, therefore, look excessive.

**Reply**. The point has been taken. The other two referees made similar comments. The number of figures in the main text and the Appendix have been substantially reduced (see reply to referee 1 for the details). The number of Figs. in the main text has been reduced from 28 to 18 (including the two newly added figures) and some of them with a reduced number of panels. Also, the number of Figs. in the appendix has been reduced from 20 to 9. 22 figures have been moved to a supplement.
Nevertheless, we would like to show in the manuscript the benefits and limitations of the parameterization. Hence all the tests have been retained and an additional one has been included as suggested by Referee 1.

>>> The work takes us back to the late 70s - early 80s of the last century, when new for that time techniques of the approximate 15-micron $CO_2$ cooling calculations (both for LTE and non-LTE conditions) were developed, see (Kutepov, 1978) and (Akmaev and Shved, 1982). The revised version of the Fomichev et al, 1998 (hereafter F98) routine described in the manuscript does not represent any innovation and does not suggest any new option for the GCM users and developers.

**Reply**. We recognize in the manuscript that this is based on the same approach as Fomichev et al, 1998. Our major aim was to extend it to higher $CO_2$ vmrs and once we tackled it, we tried to improve it also in other aspects, as detailed in the manuscript. We think it is a new option, as GCM models can now be run for higher $CO_2$ vmrs, more accurately and faster.

>>> This statement may be explained as follows. Any physical parameterization for GCM must be able to react as realistic as possible at steady changing local physical state of the atmosphere in the modeling process. This is particularly true for the infra-red radiation and its effects (local cooling or heating), which are critically important for adequate modeling of energy balance. It is well known that instantaneous p-T distributions in modern GCMs of middle and upper atmosphere exhibit very strong variability, caused by the superposition of tidal and gravity waves of different amplitude and vertical scales. Same is true for the atmospheric p-T distributions measured in both ground and space experiments. Therefore, the parameterization of the 15-micron $CO_2$ cooling, which is a main radiative cooling mechanism of these layers, must properly react to this variability.

However, the matrix parameterizations of the 15-micron cooling are unable to provide adequate reaction to strongly disturbed T distributions. This was well known already 30 years ago for those (I was among them) who worked on developing the first version of the F98 parameterization. Demonstration of large parameterization errors for wavy T profiles was not included in the paper by Fomichev et al, 1993, however, it contained at least the warning addressed to its users ***"It is recommended for the calculation of the radiative cooling for smooth temperature profiles, namely for profiles undisturbed by micro- and meso-scale motions."*** When the updated 1993 routine was released by Fomichev et al, 1998 nothing changed, the revised routine was still unable to treat wavy distributions, see (Kutepov and Feofilov, 2023, hereafter KF23). Again, as in the 1993 paper, the accuracy of F98 was

demonstrated exceptionally for smooth T profiles. However, the authors dropped from the paper text the warning for its users cited above. Since that time the F98 routine has been widely used in GCMs of middle and upper atmosphere for any T distributions.

It is also worth noting here that 15-micron CO2 radiative cooling is strongly non-linear in respect to the temperature variations. Therefore, the zonal mean cooling, which is enthusiastically discussed in this manuscript, is not equal to that calculated for the zonal mean temperature. The authors, however, pay no attention to this fact!

**Reply**. Precisely for the reasons mentioned above, we tested the parameterizations with the realistic pT profiles retrieved from MIPAS. Rather than giving a recommendation we assess its performance and give errors so the users are aware of them. One important point: The individual pT profiles of MIPAS do show a large vertical structure because they are affected by the instrument noise. They are not smoothed profiles at all. That is seen in old Fig. 20 (now moved to Appendix B, Fig. B6) and in a couple of Figs. in the response to Referee 1 above, where we can see their large vertical variability.

Let us clarify also the point about "zonal mean cooling" and the zonal mean temperature. We did NOT calculate the cooling for the zonal mean temperatures. We calculated the cooling for each individual p-T profile, then the differences wrt the reference calculations, and later they were zonally averaged. For that reason, we also show the RMS.

As Referee 1 also pointed out, we agree that we should not consider the mean as the error of the individual cooling rates profiles of the parameterisation. We have provided the mean values (bias), which will inform us about its accuracy in a global sense (eg the cooling at high altitudes in the previous version is always about 2K/Day larger than in the line-by-line calculations), and also the RMS, as an appropriate estimate of the parameterization error for individual profiles. We have clarified it in the manuscript, particularly in the abstract and conclusions.

Further, as suggested by Referee 1, we have also assessed the accuracy of the parameterization for the high-resolution WACCM-X output p-T profiles, which very much resemble the lidar measurements, as described above.

Of course, the parameterization is not perfect and one would wish to run the full non-LTE model for each p-T profile in each grid of the GCM model for each run. Here we offer a fast parameterisation (over 6000 times faster than the faster option of KF23) with reasonable (well-assessed) errors. As you have stated in the paper you have submitted (p. 11, lines 282-283), let us the modellers choose what they prefer/need.

>>> Now back to the current manuscript. The authors invested large efforts to update the F98 routine. They declare that the revised routine (hereafter L-P.23) allows calculations of 15-mucron cooling with higher accuracy for current CO2 vmr. It calculates with reasonable accuracy also the cooling for much higher CO2 concentrations. Nevertheless, again the accuracy of L-P.23 is demonstrated only for very smoothed "standard" T distributions.

**Reply**. We have shown both, the accuracy for the reference (smooth) atmospheres and also for the MIPAS pT atmospheres, which show many "wavy" profiles and very uncommon nearly isothermal profiles (see previous point and the Figs included in the reply to Referee 1). As suggested by Referee 1, we have also assessed now the accuracy of the parameterization for the high-resolution WACCM-X output p-T profiles, which are very similar to the pT profiles measured by lidar instruments. Please, see the response to Referee 1 above (p. 5).

>>> Meanwhile, the authors are aware about large errors F98 routine has for disturbed T profiles. D. Marsh was the co-investigator of a recent NASA grant (Kutepov, 2021), where I was the PI. He and his team received funding for testing the KF23 algorithm for calculating 15-micron non-LTE cooling, comparisons of this routine with F98 parameterization, and for

installing KF23 routine in the WACCM model. This study showed that F98 causes very large cooling errors (up to 25 K/day) on wavy profiles. These errors are discussed in KF23.

**Reply**. We agree that for very wavy individual profiles, the cooling rates might differ significantly from the reference calculations in the upper mesosphere. We show these differences now in Fig. 16 and more examples in the supplement, Figs. S24-S26.
We state now clearly in the text that the zonal mean of the differences when averaging over narrow latitude bands (eg. new Fig. 12 (old 22)) are indicative of the "bias" in the parameterization wrt the reference calculations, and state that they can be significant (up to 2-3 K/day). Further, as an estimator of the error of the individual cooling rate profiles, we also show the RMS, which, as shown in Fig. 13, can be significant near the mesopause.
We have changed the text in the abstract, conclusions and the corresponding sections.

We would like to point out that, as you suggested in your recent manuscript (p. 11, lines 282-283), there might be room for multiple non-LTE parameterizations and, for climate simulations, the errors introduced due to small-scale temperature variability might be well tolerated by modelling groups if it comes at a reduced numerical cost. The KF23 parameterization adds significant numerical cost in running a GCM and therefore may not be suitable for centennial-scale ensemble simulations. In limited testing by the software engineer at NCAR it was not insignificant and the option to call it only every other timestep needed to be introduced to reduce the cost. We should point out that KF2023 is about 300 times slower than F98 and about 6600 times slower than this parameterization.

>>> Meanwhile, the authors of the manuscript are quite honest when, describing the main motivation of their study, they write: ***"In our case we have the option of developing a completely new parameterization, to adapt other CO2 parameterizations (as those cited above), or to extent and improve the parameterization of Fomichev et al. (1998). Attending mainly to practical reasons of promptness, we opted for the later."*** The keyword here is "***promptness".***
This "***promptness"*** looks somewhat strange after 25 years of no interest of the authors to the problem of fast and accurate calculation of radiative cooling for the Earth's GCMs. Knowing about the drawbacks of F98 routine and other matrix parameterizations we spent these years developing accurate non-LTE radiative transfer techniques, which are free of these drawbacks and are fast enough to be applied in GCMs. The results of this long-term study are summarized in KF23.

**Reply**. The recent interest in improving/extending the parameterization is that a) climate simulations conducted for CMIP have only just begun to include models that resolve the stratosphere and mesosphere, and b) the standard climate sensitivity metric includes a 4xCO2 experiment which is incorporated into the DECK for CMIP6. This work paves the way for moving the F98 code to modern Fortran or C++. It is unlikely that, as modellers increase the model height of their model top for climate simulations, they would want to incorporate code that relies on the compatibility of compilers to 30+-year-old coding standards or does not cover the full CO2 range of the DECK.

>>> Finally, what kind of new product these authors developed with the main motivation "to be prompt"?

**Reply**. To make available a fast and efficient algorithm capable of coping with the very large CO2 VMRs. Some GCM modellers urged us to extend it to large CO2 vmrs.

>>> Again, I do not need my own judgement. It is enough to cite what the manuscript authors write when they describe large errors, much lagers than 0.5-1.0 K/day reported for standard

profiles, which they observed for all profiles at latitudes northernmost of 50 N for a single non-standard situation with a pronounced elevated stratopause event: "***Both parameterizations underestimate the cooling in that atmospheric region. The new parameterization has, however, a better performance above about 80 km, but in the strat-warm/elevated stratopause region (80–100 km) it still underestimates the cooling by 3–7 K Day−1 (~10%)".***

**Reply**. We do not understand this kind of comment. We just did what we scientists are supposed to do, e.g., to recognize the limitations of the parameterization. These elevated stratopause situations are the most difficult situations to handle with this parameterization, and this one of 2009 in particular, which was extremely strong. However, globally, they are rather sporadic and limited to certain regions and times. In any case, the parameterization, although not perfect, calculates the cooling rates with errors of ~10%.

>>> This looks like a confession that in a non-standard situation the new l-P.23 routine works no better than F98.

**Reply**. It is better in about 2 K/Day above 5e-4 hPa and also around 4e-3 hPa (see Fig. 15, previous Fig. 27); and true, it is a very unfavourable situation.

>>> And further: *"It seems clear that part of this underestimation is caused by the fact that such atypical temperature profiles (see Sec. 3.1) were not considered in the parameterization. However, its inclusion would not solve the problem as in the calculations of the coefficients a trade-off of the weighting of the different p -T reference atmospheres have to be chosen (see Secs. 5.1 and 5.2). Thus, it might ameliorate the inaccuracy for these elevated-stratopause events but would worsen the accuracy for other general situations. This manifests the difficulty/limitation of this method to provide accurate non-LTE cooling rates for all temperature structures (gradients) that we might find in the real atmosphere."* I absolutely agree with this statement: this approach for parametrizing the 15-micron non-LTE cooling in the middle and upper atmospheric layers, which was applied in previous routine versions in 1993, in F98, and repeated in L-P.23 **is a deadened approach**.

**Reply**. We do not agree with this final statement. The fact that the parameterization is not very accurate (errors of ~10%) for certain atypical and unusual conditions should not be generalized for all conditions. As very often occurs, it is a trade-off between accuracy and efficiency.

>>> In KF23 we discuss in detail its drawbacks. Briefly: it is impossible to adequately estimate the non-LTE cooling in the very variable atmosphere by dividing it in several altitude regions, where different techniques or expressions for cooling calculation (although linked in various ways) are used. Only exact algorithms, which rigorously describe the radiative energy exchange between various altitude layers and the non-LTE radiative field coupling with atmospheric heat reservoir may satisfy the current cooling accuracy requirements.

**Reply**. We think it depends on what we understand by "adequate estimate". We are giving conservative error estimates so the users can decide if it is (or not) adequate for his/her cooling rate accuracy requirements. We offer an option, but not stating that this option is the best. See also the comment above.

>>> *It is my opinion that this paper in its current shape must not be published. It does not provide any significant improvement compared to previous work(s)*.

**Reply**. We have no comment on the first sentence of the paragraph. We will accept the editor's decision. Regarding the second sentence, we think it is already clear in the manuscript the significant advantages/improvements of the parameterization.

>>> To be published the manuscript requires significant major revision:
(1) it must demonstrate how revised routine works for T profiles disturbed by the strong waves.

**Reply**. As also requested by Referee 1, we have included now a new section (Sec. 8) with 2 new figures (and some more figures in the supplement) where we show the application of the parameterization to the "wavy" pT profiles as calculated by the high-resolution WACCM-X model, which very much resembles the vertical structure present in the high-altitude resolution lidar measurements.

>>> If the routine fails on the wavy profiles, but the authors still recommend it for further usage in GCMs, then (2) they need to justify that these errors have negligible or no effects on the GCM model results.

**Reply**. We think that it corresponds to the GCM modellers to decide if the estimated accuracy, given in Figs. 22 and 23 (new Figs. 12, 13, and 17), fulfil or not their accuracy requirements.

**SPECIFIC COMMENTS**
**1. Why exact methods of the non-LTE CO2 cooling calculation cannot be applied in GCM?**
The manuscript authors write: ***The computation of the cooling under those non-LTE conditions requires the solution of the radiative transfer equation (RTE) which is a non-local problem and requires a large amount of CPU time. Therefore, the solving the RTE in general circulation models (GCMs) or climate models that extend in height above the stratopause is impractical and efficient parameterizations of the CO2 infrared cooling have been developed and implemented in such models.***
I disagree with this statement. It does not matter whether the LTE approach is applied or the non-LTE problem is considered, in both cases the calculation of cooling requires the solution of RTE, and is, therefore, the non-local problem.
Moreover, the computing costs of exact RTE solution in modern algorithms are not the main problem, which makes ***"the solving the RTE in general circulation models GCMs impractical".*** The authors mean here not just RTE but the entire non-LTE problem.

**Reply**. You are correct, even in LTE, it is still a non-local problem. About the second sentence, you are correct again, we meant the entire non-LTE problem. We have changed the text accordingly.

>>> Inversion of large matrices to get the populations in the developed in 1950s CM (Curtis, 1956) matrix algorithm, which the members of this team utilize since 1980s, is most computational costly part of the non-LTE problem solution. The authors either are not aware of or ignore dramatic progress in the developing the non-LTE techniques, see, for instance, (Hubeny and Mihalas, 2015) and (Frish, 2021). Large matrix inversion costs make CM technique usage impractical in GCMs. We discussed this in the papers by Kutepov et al, 1998, Gusev and Kutepov, 2003, and in more details in KF23.

**Reply**. The original algorithm was developed based on the Curtis matrix (CM) approach and so we used it here. Nevertheless, we recall that the parameterization does not invert CMs. The coefficients are computed based on pre-calculated CMs but these are not inverted within the parameterization. On the other hand, in the revised manuscript, we have included a reference to your recent fast non-LTE model (KF23) and that it uses another non-LTE technique.

**2. How the 15-micron cooling maximum in the lower thermosphere is formed?**
In the manuscript: Above the mesopause, the cooling rate rapidly increases following the enhancement of the kinetic temperature. Above about 130 km, the cooling rates decline because of the depletion of the $CO_2$ vmr (see Fig. 7).
Are the authors sure about this? I recall that in early publications of the 1980s about the 15-micron cooling it was demonstrated that cooling declines at higher altitudes even for constant $CO_2$ vmr. Main problem at these altitudes is the rapid decrease of atmospheric pressure and the $CO(v_2)$ collisional quenching rate, which disconnects the 15-micron radiation from the atmospheric heat reservoir. If the authors do not agree with my comment, could they demonstrate that cooling stops decaying above 130 km when $CO_2$ vmr is constant?

**Reply**. We should have said here "**mainly** due to the $CO_2$ vmr fall". As you very well know, in this region we can assume the cool-to-space approximation, in which (see, e.g. Eq. 9.1 in Lopez-Puertas and Taylor (2001)) the cooling rate is proportional (when expressed in K/Day) to the $CO_2$ vmr, to the [O] concentration (density) and to temperature through $\exp(-E/kT)$. As altitude increases, the $CO_2$ VMR decreases, and so does the O density but T increases. So what we see at the end is the overall effect of the three quantities. We have explained this in the revised version.

**3. Why the CO2-O quenching rate coefficient ~ 6.0e-12 cm3 s −1 was used as the basic one for revision of the F98 parameterization?**
The authors write that they used for updating the F98 parameterization the quenching rate coefficients which are very similar to those used for its development *"... except the k CO2−O rate (process 1c in Table 1) that has been considered here with its upper limit. That is, about a factor of two larger than in the parameterization of Fomichev et al. (1998). This rate coefficient is not well known with uncertainties of the order of a factor of two (see, e.g., García-Comas et al., 2008). While laboratory measurements are in the range of 1.5 to 2·10−12 cm3 s −1 the values derived from atmospheric observations are close to 6·10−12 cm3 s −1. ...."*
First, measured in laboratory and retrieved from the atmospheric observation values of k differ not by a factor 2, but by 3-4. Additionally, if one accounts for the k retrievals by Feofilov et al, 2012, which involved the ground-based lidar temperature measurements, then this factor will be 3-6.

**Reply**. That is correct, taking the entire variation (equivalent to 3 sigmas), we have roughly a change from 1.5 to 6, e.g. a factor of 4, which would be +/- a factor of 2 for a 3-sigma error. We believe the higher range derived by Feofilov et al, 2012 also accounts for the errors in the measured temperatures and of the estimated uncertainty in [O], which was not measured; hence its larger uncertainty.

>>> Table 1 shows that the authors selected k ~ 6.0e-12 cm3 s −1 for updating the parameterization. Although k is supposed to be an input parameter in both F98 and L-P.23 routines, however, the previous one was optimized in the transition region from LTE to non-LTE for k ~ 3.0e-12 cm3 s −1, whereas new one is optimized for 6.0e-12 cm3 s −1. The authors tell in manuscript that this causes additional differences between F98 and L-P.23 in the transition area even when both apply the same rate coefficient, and then explain the reason why they selected higher rate for optimizing L-P.23 as *"... we have optimized it for the high value (see Table 1), as this value has been used in the most recent non-LTE retrievals of temperature from SABER and MIPAS measurements"*.
It is not enough, however, to say higher k was selected because this rate provides more reasonable T retrievals from space observation. As the authors know to validate these T measurements current GCMs apply twice lower rate 3.0e-12 cm3 s −1. If the authors recommend, which follows from the text, to use L-P.23 with the most high k, then can they

demonstrate how this affects the GCM (for instance the WACCM model) runs compared to those with twice lower k? Does this provide better fitting of measured temperatures?

**Reply**. Of course, it is questionable which value of K to use to optimize the parameterization (e.g. ~3e-12 or ~6e-12, as we can discard the lower laboratory values (~1.5e-12) because it does not fit well with atmospheric measurements. We could have chosen 3e-12 instead of 6e-12 but we did not, as explained above and in the manuscript. That is because the two major temperature databases of the upper mesosphere/lower thermosphere, SABER and MIPAS, (covering several years, more than 20 years in the case of SABER) both use a larger rate. The teams responsible for those retrievals made that decision based on the temperature validation performed for both datasets. Thus, García-Comas et al (2023) have shown that the agreement with other temperature measurements of independent "non-LTE" free instruments is better when this large rate is used. You mentioned the work of Feofilov et al, 2012, who also retrieved it from SABER and lidars temperature, and obtained a value for ko of 6.5e-12, even larger than that used here. The argument of using a lower value of ~3e-12 because that is being used in GCM models to reproduce the SABER and MIPAS T measurements we think is less substantiated and weaker. First, it is not clear that the models can reproduce the measured temperature field with that rate, see, e.g. Fig. 1 in Smith (2012). Secondly, the temperature computed by GCMs depends not only on this rate but also on many other factors like the parameterisation of the GWs, the chemistry (related mainly to O3 and O), vertical descent, etc. Further, they should also explain not only the temperature fields but also, for example, the CO2 and CO observations. A good example of the difficulty of e.g. WACCM in simulating those measured temperature fields can be found in this recent work:
https://essopenarchive.org/users/568957/articles/657910-a-novel-gravity-wave-transport-parametrization-for-global-chemistry-climate-models-description-and-validation.
In any case, the errors incurred by the parameterization if using the intermediate K rate do not change appreciably.

**4. Testing the parameterization for measured temperatures**

The Fig.21 of manuscript shows an example of the MIPAS nighttime temperature profiles (15 February 2009) used for verifying the parameterization accuracy. The authors note large variability of the measured temperature profiles. These individual profiles are the good inputs for the revised parameterization to show how it works for strongly disturbed T profiles.
The authors have obviously performed these tests, but they do not show these results. Instead, they write ***"The results are presented in Fig. 22 for the zonal mean of the differences for two days of solstice and two days of equinox conditions and in Fig. 23 as the global mean difference for all latitudes for each of the four individual days."*** Why only these mean values are shown? Obviously, the averaging smashes the errors obtained for individual profiles, for which we observed in our study of F98 parameterization presented in KF23 errors up to 25 K/day. Meanwhile these large errors in our study have generally concentrated in the altitude region around 90 km, exactly where the RMSs in Fig.25 of manuscript are maximized reaching 8-9 K/day. In our paper we explain why F98 works badly in this transition region. L-P.23 has the same problems and is nothing better.

**Reply**. We think we do appropriately present the results. On one hand, by the zonal mean of the differences, as shown in Fig. 22 (new Fig. 12), where the cooling rate differences are averaged over small latitude boxes (5°). And also by using the mean of the differences (to give an idea of "global" biases), and the RMS, as an appropriate descriptor of the error in the individual profiles.

**5. The accuracy of L-P.23 for smooth T profiles**

I wrote above that L-B.23 will not be any better than the F98 for disturbed T profiles. But how about the smoothed standard T profiles? The accuracy of cooling calculations with L-P.23 less than 0.5 K/Day for preindustrial CO2 for smooth profiles is also questionable.

(1) The non-LTE model, which is used in this study for the reference calculations to optimize L-P.23 and then to check its accuracy, includes only 18 15-micron bands. From my point of view this model itself is not accurate. In the routine we suggest in KF23, which utilizes the exact solution of the non-LTE problem, we use the same bands to calculate nighttime cooling for 400 ppm of CO2 with an error not higher than 1 K/day for any T profile, including disturbed by strong waves. For smooth profiles this error reduces to ~ 0.2 K/Day. These errors were estimated by comparing KF23 routine with our reference model, which comprises 60 vibration levels of 5 CO2 isotopic species and hundreds of bands. So, the exact algorithm compared to the exact algorithm showed 0.2 K/day error when a reduced set of bands was used. I doubt that the error of 0.5 K/day the authors report for the L-P.23 routine for smooth profiles after comparing it with a very simplified "reference model" is true. It must be higher.

**Reply**. Our reference non-LTE model is not simplified at all. We just dropped the contributions of bands beyond the listed 16 bands because we tested that their contribution in the non-LTE region is negligible (below 0.1 K/Day). Some of the not-included very weak bands may have some very small contributions in the LTE region, e.g. around the stratopause and below, but those contributions are not significant for the non-LTE region.

(2) The authors write that contribution of the heating due to the absorption of solar radiation in the near infrared CO2 bands at day time is negligible compared to the 15-micron cooling. However, 2-3 K/day (for current CO2) do not look negligible for the routine, which accuracy is declared to be about ~0.5 K/day. We tested in detail the (Ogibalov and Fomichev, 2003) parameterization of this heating, found this warming increasing for some wavy T profiles, and made sure this parameterization cannot guarantee the error of the KF23 routine at daytime to be lower than 1 K/day. As a result, we extended our KF23 daytime model up to 26 CO2 vibrational levels and 56 bands to satisfy this requirement.

**Reply**. It seems there is a misunderstanding here. We are NOT neglecting the cooling by the CO2 15 μm bands during the daytime. It is just that their contributions during the daytime are accounted for by the NIR heating routine. Hence, if included in this parameterization it would be included twice.
A note of clarification. Our day/night differences of the CO2 15 μm cooling rates are smaller than 1K/day for all pT reference profiles, at any altitude and for CO2 vmrs up to 5 times the pre-industrial value, except for MLS and SAS pT profiles near 105 km for CO2 vmr 4x and 5x the pre-industrial values.

(3) The authors say nothing about how they account for the cooling effect of the micro-scale sub-grid T disturbances. Kutepov et al., 2007 and Kutepov et al., 2013 showed that these temperature fluctuations cause near the mesopause an additional cooling up to 3 K day-1. I draw the authors' attention to the results shown by Kutepov et al, 2013 in Fig.1.5. It demonstrates one of the runs of the Leibniz Institute Middle Atmosphere (LIMA) model with the 15-micron cooling modified to account for the sub-grid T disturbances. It is shown that very minor variation of cooling (not higher than 2-3 K/day) lead to significant changes of the monthly and zonal mean temperatures for July 2005.
I am sure that errors of L-P.23 routine are much higher than 2-3 K/day and can be hardly reduced due to the deficiencies of the methodology applied. These errors will obviously have a strong impact on the GCM results.

**Reply**. Effectively, we do not include (e.g. do not provide a routine) the cooling rates induced by thermal structure at a grid smaller than the input grid (or, more properly, than the internal grid of the parameterisation). To properly account for them it would be necessary to know (or make assumptions) about how the temperature varies between grid points. We assume that the cooling induced by non-resolved GWs, propagating with a vertical wavelength of the order of/or

smaller than the parameterisation grid, would be taken into account in the GCMs by using an appropriate GW parameterisation (eg. see Intro of Kutepov et al, 2013). Currently, some new parameterizations are being developed to account for these effects (see, https://essopenarchive.org/users/568957/articles/657910-a-novel-gravity-wave-transport-parametrization-for-global-chemistry-climate-models-description-and-validation).
We have included a sentence in the revised version stating that our parameterization does not account for such effects.

**6. The code availability**
It seems the manuscript was submitted as the GMD "Development and technical paper". If it is correct, then **"The code should be made available, and a model availability paragraph must be included"**. The code is, however, not available.
Once the code is available, I will demonstrate that its errors are much higher than reported in the manuscript.

**Reply:** We were not aware of that and thought of providing the code during the review process. Provisionally, we already made available the parameterization as a Python routine at (https://zenodo.org/doi/10.5281/zenodo.10547026). Together with this revised version of the manuscript, we are now making it available in its Fortran version at https://zenodo.org/doi/10.5281/zenodo.10849969.
Further, we are also providing now some estimates of the computational cost (see new Sec. 6.3). It turns out that the routine takes only 1.5e-5 to 7.5e-5 s (15-75 µs) depending on the extension of the atmospheric profile, the processor and the Fortran compiler.

A model availability paragraph has also been included in the code availability section.

**References**

García-Comas, M., Funke, B., López-Puertas, M., Glatthor, N., Grabowski, U., Kellmann, S., Kiefer, M., Linden, A., Martínez-Mondéjar, B., Stiller, G. P., & von Clarmann, T. (2023). Version 8 IMK–IAA MIPAS temperatures from 12–15-µm spectra: Middle and Upper Atmosphere modes. *Atmospheric Measurement Techniques*, *16*(21), 5357–5386. https://doi.org/10.5194/amt-16-5357-2023

Smith, A. K. (2012). Global Dynamics of the MLT. *Surveys in Geophysics*, *33*(6), 1177–1230. https://doi.org/10.1007/s10712-012-9196-9

https://essopenarchive.org/users/568957/articles/657910-a-novel-gravity-wave-transport-parametrization-for-global-chemistry-climate-models-description-and-validation).